# Three-Way Trade-Off in Multi-Objective Learning: Optimization, Generalization and Conflict-Avoidance

**Lisha Chen**[†]
Rensselaer Polytechnic Institute
Troy, NY, United States
chenl21@rpi.edu

**Heshan Fernando**[†]
Rensselaer Polytechnic Institute
Troy, NY, United States
fernah@rpi.edu

**Yiming Ying**
University of Sydney
Camperdown, Australia
yiming.ying@sydney.edu.au

**Tianyi Chen**
Rensselaer Polytechnic Institute
Troy, NY, United States
chentianyi19@gmail.com

## Abstract

Multi-objective learning (MOL) often arises in emerging machine learning problems when multiple learning criteria or tasks need to be addressed. Recent works have developed various *dynamic weighting* algorithms for MOL, including MGDA and its variants, whose central idea is to find an update direction that *avoids conflicts* among objectives. Albeit its appealing intuition, empirical studies show that dynamic weighting methods may not always outperform static alternatives. To bridge this gap between theory and practice, we focus on a new variant of stochastic MGDA - the Multi-objective gradient with Double sampling (MoDo) algorithm and study its generalization performance and the interplay with optimization through the lens of algorithm stability. We find that the rationale behind MGDA – updating along conflict-avoidant direction - may *impede* dynamic weighting algorithms from achieving the optimal $\mathcal{O}(1/\sqrt{n})$ population risk, where $n$ is the number of training samples. We further highlight the variability of dynamic weights and their impact on the three-way trade-off among optimization, generalization, and conflict avoidance that is unique in MOL. Code is available at https://github.com/heshandevaka/Trade-Off-MOL.

## 1 Introduction

Multi-objective learning (MOL) emerges frequently in recent machine learning problems such as learning under fairness and safety constraints [49]; learning across multiple tasks, including multi-task learning [39] and meta-learning [47]; and, learning across multiple agents that may not share a global utility including federated learning [40] and multi-agent reinforcement learning [33].

This work considers solving the empirical version of MOL defined on the training dataset as $S = \{z_1, \ldots, z_n\}$. The performance of a model $x \in \mathbb{R}^d$ on a datum $z$ for the $m$-th objective is denoted as $f_{z,m} : \mathbb{R}^d \mapsto \mathbb{R}$, and its performance on the entire training dataset $S$ is measured by the $m$-th empirical objective $f_{S,m}(x)$ for $m \in [M]$. MOL optimizes the vector-valued objective, given by

$$\min_{x \in \mathbb{R}^d} \ F_S(x) \coloneqq [f_{S,1}(x), \ldots, f_{S,M}(x)]. \tag{1.1}$$

---

[†]Equal contribution.

The work of L. Chen, H. Fernando, and T. Chen was supported by the National Science Foundation (NSF) MoDL-SCALE project 2134168 and the RPI-IBM Artificial Intelligence Research Collaboration (AIRC). The work of Y. Ying was partially supported by NSF (DMS-2110836, IIS-2103450, and IIS-2110546).

37th Conference on Neural Information Processing Systems (NeurIPS 2023).

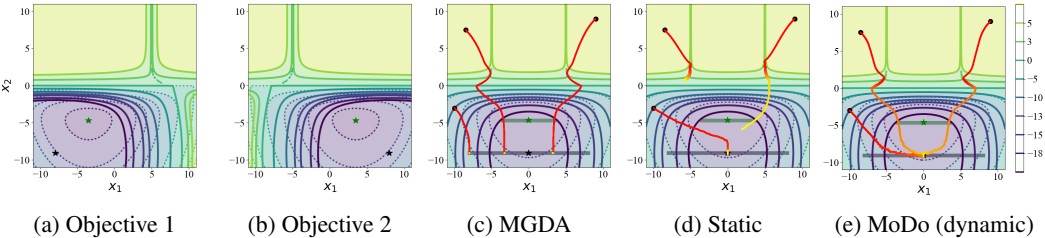

| (a) Objective 1 | (b) Objective 2 | (c) MGDA | (d) Static | (e) MoDo (dynamic) |

Figure 1: An example from [29] with two objectives (1a and 1b) to show the three-way trade-off in MOL. Figures 1c-1e show the optimization trajectories, where the **black** ● marks initializations of the trajectories, colored from **red** (start) to **yellow** (end). The background solid/dotted contours display the landscape of the average empirical/population objectives. The **gray**/**green** bar marks empirical/population Pareto front, and the **black** ⋆/**green** ⋆ marks solution to the average objectives.

One natural method for solving (1.1) is to optimize the (weighted) average of the multiple objectives, also known as *static or unitary weighting* [18, 45]. However, this method may face challenges due to *potential conflicts* among multiple objectives during the optimization process; e.g., conflicting gradient directions $\langle \nabla f_{S,m}(x), \nabla f_{S,m'}(x) \rangle < 0$. A popular alternative is thus to *dynamically weight* gradients from different objectives to avoid conflicts and obtain a direction $d(x)$ that optimizes all objective functions jointly that we call a *conflict-avoidant* (CA) direction. Algorithms in this category include the multi-gradient descent algorithm (MGDA) [9], its stochastic variants [30, 10, 52]. While the idea of finding CA direction in dynamic weighting-based approaches is very appealing, recent empirical studies reveal that dynamic weighting methods may not outperform static weighting in some MOL benchmarks [18, 45], especially when it involves stochastic updates and deep models. Specifically, observed by [18], the vanilla stochastic MGDA can be under-optimized, leading to larger training errors than static weighting. The reason behind this training performance degradation has been studied in [52, 10], which suggest the vanilla stochastic MGDA has biased updates, and propose momentum-based methods to address this issue. Nevertheless, in [45], it is demonstrated that the training errors of MGDA and static weighting are similar, while their main difference lies in the generalization performance. Unfortunately, the reason behind this testing performance degradation is not fully understood and remains an open question.

To gain a deeper understanding of the dynamic weighting-based algorithms, a natural question is

> **Q1:** *What are the major sources of errors in dynamic weighting-based MOL methods?*

To answer this question theoretically, we first introduce a proper measure of testing performance in MOL – the *Pareto stationary measure* in terms of the population objectives, which will immediately imply stronger measures such as Pareto optimality under strongly convex objectives. We then decompose this measure into *generalization* error and *optimization* error and further introduce a new metric on the *distance to CA directions* that is unique to MOL; see Sections 2.1 and 2.2.

To characterize the performance of MOL methods in a unified manner, we introduce a generic dynamic weighting-based MOL method that we term stochastic Multi-Objective gradient with DOuble sampling algorithm (**MoDo**), which uses a step size $\gamma$ to control the change of dynamic weights. Roughly speaking, by controlling $\gamma$, MoDo approximates MGDA (large $\gamma$) and static weighting algorithm ($\gamma = 0$) as two special cases; see Section 2.3. We first analyze the generalization error of the model learned by MoDo through the lens of algorithmic stability [3, 14, 24] in the framework of statistical learning theory. To our best knowledge, this is the *first-ever-known* stability analysis for MOL algorithms. Here the key contributions lie in defining a new notion of stability - MOL uniform stability and then establishing a tight upper bound (matching lower bound) on the MOL uniform stability for MoDo algorithm that involves two coupled sequences; see Section 3.1. We then analyze the optimization error of MoDo and its distance to CA directions, where the key contributions lie in relaxing *the bounded function value/gradient assumptions* and significantly improving the convergence rate of state-of-the-art dynamic weighting-based MOL methods [10]; see Section 3.2.

Different from the stability analysis for single-objective learning [14], the techniques used in our generalization and optimization analysis allow to remove conflicting assumptions and use larger step sizes to ensure both small generalization and optimization errors, which are of independent interest.

Given the holistic analysis of dynamic weighting methods provided in **Q1**, a follow-up question is

> **Q2:** *What may cause the empirical performance degradation of dynamic weighting methods?*

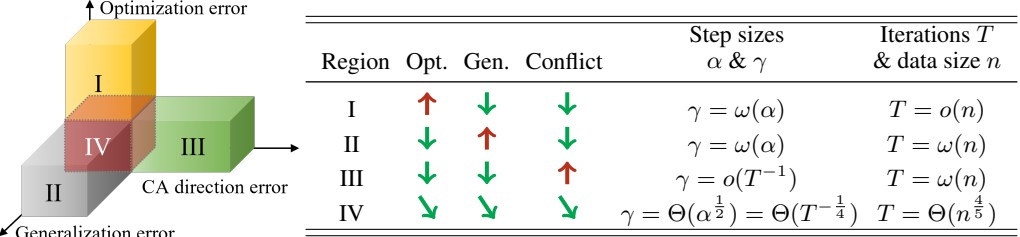

| | | | | Step sizes | Iterations $T$ |
| Region | Opt. | Gen. | Conflict | $\alpha$ & $\gamma$ | & data size $n$ |
|---|---|---|---|---|---|
| I | ↑ | ↓ | ↓ | $\gamma = \omega(\alpha)$ | $T = o(n)$ |
| II | ↓ | ↑ | ↓ | $\gamma = \omega(\alpha)$ | $T = \omega(n)$ |
| III | ↓ | ↓ | ↑ | $\gamma = o(T^{-1})$ | $T = \omega(n)$ |
| IV | ↘ | ↘ | ↘ | $\gamma = \Theta(\alpha^{\frac{1}{2}}) = \Theta(T^{-\frac{1}{4}})$ | $T = \Theta(n^{\frac{4}{5}})$ |

Figure 2: An illustration of three-way trade-off among optimization, generalization, and conflict avoidance in the strongly convex case; $\alpha$ is the step size for $x$, $\gamma$ is the step size for weights $\lambda$, where $o(\cdot)$ denotes a strictly slower growth rate, $\omega(\cdot)$ denotes a strictly faster growth rate, and $\Theta(\cdot)$ denotes the same growth rate. Arrows ↓ and ↑ respectively represent diminishing in an optimal rate and growing in a fast rate w.r.t. $n$, while ↘ represents diminishing w.r.t. $n$, but not in an optimal rate.

*Visualizing MOL solution concepts.* To reason the root cause for this, we first compare different MOL algorithms in a toy example shown in Figure 1. We find MGDA can navigate along CA directions and converge to the empirical Pareto front under all initializations, while static weighting gets stuck in some initializations; at the same time, the empirical Pareto solution obtained by MGDA may incur a larger population risk than the suboptimal empirical solution obtained by the static weighting method; finally, if the step size $\gamma$ of dynamic weights is carefully tuned, MoDo can converge along CA directions to the empirical Pareto optimal solution that also generalizes well.

Aligned with this toy example, our theoretical results suggest a novel *three-way trade-off* in the performance of dynamic weighting-based MOL algorithm; see Section 3.3. Specifically, it suggests that the step size for dynamic weighting $\gamma$ plays a central role in the trade-off among convergence to CA direction, convergence to empirical Pareto stationarity, and generalization error; see Figure 2. In this sense, MGDA has an edge in convergence to the CA direction but it could sacrifice generalization; the static weighting method cannot converge to the CA direction but guarantees convergence to the empirical Pareto solutions and their generalization. Our analysis also suggests that MoDo achieves a small population risk under a proper combination of step sizes and the number of iterations.

## 2 Problem Formulation and Target of Analysis

In this section, we first introduce the problem formulation of MOL, the target of analysis, the metric to measure its generalization, and then present the MGDA algorithm and its stochastic variant.

### 2.1 Preliminaries of MOL

Denote the vector-valued objective function on datum $z$ as $F_z(x) = [f_{z,1}(x), \ldots, f_{z,M}(x)]$. The training and testing performance of $x$ can then be measured by the empirical objective $F_S(x)$ and the population objective $F(x)$ which are, respectively, defined as $F_S(x) := \frac{1}{n} \sum_{i=1}^{n} F_{z_i}(x)$ and $F(x) := \mathbb{E}_{z \sim \mathcal{D}}[F_z(x)]$. Their corresponding gradients are denoted as $\nabla F_S(x)$ and $\nabla F(x) \in \mathbb{R}^{d \times M}$.

Analogous to the stationary solution and optimal solution in single-objective learning, we define the Pareto stationary point and Pareto optimal solution for MOL problem $\min_{x \in \mathbb{R}^d} F(x)$ as follows.

**Definition 1** (Pareto stationary and Pareto optimal). *If there exists a convex combination of the gradient vectors that equals to zero, i.e., there exists $\lambda \in \Delta^M$ such that $\nabla F(x)\lambda = 0$, then $x \in \mathbb{R}^d$ is Pareto stationary. If there is no $x \in \mathbb{R}^d$ and $x \neq x^*$ such that, for all $m \in [M]$ $f_m(x) \leq f_m(x^*)$, with $f_{m'}(x) < f_{m'}(x^*)$ for at least one $m' \in [M]$, then $x^*$ is Pareto optimal. If there is no $x \in \mathbb{R}^d$ such that for all $m \in [M]$, $f_m(x) < f_m(x^*)$, then $x^*$ is weakly Pareto optimal.*

By definition, at a Pareto stationary solution, there is no common descent direction for all objectives. A necessary and sufficient condition for $x$ being Pareto stationary for smooth objectives is that $\min_{\lambda \in \Delta^M} \|\nabla F(x)\lambda\| = 0$. Therefore, $\min_{\lambda \in \Delta^M} \|\nabla F(x)\lambda\|$ can be used as a measure of Pareto stationarity (PS) [9, 11, 42, 30, 10]. We will refer to the aforementioned quantity as the *PS population risk* henceforth and its empirical version as *PS empirical risk* or *PS optimization error*. We next introduce the target of our analysis based on the above definitions.

### 2.2 Target of analysis and error decomposition

In existing generalization analysis for MOL, measures based on function values have been used to derive generalization guarantees in terms of Pareto optimality [7, 41]. However, for general

nonconvex smooth MOL problems, it can only be guaranteed for an algorithm to converge to Pareto stationarity of the empirical objective, i.e., a small $\min_{\lambda \in \Delta^M} \|\nabla F_S(x)\lambda\|$ [9, 11, 30, 10]. Thus, it is not reasonable to measure population risk in terms of Pareto optimality in this case. Furthermore, when all the objectives are convex or strongly convex, Pareto stationarity is a sufficient condition for weak Pareto optimality or Pareto optimality, respectively, as stated in Proposition 1.

**Proposition 1** ([42, Lemma 2.2])**.** *If $f_m(x)$ are convex or strongly-convex for all $m \in [M]$, and $x \in \mathbb{R}^d$ is a Pareto stationary point of $F(x)$, then $x$ is weakly Pareto optimal or Pareto optimal.*

Next, we proceed to decompose the PS population risk.

**Error Decomposition.** Given a model $x$, the PS population risk can be decomposed into

$$\underbrace{\min_{\lambda \in \Delta^M} \|\nabla F(x)\lambda\|}_{\text{PS population risk } R_{\text{pop}}(x)} = \underbrace{\min_{\lambda \in \Delta^M} \|\nabla F(x)\lambda\| - \min_{\lambda \in \Delta^M} \|\nabla F_S(x)\lambda\|}_{\text{PS generalization error } R_{\text{gen}}(x)} + \underbrace{\min_{\lambda \in \Delta^M} \|\nabla F_S(x)\lambda\|}_{\text{PS optimization error } R_{\text{opt}}(x)} \quad (2.1)$$

where the optimization error quantifies the training performance, i.e., how well does model $x$ perform on the training data; and the generalization error (gap) quantifies the difference between the testing performance on new data sampled from $\mathcal{D}$ and the training performance, i.e., how well the model $x$ performs on unseen testing data compared to the training data.

Let $A : \mathcal{Z}^n \mapsto \mathbb{R}^d$ denote a randomized MOL algorithm. Given training data $S$, we are interested in the expected performance of the output model $x = A(S)$, which is measured by $\mathbb{E}_{A,S}[R_{\text{pop}}(A(S))]$. From (2.1) and linearity of expectation, it holds that

$$\mathbb{E}_{A,S}[R_{\text{pop}}(A(S))] = \mathbb{E}_{A,S}[R_{\text{gen}}(A(S))] + \mathbb{E}_{A,S}[R_{\text{opt}}(A(S))]. \quad (2.2)$$

**Distance to CA direction.** As demonstrated in Figure 1, the key merit of dynamic weighting over static weighting algorithms lies in its ability to navigate through conflicting gradients. Consider an update direction $d = -\nabla F_S(x)\lambda$, where $\lambda$ is the dynamic weights from a simplex $\lambda \in \Delta^M := \{\lambda \in \mathbb{R}^M \mid \mathbf{1}^\top \lambda = 1, \ \lambda \geq 0\}$. To obtain such a steepest CA direction in unconstrained learning that maximizes the minimum descent of all objectives, we can solve the following problem [11]

$$\text{CA direction} \quad d(x) = \underset{d \in \mathbb{R}^d}{\arg\min} \ \underset{m \in [M]}{\max} \left\{ \langle \nabla f_{S,m}(x), d \rangle + \frac{1}{2}\|d\|^2 \right\} \quad (2.3a)$$

$$\overset{\text{equivalent to}}{\Longleftrightarrow} d(x) = -\nabla F_S(x)\lambda^*(x) \ \text{ s.t. } \ \lambda^*(x) \in \underset{\lambda \in \Delta^M}{\arg\min} \|\nabla F_S(x)\lambda\|^2. \quad (2.3b)$$

Defining $d_\lambda(x) = -\nabla F_S(x)\lambda$ given $x \in \mathbb{R}^d$ and $\lambda \in \Delta^M$, we measure the distance to $d(x)$ via [10]

$$\text{CA direction error} \qquad \mathcal{E}_{\text{ca}}(x, \lambda) := \|d_\lambda(x) - d(x)\|^2. \quad (2.4)$$

With the above definitions of measures that quantify the performance of algorithms in different aspects, we then introduce a stochastic gradient algorithm for MOL that is analyzed in this work.

## 2.3 A stochastic algorithm for MOL

MGDA finds $\lambda^*(x)$ in (2.3b) using the full-batch gradient $\nabla F_S(x)$, and then constructs $d(x) = -\nabla F_S(x)\lambda^*(x)$, a CA direction for all empirical objectives $f_{S,m}(x)$. However, in practical statistical learning settings, the full-batch gradient $\nabla F_S(x)$ may be costly to obtain, and thus one may resort to a stochastic estimate of $\nabla F_S(x)$ instead. The direct stochastic counterpart of MGDA, referred to as the stochastic multi-gradient algorithm in [30], replaces the full-batch gradients $\nabla f_{S,m}(x)$ in (2.3b) with their

---

**Algorithm 1** Stochastic MGDA - MoDo algorithm

1: **input** Training data $S$, initial model $x_0$, weighting coefficient $\lambda_0$, and their learning rates $\{\alpha_t\}_{t=0}^T, \{\gamma_t\}_{t=0}^T$.
2: **for** $t = 0, \ldots, T-1$ **do**
3:     **for** objective $m = 1, \ldots, M$ **do**
4:         Independent gradients $\nabla f_{m,z_{t,s}}(x_t), \ \ s \in [3]$
5:     **end for**
6:     Compute dynamic weight $\lambda_{t+1}$ following (2.5a)
7:     Update model parameter $x_{t+1}$ following (2.5b)
8: **end for**
9: **output** $x_T$

---

stochastic approximations $\nabla f_{z,m}(x)$ for $z \in S$, which, however, introduces a biased stochastic estimate of $\lambda_{t+1}^*$, thus a biased CA direction; see [10, Section 2.3].

Table 1: Comparison of optimization error, generalization error, and population risk under different assumptions for static and dynamic weighting. Use "NC", "SC" to represent nonconvex and strongly convex, and "Lip-C", "S" to represent Lipschitz continuous and smooth, respectively.

| Assumption | Method | Optimization | Generalization | Risk | CA Distance |
|---|---|---|---|---|---|
| NC, Lip-C, S | Static | $(\alpha T)^{-\frac{1}{2}} + \alpha^{\frac{1}{2}}$ | $T^{\frac{1}{2}} n^{-\frac{1}{2}}$ | $n^{-\frac{1}{6}}$ | ✗ |
| | Dynamic | $(\alpha T)^{-\frac{1}{2}} + \alpha^{\frac{1}{2}} + \gamma^{\frac{1}{2}}$ | $T^{\frac{1}{2}} n^{-\frac{1}{2}}$ | $n^{-\frac{1}{6}}$ | $(\gamma T)^{-1} + \alpha^{\frac{1}{2}} \gamma^{-\frac{1}{2}} + \gamma$ |
| SC, S | Static | $(\alpha T)^{-\frac{1}{2}} + \alpha^{\frac{1}{2}}$ | $n^{-\frac{1}{2}}$ | $n^{-\frac{1}{2}}$ | ✗ |
| | Dynamic | $(\alpha T)^{-\frac{1}{2}} + \alpha^{\frac{1}{2}} + \gamma^{\frac{1}{2}}$ | $\begin{cases} n^{-\frac{1}{2}}, & \gamma = \mathcal{O}(T^{-1}) \\ T^{\frac{1}{2}} n^{-\frac{1}{2}}, & \text{o.w.} \end{cases}$ | $\begin{cases} n^{-\frac{1}{2}} \\ n^{-\frac{1}{6}} \end{cases}$ | $(\gamma T)^{-1} + \alpha^{\frac{1}{2}} \gamma^{-\frac{1}{2}} + \gamma$ |

To provide a tight analysis, we introduce a simple yet theoretically grounded stochastic variant of MGDA - stochastic Multi-Objective gradient with DOuble sampling algorithm (MoDo). MoDo obtains an unbiased stochastic estimate of the gradient of problem (2.3b) through double (independent) sampling and iteratively updates $\lambda$, because $\mathbb{E}_{z_{t,1}, z_{t,2}}[\nabla F_{z_{t,1}}(x_t)^\top \nabla F_{z_{t,2}}(x_t)\lambda_t] = \nabla F_S(x_t)^\top \nabla F_S(x_t)\lambda_t$. At each iteration $t$, denote $z_{t,s}$ as an independent sample from $S$ with $s \in [3]$, and $\nabla F_{z_{t,s}}(x_t)$ as a stochastic estimate of $\nabla F_S(x_t)$. MoDo updates $x_t$ and $\lambda_t$ as

$$\lambda_{t+1} = \Pi_{\Delta^M} \left(\lambda_t - \gamma_t \nabla F_{z_{t,1}}(x_t)^\top \nabla F_{z_{t,2}}(x_t)\lambda_t\right) \tag{2.5a}$$

$$x_{t+1} = x_t - \alpha_t \nabla F_{z_{t,3}}(x_t)\lambda_{t+1} \tag{2.5b}$$

where $\alpha_t, \gamma_t$ are step sizes, and $\Pi_{\Delta^M}(\cdot)$ denotes Euclidean projection to the simplex $\Delta^M$. We have summarized the MoDo algorithm in Algorithm 1 and will focus on MoDo in the subsequent analysis.

## 3 Optimization, Generalization and Three-Way Trade-Off

This section presents the theoretical analysis of the PS population risk associated with the MoDo algorithm, where the analysis of generalization error is in Section 3.1 and that of optimization error is in Section 3.2. A summary of our main results is given in Table 1.

### 3.1 Multi-objective generalization and uniform stability

We first bound the expected PS generalization error by the generalization in gradients in Proposition 2, then introduce the MOL uniform stability and establish its connection to the generalization in gradients. Finally, we bound the MOL uniform stability.

**Proposition 2.** *With $\|\cdot\|_F$ denoting the Frobenious norm, $R_{\text{gen}}(A(S))$ in (2.2) can be bounded by*

$$\mathbb{E}_{A,S}[R_{\text{gen}}(A(S))] \leq \mathbb{E}_{A,S}[\|\nabla F(A(S)) - \nabla F_S(A(S))\|_F]. \tag{3.1}$$

With Proposition 2, next we introduce the concept of MOL uniform stability tailored for MOL problems and show that PS generalization error in MOL can be bounded by the MOL uniform stability. Then we analyze their bound in general nonconvex case and strongly convex case, respectively.

**Definition 2** (MOL uniform stability). *A randomized algorithm $A : \mathcal{Z}^n \mapsto \mathbb{R}^d$, is MOL-uniformly stable with $\epsilon_F$ if for all neighboring datasets $S, S'$ that differ in at most one sample, we have*

$$\sup_z \mathbb{E}_A\left[\|\nabla F_z(A(S)) - \nabla F_z(A(S'))\|_F^2\right] \leq \epsilon_F^2. \tag{3.2}$$

Next we show the relation between the upper bound of PS generalization error in (3.1) and MOL uniform stability in Proposition 3.

**Proposition 3** (MOL uniform stability and generalization). *Assume for any $z$, the function $F_z(x)$ is differentiable. If a randomized algorithm $A : \mathcal{Z}^n \mapsto \mathbb{R}^d$ is MOL-uniformly stable with $\epsilon_F$, then*

$$\mathbb{E}_{A,S}[\|\nabla F(A(S)) - \nabla F_S(A(S))\|_F] \leq 4\epsilon_F + \sqrt{n^{-1}\mathbb{E}_S\left[\mathbb{V}_{z\sim\mathcal{D}}(\nabla F_z(A(S)))\right]} \tag{3.3}$$

*where the variance is defined as $\mathbb{V}_{z\sim\mathcal{D}}(\nabla F_z(A(S))) = \mathbb{E}_{z\sim\mathcal{D}}\left[\|\nabla F_z(A(S)) - \mathbb{E}_{z\sim\mathcal{D}}[\nabla F_z(A(S))]\|_F^2\right]$.*

Proposition 3 establishes a connection between the upper bound of the PS generalization error and the MOL uniform stability, where the former can be bounded above by the latter plus the variance of the stochastic gradient over the population data distribution. It is worth noting that the standard arguments of bounding the generalization error measured in function values by the uniform stability measured in function values [14, Theorem 2.2] is not applicable here as the summation and norm operators are not exchangeable. More explanations are given in the proof in Appendix B.1.

**Theorem 1** (PS generalization error of MoDo in nonconvex case). *If $\sup_z \mathbb{E}_A \left[ \|\nabla F_z(A(S))\|_{\mathrm{F}}^2 \right] \leq G^2$ for any $S$, then the MOL uniform stability, i.e., $\epsilon_{\mathrm{F}}^2$ in Definition 2 is bounded by $\epsilon_{\mathrm{F}}^2 \leq 4G^2 T/n$. And the PS generalization error $\mathbb{E}_{A,S}[R_{\mathrm{gen}}(A(S))] = \mathcal{O}(T^{\frac{1}{2}} n^{-\frac{1}{2}})$.*

Compared to the function value uniform stability upper bound in [14, Theorem 3.12] for nonconvex single-objective learning, Theorem 1 does not require a step size decay $\alpha_t = \mathcal{O}(1/t)$, thus can enjoy at least a polynomial convergence rate of optimization errors w.r.t. $T$. Combining Theorem 1 with Proposition 3, to ensure the generalization error is diminishing with $n$, one needs to choose $T = o(n)$, which lies in the "early stopping" regime and results in potentially large optimization error. We then provide a tighter bound in the strongly convex case that allows a larger choice of $T$. Below we list the standard assumptions used to derive the introduced MOL stability.

**Assumption 1** (Lipschitz continuity of $\nabla F_z(x)$). *For all $m \in [M]$, $\nabla f_{z,m}(x)$ is $\ell_{f,1}$-Lipschitz continuous for all $z$. And $\nabla F_z(x)$ is $\ell_{F,1}$-Lipschitz continuous in Frobenius norm for all $z$.*

**Assumption 2.** *For all $m \in [M]$, $z \in \mathcal{Z}$, $f_{z,m}(x)$ is $\mu$-strongly convex w.r.t. $x$, with $\mu > 0$.*

Note that in the strongly convex case, the gradient norm $\|\nabla F_z(x)\|_{\mathrm{F}}$ can be unbounded in $\mathbb{R}^d$. Therefore, one cannot assume Lipschitz continuity of $f_{z,m}(x)$ w.r.t. $x \in \mathbb{R}^d$. We address this challenge by showing that $\{x_t\}$ generated by the MoDo algorithm is bounded as stated in Lemma 1. Notably, combining with Assumption 1, we can derive that the gradient norm $\|\nabla F_z(x_t)\|_{\mathrm{F}}$ is also bounded, which serves as a stepping stone to derive the MOL stability bound.

**Lemma 1** (Boundedness of $x_t$ for strongly convex and smooth objectives). *Suppose Assumptions 1, 2 hold. For $\{x_t\}, t \in [T]$ generated by MoDo algorithm or other dynamic weighting algorithm with weight $\lambda \in \Delta^M$, step size $\alpha_t = \alpha$, and $0 \leq \alpha \leq \ell_{f,1}^{-1}$, there exists a finite positive constant $c_x$ such that $\|x_t\| \leq c_x$. And there exists finite positive constants $\ell_f, \ell_F = \sqrt{M}\ell_f$, such that for all $\lambda \in \Delta^M$, we have $\|\nabla F(x_t)\lambda\| \leq \ell_f$, and $\|\nabla F(x_t)\|_{\mathrm{F}} \leq \ell_F$.*

With Lemma 1, the stability bound and PS generalization is provided below.

**Theorem 2** (PS generalization error of MoDo in strongly convex case). *Suppose Assumptions 1, 2 hold. Let $A$ be the MoDo algorithm (Algorithm 1). For the MOL uniform stability $\epsilon_F$ of algorithm $A$ in Definition 2, if the step sizes satisfy $0 < \alpha_t \leq \alpha \leq 1/(2\ell_{f,1})$, and $0 < \gamma_t \leq \gamma \leq \min\{\frac{\mu^2}{120\ell_f^2 \ell_{g,1}}, \frac{1}{8(3\ell_f^2 + 2\ell_{g,1})}\}/T$, then it holds that*

$$\epsilon_{\mathrm{F}}^2 \leq \frac{48}{\mu n} \ell_f^2 \ell_{F,1}^2 \Big( \alpha + \frac{12 + 4M\ell_f^2}{\mu n} + \frac{10M\ell_f^4 \gamma}{\mu} \Big) \quad and \quad \mathbb{E}_{A,S}[R_{\mathrm{gen}}(A(S))] = \mathcal{O}(n^{-\frac{1}{2}}). \quad (3.4)$$

*And there exist functions $F_z(x)$ that satisfy Assumptions 1, 2, neighboring datasets $S, S'$ that differ in at most one sample, and MoDo algorithm with step sizes $0 < \alpha_t \leq \alpha \leq 1/(2\ell_{f,1})$, and $0 < \gamma_t \leq \gamma \leq \min\{\frac{\mu^2}{120\ell_f^2 \ell_{g,1}}, \frac{1}{8(3\ell_f^2 + 2\ell_{g,1})}\}/T$ such that*

$$\mathbb{E}_A \left[ \|\nabla F_z(A(S)) - \nabla F_z(A(S'))\|_{\mathrm{F}}^2 \right] \geq \frac{M\mu^2}{256n^2}. \quad (3.5)$$

Theorem 2 provides both upper and lower bounds for the MOL uniform stability. In this case, we choose $\alpha = \Theta(T^{-\frac{1}{2}})$, $\gamma = o(T^{-1})$, and $T = \Theta(n^2)$ to minimize the PS population risk upper bound, as detailed in Section 3.3. With this choice, the MOL uniform stability upper bound matches the lower bound in an order of $n^{-2}$, suggesting that our bound is tight. The generalization error bound in (3.4) is a direct implication from the MOL uniform stability bound in (3.4), Propositions 2, and 3.

It states that the PS generalization error of MoDo is $\mathcal{O}(n^{-\frac{1}{2}})$, which matches the generalization error of static weighting up to a constant coefficient [22]. Our result also indicates that when all the objectives are strongly convex, choosing small step sizes $\alpha$ and $\gamma$ can benefit the generalization error.

## 3.2 Multi-objective optimization error

In this section, we bound the multi-objective PS optimization error $\min_{\lambda \in \Delta^M} \|\nabla F_S(x)\lambda\|$ [9, 30, 10]. As discussed in Section 2.2, this measure being zero implies the model $x$ achieves a Pareto stationarity for the empirical problem.

Below we list an additional standard assumption used to derive the optimization error.

**Assumption 3** (Lipschitz continuity of $F_z(x)$). *For all $m \in [M]$, $f_{z,m}(x)$ are $\ell_f$-Lipschitz continuous for all $z$. Then $F_z(x)$ are $\ell_F$-Lipschitz continuous in Frobenius norm for all $z$ with $\ell_F = \sqrt{M}\ell_f$.*

---

**Lemma 2** (Distance to CA direction). *Suppose either: 1) Assumptions 1, 3 hold; or 2) Assumptions 1, 2 hold, with $\ell_f$ and $\ell_F$ defined in Lemma 1. For $\{x_t\}, \{\lambda_t\}$ generated by MoDo, it holds that*

$$\frac{1}{T} \sum_{t=0}^{T-1} \mathbb{E}_A[\|d_{\lambda_t}(x_t) - d(x_t)\|^2] \leq \frac{4}{\gamma T} + 6\sqrt{M\ell_{f,1}^2 \ell_f^2 \frac{\alpha}{\gamma}} + \gamma M \ell_f^4. \qquad (3.6)$$

---

Lemma 2 analyzes convergence to the CA direction using the measure introduced in Section 2.2. By, e.g., choosing $\alpha = \Theta(T^{-\frac{3}{4}})$, and $\gamma = \Theta(T^{-\frac{1}{4}})$, the RHS of (3.6) converges in a rate of $\mathcal{O}(T^{-\frac{1}{4}})$.

---

**Theorem 3** (PS optimization error of MoDo). *Suppose either: 1) Assumptions 1, 3 hold; or, 2) Assumptions 1, 2 hold, with $\ell_f$ defined in Lemma 1. Define $c_F$ such that $\mathbb{E}_A[F_S(x_0)\lambda_0] - \min_{x \in \mathbb{R}^d} \mathbb{E}_A[F_S(x)\lambda_0] \leq c_F$. Considering $\{x_t\}$ generated by MoDo (Algorithm 1), with $\alpha_t = \alpha \leq 1/(2\ell_{f,1})$, $\gamma_t = \gamma$, then under either condition 1) or 2), it holds that*

$$\frac{1}{T} \sum_{t=0}^{T-1} \mathbb{E}_A\left[ \min_{\lambda \in \Delta^M} \|\nabla F_S(x_t)\lambda\| \right] \leq \sqrt{\frac{c_F}{\alpha T}} + \sqrt{\frac{3}{2}\gamma M \ell_f^4} + \sqrt{\frac{1}{2}\alpha \ell_{f,1} \ell_f^2}. \qquad (3.7)$$

---

The choice of step sizes $\alpha = \Theta(T^{-\frac{3}{4}})$, and $\gamma = \Theta(T^{-\frac{1}{4}})$ to ensure convergence to CA direction is suboptimal for the convergence to Pareto stationarity (see Theorem 3), exhibiting a trade-off between convergence to the CA direction and convergence to Pareto stationarity; see discussion in Section 3.3.

## 3.3 Optimization, generalization and conflict avoidance trade-off

Combining the results in Sections 3.1 and 3.2, we are ready to analyze and summarize the three-way trade-off of MoDo in MOL. With $A_t(S) = x_t$ denoting the output of algorithm $A$ at the $t$-th iteration, we can decompose the PS population risk $R_{\text{pop}}(A_t(S))$ as (cf. (2.1) and (3.1))

$$\mathbb{E}_{A,S}\big[R_{\text{pop}}(A_t(S))\big] \leq \mathbb{E}_{A,S}\Big[ \min_{\lambda \in \Delta^M} \|\nabla F_S(A_t(S))\lambda\| \Big] + \mathbb{E}_{A,S}\Big[\|\nabla F(A_t(S)) - \nabla F_S(A_t(S))\|_{\text{F}}\Big].$$

**The general nonconvex case.** Suppose Assumptions 1, 3 hold. By the generalization error in Theorem 1, and the optimization error bound in Theorem 3, the PS population risk of the output of MoDo can be bounded by

$$\frac{1}{T} \sum_{t=0}^{T-1} \mathbb{E}_{A,S}\big[R_{\text{pop}}(A_t(S))\big] = \mathcal{O}\left( \alpha^{-\frac{1}{2}}T^{-\frac{1}{2}} + \alpha^{\frac{1}{2}} + \gamma^{\frac{1}{2}} + T^{\frac{1}{2}}n^{-\frac{1}{2}} \right). \qquad (3.8)$$

*Discussion of trade-off.* Choosing step sizes $\alpha = \Theta(T^{-\frac{1}{2}})$, $\gamma = \Theta(T^{-\frac{1}{2}})$, and number of steps $T = \Theta(n^{\frac{2}{3}})$, then the expected PS population risk is $\mathcal{O}(n^{-\frac{1}{6}})$, which matches the PS population risk upper bound of a general nonconvex single objective in [22]. A clear trade-off in this case is between the optimization error and generalization error, controlled by $T$. Indeed, increasing $T$ leads to smaller optimization errors but larger generalization errors, and vice versa. To satisfy convergence to CA

direction, it requires $\gamma = \omega(\alpha)$ based on Lemma 2, and the optimization error in turn becomes worse, so does the PS population risk. Specifically, choosing $\alpha = \Theta(T^{-\frac{1}{2}})$, $\gamma = \Theta(T^{-\frac{1}{4}})$, and $T = \Theta(n^{\frac{4}{5}})$ leads to the expected PS population risk in $\mathcal{O}(n^{-\frac{1}{10}})$, and the distance to CA direction in $\mathcal{O}(n^{-\frac{1}{10}})$. This shows another trade-off between conflict avoidance and optimization error.

**The strongly convex case.** Suppose Assumptions 1, 2 hold. By the generalization error and the optimization error given in Theorems 2 and 3, MoDo's PS population risk can be bounded by

$$\frac{1}{T} \sum_{t=0}^{T-1} \mathbb{E}_{A,S}\left[R_{\text{pop}}(A_t(S))\right] = \mathcal{O}\left(\alpha^{-\frac{1}{2}} T^{-\frac{1}{2}} + \alpha^{\frac{1}{2}} + \gamma^{\frac{1}{2}} + n^{-\frac{1}{2}}\right). \tag{3.9}$$

*Discussion of trade-off.* Choosing step sizes $\alpha = \Theta(T^{-\frac{1}{2}})$, $\gamma = o(T^{-1})$, and number of steps $T = \Theta(n^2)$, we have the expected PS population risk in gradients is $\mathcal{O}(n^{-\frac{1}{2}})$. However, choosing $\gamma = o(T^{-1})$ leads to large distance to the CA direction according to Lemma 2 because the term $\frac{4}{\gamma T}$ in (3.6) increases with $T$. To ensure convergence to the CA direction, it requires $\gamma = \omega(T^{-1})$, under which the tighter bound in Theorem 2 does not hold but the bound in Theorem 1 still holds. In this case, the PS population risk under proper choices of $\alpha, \gamma, T$ is $\mathcal{O}(n^{-\frac{1}{6}})$ as discussed in the previous paragraph. Therefore, to avoid conflict of gradients, one needs to sacrifice the sample complexity of PS population risk, demonstrating a trade-off between conflict avoidance and PS population risk.

## 4 Related Works and Our Technical Contributions

**Multi-task learning (MTL).** MTL, as one application of MOL, leverages shared information among different tasks to train a model that can perform multiple tasks. MTL has been widely applied to natural language processing, computer vision, and robotics [15, 38, 50, 43]. From the optimization perspective, a simple method for MTL is to take the weighted average of the per-task losses as the objective. However, as studied in [16], the static weighting method is not able to find all the Pareto optimal models in general. Alternatively, the weights can be updated dynamically during optimization, and the weights for different tasks can be chosen based on different criteria such as uncertainty [17], gradient norms [6], or task difficulty [13]. These methods are often heuristic and designed for specific applications. Another line of work tackles MTL through MOL [39, 48, 29, 12]. A foundational algorithm in this regard is MGDA [9], which takes dynamic weighting of gradients to obtain a CA direction for all objectives. Stochastic versions of MGDA have been proposed in [30, 52, 10]. Besides finding one single model, algorithms for finding a set of Pareto optimal models rather than one have been proposed in [28, 35, 31, 20, 46, 51, 32, 27, 34].

**Theory of MOL.** *Optimization* convergence analysis for the deterministic MGDA algorithm has been provided in [11]. Later on, stochastic variants of MGDA were introduced [30, 52, 10]. However, the vanilla stochastic MGDA introduces biased estimates of the dynamic weight, resulting in biased estimates of the CA directions during optimization. To address this, Liu et al. [30] proposed to increase the batch size during optimization, Zhou et al. [52] and Fernando et al. [10] proposed to use momentum-based bias reduction techniques. Compared to these works on the optimization analysis, we have improved the assumptions and/or the final convergence rate of the PS optimization error. A detailed comparison is summarized in Table 2. While the community has a rich history of investigating the optimization of MOL algorithms, their *generalization* guarantee remains unexplored until recently. In [7], a min-max formulation to solve the MOL problem is analyzed, where the weights are chosen based on the maximum function values, rather than the CA direction. More recently, [41] provides generalization guarantees for MOL for a more general class of weighting. These two works analyze generalization based on the Rademacher complexity of the hypothesis class, with generalization bound independent of the training process. Different from these works, we use algorithm stability to derive the first algorithm-dependent generalization error bounds, highlighting the effect of the training dynamics. In contrast to previous MOL theoretical works that focus solely on either optimization [10, 52] or generalization [7, 41], we propose a holistic framework to analyze three types of errors, namely, optimization, generalization, and CA distance in MOL with an instantiation of the MoDo algorithm. This allows us to study the impact of the hyperparameters on the theoretical testing performance, and their optimal values to achieve the best trade-off among the errors. Our theory and algorithm design can also be applied to algorithms such as PCGrad [48], CAGrad [29], and GradNorm [6]. Specifically, for example, the implementation of CAGrad takes iterative updates of the dynamic weight using a single stochastic estimate, resulting in a biased estimate of the update

Table 2: Comparison with prior stochastic MOL algorithms in terms of assumptions and the guarantees of the three errors, where logarithmic dependence is omitted, and Opt., CA dist., and Gen. are short for optimization error, CA distance, and generalization error, respectively.

| Algorithm | Batch size | NC | Lipschitz $\lambda^*(x)$ | Bounded function | Opt. | CA dist. | Gen. |
|---|---|---|---|---|---|---|---|
| SMG [30, Thm 5.3] | $\mathcal{O}(t)$ | ✗ | ✓ | ✗ | $T^{-\frac{1}{8}}$ | - | - |
| CR-MOGM [52, Thm 3] | $\mathcal{O}(1)$ | ✓ | ✗ | ✓ | $T^{-\frac{1}{4}}$ | - | - |
| MoCo [10, Thm 2] | $\mathcal{O}(1)$ | ✓ | ✗ | ✗ | $T^{-\frac{1}{20}}$ | $T^{-\frac{1}{5}}$ | - |
| MoCo [10, Thm 4] | $\mathcal{O}(1)$ | ✓ | ✗ | ✓ | $T^{-\frac{1}{4}}$ | $\mathcal{O}(1)$ | - |
| **MoDo (Ours, Thms 1,2,3)** | $\mathcal{O}(1)$ | ✓ | ✗ | ✗ | $T^{-\frac{1}{4}}$ | $\mathcal{O}(1)$ | $T^{\frac{1}{2}}n^{-\frac{1}{2}}$ |
| **MoDo (Ours, Thms 1,2,3)** | $\mathcal{O}(1)$ | ✓ | ✗ | ✗ | $T^{-\frac{1}{8}}$ | $T^{-\frac{1}{4}}$ | $T^{\frac{1}{2}}n^{-\frac{1}{2}}$ |

direction, thus no guarantee of convergence for the stochastic algorithm. This issue can be addressed by the double sampling technique introduced in this paper. In addition, our analysis techniques have been applied to improve the analysis of convergence rates for other algorithms [44, 5].

**Algorithm stability and generalization.** Stability analysis dates back to the work [8] in 1970s. Uniform stability and its relationship with generalization were studied in [3] for the exact minimizer of the ERM problem with strongly convex objectives. The work [14] pioneered the stability analysis for stochastic gradient descent (SGD) algorithms with convex and smooth objectives. The results were extended and refined in [19] with data-dependent bounds, in [4, 37, 23] for non-convex objectives, and in [1, 24] for SGD with non-smooth and convex losses. However, all these studies mainly focus on single-objective learning problems. To our best knowledge, there is no existing work on the stability and generalization analysis for multi-objective learning problems and our results on its stability and generalization are the *first-ever-known* ones.

**Challenges and contributions.** Our contributions are highly non-trivial as summarized below.
• The definition of PS testing risk in gradient (2.1) is unique in MOL, and overcomes the unnecessarily small step sizes usually brought by the classical function value-based risk analysis. Specifically, prior stability analysis in function values for single objective learning [14] requires $1/t$ step size decay in the nonconvex case, otherwise, the generalization error bound will depend exponentially on the number of iterations. However such step sizes lead to very slow convergence of the optimization error. This is addressed by the definitions of gradient-based measures and sampling-determined MOL algorithms, which yield stability bounds in $\mathcal{O}(T/n)$ without any step size decay. See Theorem 1.
• The stability of the dynamic weighting algorithm in the strongly convex (SC) case is non-trivial compared to single objective learning [14] because it involves two coupled sequences during the update. As a result, the classical contraction property for the update function of the model parameters that are often used to derive stability does not hold. This is addressed by controlling the change of $\lambda_t$ by the step size $\gamma$, and using mathematical induction to derive a tighter bound. See Appendix B.4.
• In the SC case with an unbounded domain, the function is not Lipschitz or the gradients are unbounded, which violates the commonly used bounded gradient assumption for proving the stability and optimization error. We relax this assumption by proving that the iterates generated by dynamic weighting algorithms in the SC case are bounded on the optimization trajectory in Lemma 1.

## 5 Experiments

In this section, we conduct experiments to further demonstrate the three-way trade-off among the optimization, generalization, and conflict avoidance of the MoDo algorithm. An average of 10 random seeds with 0.5 standard deviation is reported if not otherwise specified.

### 5.1 Synthetic experiments

Our theory in the SC case is first verified through a synthetic experiment; see the details in Appendix D.1. Figure 3a shows the PS optimization error and PS population risk, as well as the distance to CA direction, decreases as $T$ increases, which corroborates Lemma 2, and Theorem 3. In addition, the generalization error, in this case, does not vary much with $T$, verifying Theorems 2. In Figure 3b, the optimization error first decreases and then increases as $\alpha$ increases, which is consistent with Theorem 3. Notably, we observe a threshold for $\alpha$ below which the distance to the CA direction converges even when the optimization error does not converge, while beyond which the distance to the CA direction becomes larger, verifying Lemma 2. Additionally, Figure 3c demonstrates that

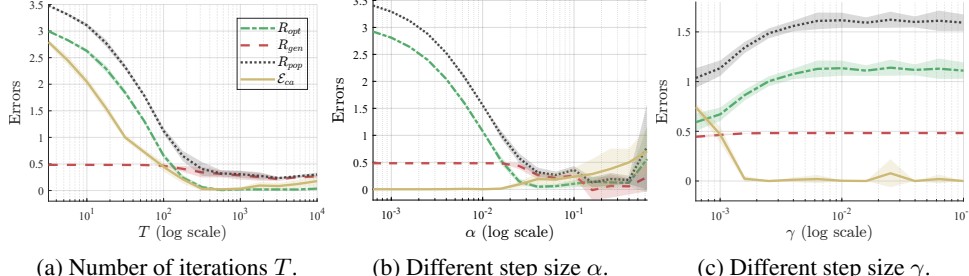

(a) Number of iterations $T$.  (b) Different step size $\alpha$.  (c) Different step size $\gamma$.

Figure 3: Optimization, generalization, and CA direction errors of MoDo in the strongly convex case under different $T, \alpha, \gamma$. The default parameters are $T = 100$, $\alpha = 0.01$, $\gamma = 0.001$.

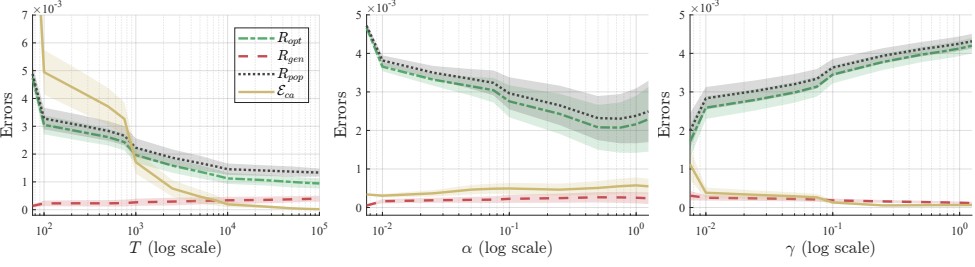

(a) Number of iterations $T$  (b) Different step size $\alpha$  (c) Different step size $\gamma$

Figure 4: Optimization, generalization, and CA direction errors of MoDo for MNIST image classification under different $T$, $\alpha$, and $\gamma$. The default parameters are $T = 1000$, $\alpha = 0.1$, and $\gamma = 0.01$.

increasing $\gamma$ enlarges the PS optimization error, PS generalization error, and thus the PS population risk, but decreases the distance to CA direction, which supports Lemma 2.

## 5.2 Image classification experiments

We further verify our theory in the NC case on MNIST image classification [21] using a multi-layer perceptron and three objectives: cross-entropy, mean squared error (MSE), and Huber loss. Following Section 2.2, we evaluate the performance in terms of $R_{\text{pop}}(x)$, $R_{\text{opt}}(x)$, $R_{\text{gen}}(x)$, and $\mathcal{E}_{\text{ca}}(x, \lambda)$. The exact PS population risk $R_{\text{pop}}(x)$ is not accessible without the true data distribution. To estimate the PS population risk, we evaluate $\min_{\lambda \in \Delta^M} \|\nabla F_{S_{\text{te}}}(x)\lambda\|$ on the testing data set $S_{\text{te}}$ that is independent of training data set $S$. The PS optimization error $R_{\text{opt}}(x)$ is obtained by $\min_{\lambda \in \Delta^M} \|\nabla F_S(x)\lambda\|$, and the PS generalization error $R_{\text{gen}}(x)$ is estimated by $\min_{\lambda \in \Delta^M} \|\nabla F_{S_{\text{te}}}(x)\lambda\| - R_{\text{opt}}(x)$.

We examine the impact of different $T, \alpha, \gamma$ on the errors in Figure 4. Figure 4a shows that increasing $T$ reduces optimization error and CA direction distance but increases generalization error, aligning with Theorems 1, 2, and 3. Figure 4b shows that increasing $\alpha$ leads to an initial decrease and subsequent increase

Table 3: Classification results on Office-31 dataset.

| Method | Amazon | DSLR | Webcam | $\Delta\mathcal{A}\% \downarrow$ |
|--------|--------|------|--------|-----------|
|  | Test Acc ↑ | Test Acc ↑ | Test Acc ↑ | |
| Static | $84.62 \pm 0.71$ | $94.43 \pm 0.96$ | $97.44 \pm 1.20$ | $2.56 \pm 0.37$ |
| MGDA | $79.45 \pm 0.11$ | $\mathbf{96.56 \pm 1.20}$ | $\mathbf{97.89 \pm 0.74}$ | $3.65 \pm 0.64$ |
| **MoDo** | $\mathbf{85.13 \pm 0.58}$ | $95.41 \pm 0.98$ | $96.78 \pm 0.65$ | $\mathbf{2.26 \pm 0.31}$ |

in PS optimization error and population risk. which aligns with Theorem 3 and (3.8). On the other hand, there is an overall increase in CA direction distance with $\alpha$, which aligns with Theorem 2. Figure 4c shows that increasing $\gamma$ increases both the PS population and optimization errors but decreases CA direction distance. This matches our bounds for PS optimization error in Theorem 3, PS population risk in (3.8), and CA direction distance in Theorem 2.

## 6 Conclusions

This work studies the three-way trade-off in MOL – among optimization, generalization, and conflict avoidance. Our results show that, in the general nonconvex setting, the traditional trade-off between optimization and generalization depending on the number of iterations also exists in MOL. Moreover, dynamic weighting algorithms like MoDo introduce a new dimension of trade-off in terms of conflict avoidance compared to static weighting. We demonstrate that this three-way trade-off can be controlled by the step size for updating the dynamic weighting parameter and the number of iterations. Proper choice of these parameters can lead to decent performance on all three metrics.

## Broader impacts and limitations

This work has a potential impact on designing dynamic weighting algorithms and choosing hyper-parameters such as step sizes and number of iterations based on the trade-off for MTL applications such as multi-language translation, and multi-agent reinforcement learning. No ethical concerns arise from this work. A limitation of this study is that the theory focuses on a specific algorithm, MoDo, for smooth objectives in unconstrained learning. Future research could explore the theory of other algorithms for non-smooth objectives or constrained learning, which would be interesting directions to pursue.

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

# Appendix for "Three-Way Trade-Off in Multi-Objective Learning: Optimization, Generalization and Conflict-Avoidance "

## Table of Contents

## A  Notations

A summary of notations used in this work is listed in Table 4 for ease of reference.

Table 4: Notations and their descriptions.

| Notations | Descriptions |
|---|---|
| $x \in \mathbb{R}^d$ | Model parameter, or decision variable |
| $z \in \mathcal{Z}$ | Data point for training or testing |
| $S \in \mathcal{Z}^n$ | Dataset such that $S = \{z_1, \ldots, z_n\}$ |
| $f_{z,m}(x),\, f_{S,m}(x)$ | A scalar-valued objective function evaluated on data point $z$, with $f_{z,m} : \mathbb{R}^d \mapsto \mathbb{R}$, or on dataset $S$, $f_{S,m}$, with $f_{S,m} \coloneqq \frac{1}{|S|} \sum_{z \in S} f_{z,m}(x)$ |
| $f_m(x)$ | A scalar-valued population objective function, $f_m(x) \coloneqq \mathbb{E}_z[f_{z,m}(x)]$ |
| $\nabla f_m(x)$ | Gradient of $f_m(x)$, with $\nabla f_m(x) : \mathbb{R}^d \mapsto \mathbb{R}^d$ |
| $F_z(x),\, F_S(x)$ | A vector-valued objective function evaluated on data point $z$, with $F_z : \mathbb{R}^d \mapsto \mathbb{R}^M$, or on dataset $S$, with $F_S \coloneqq \frac{1}{|S|} \sum_{z \in S} F_z(x)$ |
| $F(x)$ | A vector-valued population objective, $F(x) \coloneqq \mathbb{E}_z[f_{z,m}(x)]$ |
| $\nabla F(x)$ | Gradient of $F(x)$, with $\nabla F(x) : \mathbb{R}^d \mapsto \mathbb{R}^{d \times M}$ |
| $\lambda \in \Delta^M$ | Weighting parameter in an $(M-1)$-simplex |
| $\lambda_\rho^*(x) \in \Delta^M$ | CA weight, optimal solution to (2.3), when $\rho = 0$, it is simplified as $\lambda^*(x)$ |
| $\mathbf{1} \in \mathbb{R}^M$ | All-one vector with dimension $M$ |
| $\alpha$ | Step size to update model parameter $x$ |
| $\gamma$ | Step size to update weight $\lambda$ |

In the proof, we use $\| \cdot \|$ to denote the spectral norm, and $\| \cdot \|_{\mathrm{F}}$ to denote the Frobenius norm.

# B Bounding the generalization error

## B.1 Proof of Propositions 2-3

In this subsection, we prove Propositions 2-3, which establishes the relation between PS generalization error and MOL uniform stability.

**Proof of Proposition 2.** For a given model $x$, it holds that

$$
\begin{aligned}
R_{\mathrm{gen}}(x) &= \min_{\lambda \in \Delta^M} \|\nabla F(x)\lambda\| - \min_{\lambda \in \Delta^M} \|\nabla F_S(x)\lambda\| \\
&= -\max_{\lambda \in \Delta^M} -\|\nabla F(x)\lambda\| + \max_{\lambda \in \Delta^M} -\|\nabla F_S(x)\lambda\| \\
&\overset{(a)}{\leq} \max_{\lambda \in \Delta^M} (\|\nabla F(x)\lambda\| - \|\nabla F_S(x)\lambda\|) \overset{(b)}{\leq} \max_{\lambda \in \Delta^M} (\|(\nabla F(x) - \nabla F_S(x))\lambda\|) \\
&\overset{(c)}{\leq} \max_{\lambda \in \Delta^M} (\|\nabla F(x) - \nabla F_S(x)\|_{\mathrm{F}} \|\lambda\|_{\mathrm{F}}) \leq \|\nabla F(x) - \nabla F_S(x)\|_{\mathrm{F}}
\end{aligned}
\tag{B.1}
$$

where $(a)$ follows from the subadditivity of max operator, $(b)$ follows from triangle inequality, $(c)$ follows from Cauchy-Schwartz inequality.

Setting $x = A(S)$, and taking expectation over $A, S$ on both sides of the above inequality, we have

$$
\mathbb{E}_{A,S}[R_{\mathrm{gen}}(A(S))] \leq \mathbb{E}_{A,S}[\|\nabla F(A(S)) - \nabla F_S(A(S))\|_{\mathrm{F}}].
\tag{B.2}
$$

$\square$

**Proof of Proposition 3.** The proof extends that of [22] for single objective learning to our MOL setting. Recall that $S = \{z_1, \ldots, z_n\}$, which are drawn i.i.d. from the data distribution $\mathcal{D}$. Define the perturbed dataset $S^{(i)} = \{z_1, \ldots, z_i', \ldots, z_n\}$ sampled i.i.d. from $\mathcal{D}$ with $z_i'$ independent of $z_j$, for all $i, j \in [n]$. Let $\tilde{z}$ be an independent sample of $z_j, z_j'$, for all $j \in [n]$, and from the same distribution $\mathcal{D}$. We first decompose the difference of population gradient and empirical gradient on the algorithm output $n(\nabla F(A(S)) - \nabla F_S(A(S)))$ as follows using the gradient on $A(S^{(i)})$. Since $\mathbb{E}_{\tilde{z}}[\nabla F_{\tilde{z}}(A(S))] = \nabla F(A(S))$, it holds that

$$
n\big(\nabla F(A(S)) - \nabla F_S(A(S))\big) = n\mathbb{E}_{\tilde{z}}[\nabla F_{\tilde{z}}(A(S))] - n\nabla F_S(A(S))
$$

$$
= n\mathbb{E}_{\tilde{z}}[\nabla F_{\tilde{z}}(A(S))] - \left(\sum_{i=1}^{n} \nabla F_{z_i}(A(S))\right) + \sum_{i=1}^{n} \left(\mathbb{E}_{z_i'}[\nabla F(A(S^{(i)}))] - \mathbb{E}_{z_i'}[\nabla F(A(S^{(i)}))]\right)
$$

$$
+ \sum_{i=1}^{n} \left(\mathbb{E}_{z_i'}[\nabla F_{z_i}(A(S^{(i)}))] - \mathbb{E}_{z_i'}[\nabla F_{z_i}(A(S^{(i)}))]\right)
$$

$$
= \sum_{i=1}^{n} \mathbb{E}_{\tilde{z}, z_i'}[\nabla F_{\tilde{z}}(A(S)) - \nabla F_{\tilde{z}}(A(S^{(i)}))] + \sum_{i=1}^{n} \underbrace{\mathbb{E}_{z_i'}[\mathbb{E}_{\tilde{z}}[\nabla F_{\tilde{z}}(A(S^{(i)}))] - \nabla F_{z_i}(A(S^{(i)}))]}_{\xi_i(S)}
$$

$$
+ \sum_{i=1}^{n} \mathbb{E}_{z_i'}\big[\nabla F_{z_i}(A(S^{(i)})) - \nabla F_{z_i}(A(S))\big]
\tag{B.3}
$$

where the last equality follows from rearranging and that $z_i, z_i', \tilde{z}$ are mutually independent. Applying triangle inequality to (B.3), it then follows that

$$
n\|\nabla F(A(S)) - \nabla F_S(A(S))\|_{\mathrm{F}} \leq \sum_{i=1}^{n} \mathbb{E}_{\tilde{z}, z_i'}[\|\nabla F_{\tilde{z}}(A(S)) - \nabla F_{\tilde{z}}(A(S^{(i)}))\|_{\mathrm{F}}] + \left\|\sum_{i=1}^{n} \xi_i(S)\right\|_{\mathrm{F}}
$$

$$
+ \sum_{i=1}^{n} \mathbb{E}_{z_i'}[\|\nabla F_{z_i}(A(S^{(i)})) - \nabla F_{z_i}(A(S))\|_{\mathrm{F}}].
\tag{B.4}
$$

Note $S$ and $S^{(i)}$ differ by a single sample. By Definition 2, the MOL uniform stability $\epsilon_{\mathrm{F}}$, and Jensen's inequality, we further get

$$
n\mathbb{E}\left[\|\nabla F(A(S)) - \nabla F_S(A(S))\|_{\mathrm{F}}\right] \leq 2n\epsilon_{\mathrm{F}} + \mathbb{E}\left[\left\|\sum_{i=1}^{n} \xi_i(S)\right\|_{\mathrm{F}}\right].
\tag{B.5}
$$

We then proceed to bound $\mathbb{E}\left[\left\|\sum_{i=1}^{n}\xi_i(S)\right\|_{\mathrm{F}}\right]$, which satisfies

$$\left(\mathbb{E}\left[\left\|\sum_{i=1}^{n}\xi_i(S)\right\|_{\mathrm{F}}\right]\right)^2 \leq \mathbb{E}\left[\left\|\sum_{i=1}^{n}\xi_i(S)\right\|_{\mathrm{F}}^2\right] = \sum_{i=1}^{n}\underbrace{\mathbb{E}\left[\|\xi_i(S)\|_{\mathrm{F}}^2\right]}_{J_{1,i}} + \sum_{i,j\in[n]:i\neq j}\underbrace{\mathbb{E}[\langle\xi_i(S),\xi_j(S)\rangle]}_{J_{2,i,j}}.$$
(B.6)

For $J_{1,i}$, according to the definition of $\xi_i(S)$ in (B.3) and Jensen inequality, it holds that

$$\begin{aligned}
J_{1,i} = \mathbb{E}[\|\xi_i(S)\|_{\mathrm{F}}^2] &= \mathbb{E}\left[\left\|\mathbb{E}_{z_i'}\left[\mathbb{E}_{\tilde{z}}[\nabla F_{\tilde{z}}(A(S^{(i)}))] - \nabla F_{z_i}(A(S^{(i)}))\right]\right\|_{\mathrm{F}}^2\right] \\
&\overset{(a)}{\leq} \mathbb{E}\left[\left\|\mathbb{E}_{\tilde{z}}[\nabla F_{\tilde{z}}(A(S^{(i)}))] - \nabla F_{z_i}(A(S^{(i)}))\right\|_{\mathrm{F}}^2\right] \\
&\overset{(b)}{=} \mathbb{E}\left[\left\|\mathbb{E}_{\tilde{z}}[\nabla F_{\tilde{z}}(A(S))] - \nabla F_{z_i'}(A(S))\right\|_{\mathrm{F}}^2\right] \\
&= \mathbb{E}\left[\mathbb{V}_{\tilde{z}}(\nabla F_{\tilde{z}}(A(S)))\right],
\end{aligned}$$
(B.7)

where $(a)$ follows from Jensen's inequality, $(b)$ follows from the symmetry between $z_i$ and $z_i'$. To bound $J_{2,i,j}$ with $i \neq j$, further introduce $S'' = \{z_1'', \ldots, z_n''\}$ which are drawn i.i.d. from the data distribution $\mathcal{D}$. Then for each $i, j \in [n]$ with $i \neq j$, introduce $S_j$ as a neighboring dataset of $S$ by replacing its $z_j$ with $z_j''$, and $S_j^{(i)}$ as a neighboring dataset of $S^{(i)}$ by replacing its $z_j$ with $z_j''$, i.e.,

$$S_j = \{z_1, \ldots, z_{j-1}, z_j'', z_{j+1}, \ldots, z_n\},$$
(B.8a)

$$S_j^{(i)} = \{z_1, \ldots, z_{i-1}, z_i', z_{i+1}, \ldots, z_{j-1}, z_j'', z_{j+1}, \ldots, z_n\}.$$
(B.8b)

Then the idea is to bound $J_{2,i,j}$ using the newly introduced neighboring datasets $S_j$ and $S_j^{(i)}$, so as to connect to the definition of the stability $\epsilon_{\mathrm{F}}$. We first show that $\mathbb{E}\left[\langle\xi_i(S), \xi_j(S)\rangle\right] = \mathbb{E}\left[\langle\xi_i(S) - \xi_i(S_j), \xi_j(S) - \xi_j(S_i)\rangle\right]$ because for $i \neq j$,

$$\mathbb{E}\left[\langle\xi_i(S_j), \xi_j(S)\rangle\right] \overset{(c)}{=} 0, \quad \mathbb{E}\left[\langle\xi_i(S_j), \xi_j(S_i)\rangle\right] \overset{(d)}{=} 0, \quad \mathbb{E}\left[\langle\xi_i(S_j), \xi_j(S_i)\rangle\right] \overset{(e)}{=} 0. \quad (\text{B.9})$$

For $i \neq j$, $(c)$ follows from

$$\mathbb{E}\left[\langle\xi_i(S_j), \xi_j(S)\rangle\right] = \mathbb{E}\mathbb{E}_{z_j}\left[\langle\xi_i(S_j), \xi_j(S)\rangle\right] = \mathbb{E}\left[\langle\xi_i(S_j), \mathbb{E}_{z_j}[\xi_j(S)]\rangle\right] = 0, \quad (\text{B.10})$$

where the second identity holds since $\xi_i(S_j)$ is independent of $z_j$ and the last identity follows from $\mathbb{E}_{z_j}[\xi_j(S)] = 0$ due to the symmetry between $\tilde{z}$ and $z_i$, and their independence with $S^{(i)}$, derived as

$$\mathbb{E}_{z_i}[\xi_i(S)] = \mathbb{E}_{z_i}\left[\mathbb{E}_{z_i'}[\mathbb{E}_{\tilde{z}}[\nabla F_{\tilde{z}}(A(S^{(i)}))] - \nabla F_{z_i}(A(S^{(i)}))]\right] = 0, \quad \forall i \in [n]. \quad (\text{B.11})$$

In a similar way, for $i \neq j$, $(d)$ and $(e)$ follow from

$$\mathbb{E}\left[\langle\xi_i(S), \xi_j(S_i)\rangle\right] = \mathbb{E}\mathbb{E}_{z_i}\left[\langle\xi_i(S), \xi_j(S_i)\rangle\right] = \mathbb{E}\left[\langle\xi_j(S_i), \mathbb{E}_{z_i}[\xi_i(S)]\rangle\right] = 0, \quad (\text{B.12})$$

$$\mathbb{E}\left[\langle\xi_i(S_j), \xi_j(S_i)\rangle\right] = \mathbb{E}\mathbb{E}_{z_i}\left[\langle\xi_i(S_j), \xi_j(S_i)\rangle\right] = \mathbb{E}\left[\langle\xi_j(S_i), \mathbb{E}_{z_i}[\xi_i(S_j)]\rangle\right] = 0. \quad (\text{B.13})$$

Based on (B.9), for $i \neq j$ we have

$$\begin{aligned}
J_{2,i,j} &= \mathbb{E}\left[\langle\xi_i(S), \xi_j(S)\rangle\right] = \mathbb{E}\left[\langle\xi_i(S) - \xi_i(S_j), \xi_j(S) - \xi_j(S_i)\rangle\right] \\
&\leq \mathbb{E}\left[\|\xi_i(S) - \xi_i(S_j)\|_{\mathrm{F}}\|\xi_j(S) - \xi_j(S_i)\|_{\mathrm{F}}\right] \\
&\leq \frac{1}{2}\mathbb{E}\left[\|\xi_i(S) - \xi_i(S_j)\|_{\mathrm{F}}^2\right] + \frac{1}{2}\mathbb{E}\left[\|\xi_j(S) - \xi_j(S_i)\|_{\mathrm{F}}^2\right]
\end{aligned}$$
(B.14)

where we have used $ab \leq \frac{1}{2}\left(a^2 + b^2\right)$. According to the definition of $\xi_i(S)$ and $\xi_i(S_j)$ we know the following identity for $i \neq j$

$$\begin{aligned}
\mathbb{E}\left[\|\xi_i(S) - \xi_i(S_j)\|_{\mathrm{F}}^2\right] = \mathbb{E}\Big[\Big\|&\mathbb{E}_{z_i'}\mathbb{E}_{\tilde{z}}\left[\nabla F_{\tilde{z}}(A(S^{(i)})) - \nabla F_{\tilde{z}}(A(S_j^{(i)}))\right] \\
&+ \mathbb{E}_{z_i'}\left[\nabla F_{z_i}(A(S_j^{(i)})) - \nabla F_{z_i}(A(S^{(i)}))\right]\Big\|_{\mathrm{F}}^2\Big].
\end{aligned}$$
(B.15)

It then follows from the inequality $(a + b)^2 \leq 2\left(a^2 + b^2\right)$ and the Jensen's inequality that

$$\mathbb{E}[\|\xi_i(S) - \xi_i(S_j)\|_{\mathrm{F}}^2] \leq 2\mathbb{E}[\|\nabla F_{\tilde{z}}(A(S^{(i)})) - \nabla F_{\tilde{z}}(A(S_j^{(i)}))\|_{\mathrm{F}}^2]$$
$$+ 2\mathbb{E}[\|\nabla F_{z_i}(A(S_j^{(i)})) - \nabla F_{z_i}(A(S^{(i)}))\|_{\mathrm{F}}^2]. \qquad \text{(B.16)}$$

Since $S^{(i)}, S_j^{(i)}$ and $S^{(j)}, S_i^{(j)}$ are two pairs of neighboring datasets, it follows from the definition of stability that

$$\mathbb{E}\left[\|\xi_i(S) - \xi_i(S_j)\|_{\mathrm{F}}^2\right] \leq 4\epsilon_{\mathrm{F}}^2, \text{ and } \mathbb{E}\left[\|\xi_j(S) - \xi_j(S_i)\|_{\mathrm{F}}^2\right] \leq 4\epsilon_{\mathrm{F}}^2, \quad \forall i \neq j. \qquad \text{(B.17)}$$

We can plug the above inequalities back into (B.14) and bound $J_{2,i,j}$ by

$$J_{2,i,j} = \mathbb{E}\left[\langle\xi_i(S), \xi_j(S)\rangle\right] \leq 4\epsilon_{\mathrm{F}}^2, \quad \forall i \neq j. \qquad \text{(B.18)}$$

Combining the bound for $J_{1,i}$ in (B.7) and $J_{2,i,j}$ in (B.18) and substituting them back into (B.6), it then follows that

$$\mathbb{E}\left[\left\|\sum_{i=1}^n \xi_i(S)\right\|_{\mathrm{F}}^2\right] = \mathbb{E}\left[\sum_{i=1}^n \|\xi_i(S)\|_{\mathrm{F}}^2\right] + \sum_{i,j\in[n]:i\neq j} \mathbb{E}[\langle\xi_i(S), \xi_j(S)\rangle]$$
$$\leq n\mathbb{E}\left[\mathbb{V}_{\tilde{z}}(\nabla F_{\tilde{z}}(A(S)))\right] + 4n(n-1)\epsilon_{\mathrm{F}}^2. \qquad \text{(B.19)}$$

Plugging the above inequality back into (B.5), using the subadditivity of the square root function, we get

$$n\mathbb{E}[\|\nabla F(A(S)) - \nabla F_S(A(S))\|_{\mathrm{F}}] \leq 4n\epsilon_{\mathrm{F}} + \sqrt{n\mathbb{E}\left[\mathbb{V}_{\tilde{z}}(\nabla F_{\tilde{z}}(A(S)))\right]}. \qquad \text{(B.20)}$$

The proof is complete. $\qquad\qquad\square$

## B.2 Proof of Theorem 1 – PS generalization error in nonconvex case

In this subsection, we prove Theorem 1, which establishes the PS generalization error of MoDo in the nonconvex case.

**Organization of proof.** To prove the PS generalization error of MoDo, we first define the concept of Sampling-determined algorithms in Definition 3. This concept has been defined in [22] for the analysis in single-objective learning. Then we show that MoDo is sampling-determined in Proposition 4. Finally, combining Propositions 2-4, we can prove Theorem 1, the MOL uniform stability and PS generalization error of MoDo.

**Definition 3** (Sampling-determined algorithm [22]). *Let $A$ be a randomized algorithm that randomly chooses an index sequence $I(A) = \{i_{t,s}\}$ to compute stochastic gradients. We say a symmetric algorithm $A$ is sampling-determined if the output model is fully determined by $\{z_i : i \in I(A)\}$.*

**Proposition 4** (MoDo is sampling determined). *MoDo (Algorithm 1) is sampling determined. In other words, Let $I(A) = \{i_t\}$ be the sequence of index chosen by these algorithms from training set $S = \{z_1, \ldots, z_n\}$, and $z_i \overset{\text{i.i.d.}}{\sim} \mathcal{P}$ for all $i \in [n]$ to build stochastic gradients, the output $A(S)$ is determined by $\{z_j : j \in I(A)\}$. To be precise, $A(S)$ is independent of $z_j$ if $j \notin I(A)$.*

**Proof of Proposition 4.** Let $I(A) = \{I_1, \ldots, I_T\}$, $I_t = \{i_{t,s}\}_{s=1}^3$ and $i_{t,s} \in [n]$ for all $1 \leq t \leq T$. And $S_{I(A)} = \{z_{i_{t,s}}\}$. By the description in Algorithm 1, $A(S) = G_{z_{I_T}} \circ \cdots \circ G_{z_{I_1}}(x_0)$, where $G_z(\cdot)$ is the stochastic update function of the model parameter given random sample $z$. Therefore, for all possible sample realization $z$, we have

$$\mathbb{P}(A(S) = x \mid z_j = z, j \notin I(A)) = \mathbb{P}(G_{z_{I_T}} \circ \cdots \circ G_{z_{I_1}}(x_0) = x \mid z_j = z, j \notin I(A))$$
$$= \mathbb{P}(G_{z_{I_T}} \circ \cdots \circ G_{z_{I_1}}(x_0) = x \mid j \notin I(A))$$
$$= \mathbb{P}(A(S) = x \mid j \notin I(A)) \qquad \text{(B.21)}$$

where the last equality holds because $z_j \notin S_{I(A)}$, and $z_j$ is independent of all elements in $S_{I(A)}$ by i.i.d. sampling. Therefore, $A(S)$ is independent of $z_j$ if $j \notin I(A)$. The proof is complete. $\qquad\square$

Note that, besides MoDo, other popular stochastic randomized MTL algorithms such as SMG [30] and MoCo [10] are also sampling-determined. Therefore, the result is also applicable to these algorithms.

**Lemma 3** ([22, Theorem 5 (b)]). *Let $A$ be a sampling-determined random algorithm (Definition 3) and $S, S'$ be neighboring datasets with $n$ data points that differ only in the $i$-th data point. If $\sup_z \mathbb{E}_A \left[ \|\nabla F_z(A(S))\|_{\mathrm{F}}^2 \mid i \in I(A) \right] \leq G^2$ for any $S$, then*

$$\sup_z \mathbb{E}_A[\|\nabla F_z(A(S)) - \nabla F_z(A(S'))\|_{\mathrm{F}}^2] \leq 4G^2 \cdot \mathbb{P}\{i \in I(A)\}. \tag{B.22}$$

**Proof of Theorem 1.** From Proposition 4, algorithm $A$, MoDo is sampling-determined. Then based on Lemma 3, its MOL uniform stability in Definition 2 can be bounded by

$$\epsilon_{\mathrm{F}}^2 \leq 4G^2 \cdot \mathbb{P}\{i \in I(A)\}. \tag{B.23}$$

Let $i_t$ be the index of the sample selected by $A$ at the $t$-th step, and $i^*$ be the index of the data point that is different in $S$ and $S'$. Then

$$\mathbb{P}\{i^* \in I(A)\} \leq \sum_{t=0}^{T-1} \mathbb{P}\{i_t = i^*\} \leq \frac{T}{n}. \tag{B.24}$$

Combining (B.23) and (B.24) gives

$$\epsilon_{\mathrm{F}}^2 \leq \frac{4G^2 T}{n}. \tag{B.25}$$

Then based on Propositions 2-3, we have

$$\begin{aligned}
\mathbb{E}_{A,S}[R_{\mathrm{gen}}(A(S))] &\leq \mathbb{E}_{A,S}[\|\nabla F(A(S)) - \nabla F_S(A(S))\|_{\mathrm{F}}] && \text{by Proposition 2} \\
&\leq 4\epsilon_{\mathrm{F}} + \sqrt{n^{-1}\mathbb{E}_S\left[\mathbb{V}_{z\sim\mathcal{D}}(\nabla F_z(A(S)))\right]} && \text{by Proposition 3} \\
&= \mathcal{O}(T^{\frac{1}{2}} n^{-\frac{1}{2}}) && \text{by (B.25)}
\end{aligned}$$

The proof is complete. $\qquad\square$

### B.3 Proof of Lemma 1 – Boundedness of $x_t$ in strongly convex case

**Technical challenges.** In this work, we focus on analyzing stochastic MGDA-based MOL algorithms in the unconstrained setting. This is because in the constrained setting, MGDA with projected gradient descent on $x$ has no guarantee to find the CA direction, and a new algorithm needs to be developed to achieve this [42]. However, a fundamental challenge in the unconstrained strongly convex setting is that a strongly convex function is not Lipschitz continuous on $\mathbb{R}^d$. We overcome this challenge by showing that $\{x_t\}_{t=1}^T$ generated by the MoDo algorithm is bounded on the trajectory, so is the gradient $\|\nabla f_{z,m}(x_t)\|$ for all $m \in [M]$, and $z \in \mathcal{Z}$. Thereby, we can derive the upper bound of PS optimization and generalization errors without the Lipschitz continuity assumption for strongly convex objectives.

**Organization of proof.** Without loss of generality, we assume $\inf_{x\in\mathbb{R}^d} f_{m,z}(x) < \infty$ for all $m \in [M]$ and $z \in \mathcal{Z}$ in the strongly convex case. In Lemma 4, we show that the optimal solution of $F_z(x)\lambda$ given any stochastic sample $z \in \mathcal{Z}$, and weighting parameter $\lambda \in \Delta^M$, is bounded. In Lemma 5, we show that if the argument parameter is bounded, then the updated parameter by MoDo at each iteration is also bounded by exploiting the co-coerciveness of strongly convex and smooth objectives. Finally, based on Lemma 4 and Lemma 5, we first prove in Corollary 4 that with a bounded initialization $x_0$, the model parameter $\{x_t\}_{t=1}^T$ generated by MoDo algorithm is bounded on the trajectory. Then based on Lemma 6, by the Lipschitz smoothness assumption of $f_{z,m}(x)$, we immediately have that $\|\nabla f_{z,m}(x)\|$ is bounded for $x \in \{x_t\}_{t=1}^T$ generated by MoDo algorithm, which completes the proof of Lemma 1. Lemma 1 paves the way for deriving the MOL uniform stability and PS generalization error of MoDo in the strongly convex and unconstrained setting.

**Lemma 4.** *Suppose Assumptions 1, 2 hold. WLOG, assume $\inf_{x\in\mathbb{R}^d} f_{m,z}(x) < \infty$ for all $m \in [M]$ and $z \in \mathcal{Z}$. For any given $\lambda \in \Delta^M$, and stochastic sample $z \in \mathcal{Z}$, define $x^*_{\lambda,z} = \arg\min_{x\in\mathbb{R}^d} F_z(x)\lambda$, then $\inf_{x\in\mathbb{R}^d} F_z(x)\lambda < \infty$ and $\|x^*_{\lambda,z}\| < \infty$, i.e., there exist finite positive constants $c_{F^*}$ and $c_{x^*}$ such that*

$$\inf_{x\in\mathbb{R}^d} F_z(x)\lambda \leq c_{F^*} \quad \text{and} \quad \|x^*_{\lambda,z}\| \leq c_{x^*}. \tag{B.26}$$

*Proof.* Under Assumption 2, for all $m \in [M]$, $f_{m,z}(x)$ is strongly convex w.r.t. $x$, thus has a unique minimizer. Define the minimizer $x^*_{m,z} = \arg\min_{x\in\mathbb{R}^d} f_{m,z}(x)$. Since a strongly convex function is coercive, $\inf_{x\in\mathbb{R}^d} f_{m,z}(x) < \infty$, i.e., $f_{m,z}(x^*_{m,z}) < \infty$, implies that $\|x^*_{m,z}\| < \infty$.

By Assumption 1, the $\ell_{f,1}$-Lipschitz smoothness of $f_{m,z}(x)$, for $x$ such that $\|x\| < \infty$

$$
\begin{aligned}
f_{m,z}(x) \leq &f_{m,z}(x^*_{m,z}) + \langle \nabla f_{m,z}(x^*_{m,z}), x - x^*_{m,z} \rangle + \frac{\ell_{f,1}}{2}\|x - x^*_{m,z}\|^2 \\
\leq &f_{m,z}(x^*_{m,z}) + \frac{\ell_{f,1}}{2}\|x - x^*_{m,z}\|^2 < \infty.
\end{aligned} \tag{B.27}
$$

Since $F_z(x)\lambda$ is convex w.r.t. $x$, for all $\lambda \in \Delta^M$, with $\lambda = [\lambda_1, \ldots, \lambda_M]^\top$, we have

$$F_z\Big(\frac{1}{M}\sum_{m=1}^M x^*_{m,z}\Big)\lambda \leq \frac{1}{M}\sum_{m=1}^M F_z(x^*_{m,z})\lambda = \frac{1}{M}\sum_{m=1}^M\sum_{m'=1}^M f_{m',z}(x^*_{m,z})\lambda_{m'} < \infty. \tag{B.28}$$

Therefore, for all $\lambda \in \Delta^M$, we have

$$\inf_{x\in\mathbb{R}^d} F_z(x)\lambda \leq F_z\Big(\frac{1}{M}\sum_{m=1}^M x^*_{m,z}\Big)\lambda < \infty. \tag{B.29}$$

Since $F_z(x)\lambda$ is strongly convex, thus is coercive, we have $\|x^*_{\lambda,z}\| < \infty$, which proves the result. $\quad\square$

**Lemma 5.** *Suppose Assumptions 1, 2 hold, and define $\kappa = 3\ell_{f,1}/\mu \geq 3$. For any given $\lambda \in \Delta^M$, and a stochastic sample $z \in \mathcal{Z}$, define $x^*_{\lambda,z} = \arg\min_x F_z(x)\lambda$. Then by Lemma 4, there exists a positive finite constant $c_{x,1} \geq c_{x^*}$ such that $\|x^*_{\lambda,z}\| \leq c_{x^*} \leq c_{x,1}$. Recall the multi-objective gradient update is*

$$G_{\lambda,z}(x) = x - \alpha \nabla F_z(x)\lambda \tag{B.30}$$

*with step size $0 \leq \alpha \leq \ell_{f,1}^{-1}$. Defining $c_{x,2} = (1 + \sqrt{2\kappa})c_{x,1}$, we have that*

$$\text{if} \quad \|x\| \leq c_{x,2}, \quad \text{then} \quad \|G_{\lambda,z}(x)\| \leq c_{x,2}. \tag{B.31}$$

*Proof.* We divide the proof into two cases: 1) when $\|x\| < c_{x,1}$; and, 2) when $c_{x,1} \leq \|x\| \leq c_{x,2}$.

1) For the first case, $\|x\| < c_{x,1} \leq c_{x,2}$, then we have

$$
\begin{aligned}
\|G_{\lambda,z}(x)\| \leq &\|G_{\lambda,z}(x) - x^*\| + \|x^*\| \\
\overset{(a)}{=} &\|G_{\lambda,z}(x) - G_{\lambda,z}(x^*)\| + \|x^*\| \overset{(b)}{\leq} \|x - x^*\| + \|x^*\| \\
\leq &\|x\| + 2\|x^*\| \leq 3c_{x,1} \leq (1 + \sqrt{6})c_{x,1} \leq (1 + \sqrt{2\kappa})c_{x,1} \leq c_{x,2}
\end{aligned} \tag{B.32}
$$

where $(a)$ follows from $\nabla F_z(x^*)\lambda = 0$, and $(b)$ follows from the non-expansiveness of the gradient update for strongly convex and smooth function.

2) For the second case, $c_{x,1} \leq \|x\| \leq c_{x,2}$, we first consider $\alpha = \ell_{f,1}^{-1}$. Let $\mu' = \mu/3$. Note that since $F_z(x)\lambda$ is $\mu$-strongly convex, it is also $\mu'$-strongly convex. By strong convexity and smoothness of $F_z(x)\lambda$, the gradients are co-coercive [36, Theorem 2.1.12], i.e., for any $x$ we have

$$(\nabla F_z(x)\lambda)^\top(x - x^*) \geq \frac{\ell_{f,1}^{-1}\|\nabla F_z(x)\lambda\|^2}{1 + \kappa^{-1}} + \frac{\mu'\|x - x^*\|^2}{1 + \kappa^{-1}}. \tag{B.33}$$

Rearranging and applying Cauchy-Schwartz inequality, we have

$$(\nabla F_z(x)\lambda)^\top x \geq (\nabla F_z(x)\lambda)^\top x^* + \frac{\ell_{f,1}^{-1}\|\nabla F_z(x)\lambda\|^2}{1+\kappa^{-1}} + \frac{\mu'\|x-x^*\|^2}{1+\kappa^{-1}}$$

$$\geq -c_{x,1}\|\nabla F_z(x)\lambda\| + \frac{\ell_{f,1}^{-1}\|\nabla F_z(x)\lambda\|^2}{1+\kappa^{-1}} + \frac{\mu'\|x-x^*\|^2}{1+\kappa^{-1}}. \qquad (B.34)$$

By the definition of $G_{\lambda,z}(x)$,

$$\|G_{\lambda,z}(x)\|^2 = \left\|x - \frac{1}{\ell_{f,1}}\nabla F_z(x)\lambda\right\|^2 = \|x\|^2 + \frac{1}{\ell_{f,1}^2}\|\nabla F_z(x)\lambda\|^2 - \frac{2}{\ell_{f,1}}(\nabla F_z(x)\lambda)^\top x. \quad (B.35)$$

Substituting (B.34) into (B.35) yields

$$\|G_{\lambda,z}(x)\|^2 \leq \|x\|^2 + \frac{1}{\ell_{f,1}^2}\|\nabla F_z(x)\lambda\|^2 + \frac{2}{\ell_{f,1}}\Big(c_{x,1}\|\nabla F_z(x)\lambda\| - \frac{\ell_{f,1}^{-1}\|\nabla F_z(x)\lambda\|^2}{1+\kappa^{-1}} - \frac{\mu'\|x-x^*\|^2}{1+\kappa^{-1}}\Big)$$

$$= \|x\|^2 + \frac{2}{\ell_{f,1}}\Big(c_{x,1}\|\nabla F_z(x)\lambda\| - \frac{1}{2\ell_{f,1}}(\frac{1-\kappa^{-1}}{1+\kappa^{-1}})\|\nabla F_z(x)\lambda\|^2 - \frac{\mu'}{1+\kappa^{-1}}\|x-x^*\|^2\Big)$$

$$\leq \|x\|^2 + \frac{2}{\ell_{f,1}}\sup_{\tau\in\mathbb{R}}\Big(\underbrace{c_{x,1}\cdot\tau - \frac{1}{2\ell_{f,1}}(\frac{1-\kappa^{-1}}{1+\kappa^{-1}})\tau^2}_{I_1} - \frac{\mu'\|x-x^*\|^2}{1+\kappa^{-1}}\Big). \qquad (B.36)$$

Since $\kappa \geq 3$, thus $\frac{1-\kappa^{-1}}{1+\kappa^{-1}} > 0$, then $I_1$ is a quadratic function w.r.t. $\tau$, and is strictly concave, thus can be bounded above by

$$\sup_{\tau\in\mathbb{R}} c_{x,1}\cdot\tau - \frac{1}{2\ell_{f,1}}(\frac{1-\kappa^{-1}}{1+\kappa^{-1}})\tau^2 \leq \frac{c_{x,1}^2\ell_{f,1}}{2}\frac{1+\kappa^{-1}}{1-\kappa^{-1}}. \qquad (B.37)$$

Substituting this back into (B.36) gives that

$$\|G_{\lambda,z}(x)\|^2 \leq \|x\|^2 + \frac{2}{\ell_{f,1}}\Big(\frac{c_{x,1}^2\ell_{f,1}}{2}\frac{1+\kappa^{-1}}{1-\kappa^{-1}} - \frac{\mu'}{1+\kappa^{-1}}\|x-x^*\|^2\Big)$$

$$= \|x\|^2 + c_{x,1}^2\frac{1+\kappa^{-1}}{1-\kappa^{-1}} - 2\frac{\kappa^{-1}}{1+\kappa^{-1}}\|x-x^*\|^2$$

$$\leq \|x\|^2 + c_{x,1}^2\frac{1+\kappa^{-1}}{1-\kappa^{-1}} - 2\frac{\kappa^{-1}}{1+\kappa^{-1}}(\|x\|-\|x^*\|)^2$$

$$\leq \underbrace{\|x\|^2 + 2c_{x,1}^2 - \kappa^{-1}(\|x\|-c_{x,1})^2}_{I_2} \qquad (B.38)$$

where the last inequality follows from $\kappa \geq 3$, thus $\frac{1+\kappa^{-1}}{1-\kappa^{-1}} \leq 2$, $-2\frac{\kappa^{-1}}{1+\kappa^{-1}} \leq -\kappa^{-1}$, and $\|x^*\| \leq c_{x,1} \leq \|x\|$ by assumption.

For $c_{x,1} \leq \|x\| \leq c_{x,2}$, $I_2$ is a strictly convex quadratic function of $\|x\|$, which achieves its maximum at $\|x\| = c_{x,1}$ or $\|x\| = c_{x,2}$. Therefore,

$$\|G_{\lambda,z}(x)\|^2 \leq \max\{3c_{x,1}^2, c_{x,2}^2 + 2c_{x,1}^2 - \kappa^{-1}(c_{x,2}-c_{x,1})^2\}$$

$$\overset{(c)}{=} \max\{3c_{x,1}^2, c_{x,2}^2\} \overset{(d)}{<} c_{x,2}^2 \qquad (B.39)$$

where $(c)$ follows from the definition that $c_{x,2} = (1+\sqrt{2\kappa})c_{x,1}$; $(d)$ follows from $\kappa \geq 3$, and thus $3c_{x,1}^2 < (1+\sqrt{2\kappa})^2 c_{x,1}^2 = c_{x,2}^2$.

We have proved the case for $\alpha = \ell_{f,1}^{-1}$. The result for $0 \leq \alpha < \ell_{f,1}^{-1}$ follows by observing that,

$$\|G_{\lambda,z}(x)\| = \|x - \alpha\nabla F_z(x)\lambda\|$$

$$= \|(1-\alpha\ell_{f,1})x + \alpha\ell_{f,1}(x - \ell_{f,1}^{-1}\nabla F_z(x)\lambda)\|$$

$$\leq (1-\alpha\ell_{f,1})\|x\| + \alpha\ell_{f,1}\|x - \ell_{f,1}^{-1}\nabla F_z(x)\lambda\| \leq c_{x,2}. \qquad (B.40)$$

The proof is complete. $\qquad\qquad\qquad\qquad\qquad\qquad\qquad\qquad\qquad\qquad\qquad\qquad\qquad\qquad\square$

**Lemma 6.** *Suppose Assumptions 1, 2 hold. For all $\lambda \in \Delta^M$ and $z \in S$, define $x^*_{\lambda,z} = \arg\min_x F_z(x)\lambda$, then there exist finite positive constants $c_{F^*}$ and $c_{x^*}$ such that $F_z(x^*_{\lambda,z})\lambda \leq c_{F^*}$ and $\|x^*_{\lambda,z}\| \leq c_{x^*}$. And for $x \in \mathbb{R}^d$ such that $\|x\|$ is bounded, i.e., there exists a finite positive constant $c_x$ such that $\|x\| \leq c_x$, then*

$$\|\nabla F_z(x)\lambda\| \leq \ell_{f,1}(c_x + c_{x^*}), \qquad and \qquad F_z(x)\lambda \leq \frac{\ell_{f,1}}{2}(c_x + c_{x^*})^2 + c_{F^*}. \tag{B.41}$$

*Proof.* Under Assumptions 1, 2, by Lemma 4, there exist finite positive constants $c_{F^*}$ and $c_{x^*}$ such that $F_z(x^*_{\lambda,z})\lambda \leq c_{F^*}$ and $\|x^*_{\lambda,z}\| \leq c_{x^*}$.

By Assumption 1, the $\ell_{f,1}$-Lipschitz continuity of the gradient $\nabla F_z(x)\lambda$, we have

$$\begin{aligned}
\|\nabla F_z(x)\lambda\| =& \|\nabla F_z(x)\lambda - \nabla F_z(x^*_{\lambda,z})\lambda\| \\
\leq& \ell_{f,1}\|x - x^*_{\lambda,z}\| \leq \ell_{f,1}(\|x\| + \|x^*_{\lambda,z}\|) \leq \ell_{f,1}(c_x + c_{x^*})
\end{aligned} \tag{B.42}$$

where the first equality uses the fact that $\nabla F_z(x^*_{\lambda,z})\lambda = 0$.

For the function value, by Assumption 1, the $\ell_{f,1}$-Lipschitz smoothness of $F_z(x)\lambda$, we have

$$\begin{aligned}
F_z(x)\lambda \leq& F_z(x^*_{\lambda,z})\lambda + \langle \nabla F_z(x^*_{\lambda,z})\lambda, x - x^*_{\lambda,z}\rangle + \frac{\ell_{f,1}}{2}\|x - x^*_{\lambda,z}\|^2 \\
\leq& F_z(x^*_{\lambda,z})\lambda + \frac{\ell_{f,1}}{2}\|x - x^*_{\lambda,z}\|^2 \\
\leq& c_{F^*} + \frac{\ell_{f,1}}{2}(c_x + c_{x^*})^2
\end{aligned} \tag{B.43}$$

from which the proof is complete. $\qquad\square$

**Corollary 4** ($x_t$ bounded on the MoDo trajectory). *Suppose Assumptions 1, 2 hold. Define $\kappa = 3\ell_{f,1}/\mu$ and $x^*_{\lambda,z} = \arg\min_x F_z(x)\lambda$ with $\lambda \in \Delta^M$. Then there exists a finite positive constant $c_{x^*}$ such that $\|x^*_{\lambda,z}\| \leq c_{x^*}$. Choose the initial iterate to be bounded, i.e., there exists a finite positive constant $c_{x_0}$ such that $\|x_0\| \leq c_{x_0}$, then for $\{x_t\}$ generated by MoDo algorithm with $\alpha_t = \alpha$ and $0 \leq \alpha \leq \ell_{f,1}^{-1}$, we have*

$$\|x_t\| \leq c_x, \quad with \quad c_x = \max\{(1 + \sqrt{2\kappa})c_{x^*}, c_{x_0}\}. \tag{B.44}$$

*Proof of Corollary 4.* Under Assumptions 1, 2, by Lemma 4, $\|x^*_{\lambda,z}\| < \infty$, i.e., there exists a finite positive constant $c_{x^*}$ such that $\|x^*_{\lambda,z}\| \leq c_{x^*}$. Let $c_{x,1} = \max\{(1 + \sqrt{2\kappa})^{-1}c_{x_0}, c_{x^*}\}$, and $c_{x,2} = (1 + \sqrt{2\kappa})c_{x,1} = \max\{c_{x_0}, (1 + \sqrt{2\kappa})c_{x^*}\}$ in Lemma 5. We then consider the following two cases:

1) If $(1 + \sqrt{2\kappa})c_{x^*} \leq c_{x_0}$, then $\|x^*_{\lambda,z}\| \leq c_{x^*} \leq (1 + \sqrt{2\kappa})^{-1}c_{x_0}$. Then it satisfies the condition in Lemma 5 that $\|x^*_{\lambda,z}\| \leq c_{x,1}$ and $\|x_0\| \leq c_{x,2}$. Applying Lemma 5 yields $\|x_1\| \leq c_{x,2}$.

2) If $(1 + \sqrt{2\kappa})c_{x^*} > c_{x_0}$, then $\|x_0\| \leq c_{x_0} < (1 + \sqrt{2\kappa})c_{x^*}$. Then it satisfies the condition in Lemma 5 that $\|x^*_{\lambda,z}\| \leq c_{x,1}$ and $\|x_0\| \leq c_{x,2}$. Applying Lemma 5 yields $\|x_1\| \leq c_{x,2}$.

Therefore, (B.44) holds for $t = 1$. We then prove by induction that (B.44) also holds for $t \in [T]$. Assume (B.44) holds at $1 \leq k \leq T - 1$, i.e.,

$$\|x_k\| \leq c_x = c_{x,2} \tag{B.45}$$

Then by Lemma 5, at $k + 1$,

$$\|x_{k+1}\| = \|G_{\lambda_{k+1}, z_{k,3}}(x_k)\| \leq c_{x,2}. \tag{B.46}$$

Since $\|x_1\| \leq c_{x,2}$, for $t = 0, \ldots, T - 1$, we have

$$\|x_{t+1}\| = \|G_{\lambda_{t+1}, z_{t,3}}(x_t)\| \leq c_{x,2}. \tag{B.47}$$

Therefore, by mathematical induction, $\|x_t\| \leq c_{x,2} = c_x$, for all $t \in [T]$. The proof is complete. $\quad\square$

**Proof of Lemma 1.** By Corollary 4, for $\{x_t\}$ generated by MoDo algorithm with $\alpha_t = \alpha$ and $0 \le \alpha \le \ell_{f,1}^{-1}$, we have

$$\|x_t\| \le c_x, \quad \text{with} \quad c_x = \max\{(1 + \sqrt{2\kappa})c_{x^*}, c_{x_0}\}. \tag{B.48}$$

According to Lemma 6, define $\ell_f = \ell_{f,1}(c_x + c_{x^*})$, and $\ell_F = \sqrt{M}\ell_f$, then it holds for all $\lambda \in \Delta^M$

$$\|\nabla F(x_t)\lambda\| \le \ell_f \quad \text{and} \quad \|\nabla F(x_t)\| \le \|\nabla F(x_t)\|_{\mathrm{F}} \le \ell_F. \tag{B.49}$$

$\square$

## B.4 Proof of Theorem 2 – PS generalization error in strongly convex case

**Technical challenges.** One challenge that the strongly convex objectives are not Lipschitz continuous for $x \in \mathbb{R}^d$ is addressed by Lemma 1. Another challenge compared to static weighting or single-objective learning is that the MoDo algorithm involves the update of two coupled sequences $\{x_t\}$ and $\{\lambda_t\}$. Consequently, the traditional standard argument that the SGD update for strongly convex objectives has the contraction property [14] does not necessarily hold in our case since the weighting parameter $\lambda$ is changing, as detailed in Section B.4.1. Nevertheless, we manage to derive a tight stability bound when $\gamma = \mathcal{O}(T^{-1})$, as detailed in Section B.4.3.

**Organization of proof.** In Section B.4.1, we prove the properties of the MoDo update, including expansiveness or non-expansiveness and boundedness. Building upon these properties, in Section B.4.3, we prove the upper bound of argument stability in Theorem 5, and the upper bound of MOL uniform stability. To show the tightness of the upper bound, in Section B.4.4, Theorem 6, we derive a matching lower bound of MOL uniform stability. Combining the upper bound in Section B.4.3 and the lower bound in Section B.4.4 leads to the results in Theorem 2, whose proof is in Section B.4.5.

### B.4.1 Expansiveness and boundedness of MoDo update

In this section, we prove the properties of the update function of MoDo at each iteration, including boundedness and approximate expansiveness, which is then used to derive the algorithm stability. For $z, z_1, z_2 \in S$, $\lambda \in \Delta^M$, recall that the update functions of MoDo is

$$G_{x,z_1,z_2}(\lambda) = \Pi_{\Delta^M}\left(\lambda - \gamma \nabla F_{z_1}(x)^\top \nabla F_{z_2}(x)\lambda\right)$$
$$G_{\lambda,z}(x) = x - \alpha \nabla F_z(x)\lambda.$$

**Lemma 7** (Boundedness of update function of MoDo). *Let $\ell_f$ be a positive constant. If $\|\nabla F_z(x)\lambda\| \le \ell_f$ for all $\lambda \in \Delta^M$, $z \in S$ and $x \in \{x_t\}_{t=1}^T$ generated by the MoDo algorithm with step size $\alpha_t \le \alpha$, then $G_{\lambda,z}(x)$ is $(\alpha\ell_f)$-bounded on the trajectory of MoDo, i.e.,*

$$\sup_{x \in \{x_t\}_{t=1}^T} \|G_{\lambda,z}(x) - x\| \le \alpha\ell_f. \tag{B.50}$$

*Proof.* For all $x \in \{x_t\}_{t=1}^T$, $\lambda \in \Delta^M$, and $z \in S$, since $\|\nabla F_z(x)\lambda\| \le \ell_f$, we have

$$\|G_{\lambda,z}(x) - x\| \le \|\alpha \nabla F_z(x)\lambda\| \le \alpha\ell_f \tag{B.51}$$

which proves the boundedness. $\square$

**Lemma 8** (Properties of update function of MoDo in convex case). *Suppose Assumptions 1, 2 hold. Let $\ell_f$ be a positive constant. If for all $\lambda, \lambda' \in \Delta^M$, $z \in S$, and $x \in \{x_t\}_{t=1}^T$, $x' \in \{x_t'\}_{t=1}^T$ generated by the MoDo algorithm on datasets $S$ and $S'$, respectively, we have $\|\nabla F_z(x)\lambda\| \le \ell_f$, $\|\nabla F_z(x')\lambda'\| \le \ell_f$, and $\|\nabla F_z(x)\| \le \ell_F$, $\|\nabla F_z(x')\| \le \ell_F$, and step sizes of MoDo satisfy $\alpha_t \le \alpha$, $\gamma_t \le \gamma$, it holds that*

$$\|G_{\lambda,z}(x) - G_{\lambda',z}(x')\|^2 \le (1 - 2\alpha\mu + 2\alpha^2\ell_{f,1}^2)\|x - x'\|^2$$
$$+ 2\alpha\ell_F\|x - x'\|\|\lambda - \lambda'\| + 2\alpha^2\ell_F^2\|\lambda - \lambda'\|^2 \tag{B.52}$$

$$\|G_{x,z_1,z_2}(\lambda) - G_{x',z_1,z_2}(\lambda')\|^2 \le \left((1 + \ell_F^2\gamma)^2 + (1 + \ell_F^2\gamma)\ell_{g,1}\gamma\right)\|\lambda - \lambda'\|^2$$
$$+ \left((1 + \ell_F^2\gamma)\ell_{g,1}\gamma + \ell_{g,1}^2\gamma^2\right)\|x - x'\|^2. \tag{B.53}$$

*Proof.* The squared norm of the difference of $G_{\lambda,z}(x)$ and $G_{\lambda',z}(x')$ can be bounded by

$$\|G_{\lambda,z}(x) - G_{\lambda',z}(x')\|^2$$
$$= \|x - x'\|^2 - 2\alpha\langle x - x', \nabla F_z(x)\lambda - \nabla F_z(x')\lambda'\rangle + \alpha^2\|\nabla F_z(x)\lambda - \nabla F_z(x')\lambda'\|^2$$
$$\overset{(a)}{\leq} \|x - x'\|^2 - 2\alpha\langle x - x', (\nabla F_z(x) - \nabla F_z(x'))\lambda\rangle + 2\alpha^2\|(\nabla F_z(x) - \nabla F_z(x'))\lambda\|^2$$
$$\quad + 2\alpha\langle x - x', \nabla F_z(x')(\lambda' - \lambda)\rangle + 2\alpha^2\|\nabla F_z(x')(\lambda - \lambda')\|^2$$
$$\overset{(b)}{\leq} (1 - 2\alpha\mu + 2\alpha^2\ell_{f,1}^2)\|x - x'\|^2 + 2\alpha\langle x - x', \nabla F_z(x')(\lambda' - \lambda)\rangle + 2\alpha^2\ell_F^2\|\lambda' - \lambda\|^2$$
$$\overset{(c)}{\leq} (1 - 2\alpha\mu + 2\alpha^2\ell_{f,1}^2)\|x - x'\|^2 + 2\alpha\ell_F\|x - x'\|\|\lambda' - \lambda\| + 2\alpha^2\ell_F^2\|\lambda' - \lambda\|^2 \qquad \text{(B.54)}$$

where $(a)$ follows from rearranging and that $\|a + b\|^2 \leq 2\|a\|^2 + 2\|b\|^2$; $(b)$ follows from the $\mu$-strong convexity of $F_z(x)\lambda$, $\ell_{f,1}$-Lipschitz continuity of $\nabla F_z(x)\lambda$, and that $\|\nabla F_z(x')\| \leq \ell_F$ for $x' \in \{x'_t\}_{t=1}^T$; and, $(c)$ follows from Cauchy-Schwartz inequality.

And $\|G_{x,z_1,z_2}(\lambda) - G_{x',z_1,z_2}(\lambda')\|$ can be bounded by

$$\|G_{x,z_1,z_2}(\lambda) - G_{x',z_1,z_2}(\lambda')\|$$
$$= \|\Pi_{\Delta^M}(\lambda - \gamma(\nabla F_{z_1}(x)^\top \nabla F_{z_2}(x))\lambda) - \Pi_{\Delta^M}(\lambda' - \gamma(\nabla F_{z_1}(x')^\top \nabla F_{z_2}(x'))\lambda')\|$$
$$\overset{(d)}{\leq} \|\lambda - \lambda' - \gamma(\nabla F_{z_1}(x)^\top \nabla F_{z_2}(x)\lambda - \nabla F_{z_1}(x')^\top \nabla F_{z_2}(x')\lambda')\|$$
$$\overset{(e)}{\leq} \|\lambda - \lambda'\| + \gamma\|\nabla F_{z_1}(x)^\top \nabla F_{z_2}(x)(\lambda - \lambda')\| + \gamma\|(\nabla F_{z_1}(x)^\top \nabla F_{z_2}(x) - \nabla F_{z_1}(x')^\top \nabla F_{z_2}(x'))\lambda'\|$$
$$\overset{(f)}{\leq} \|\lambda - \lambda'\| + \gamma\ell_F^2\|\lambda - \lambda'\| + \gamma\|(\nabla F_{z_1}(x)^\top \nabla F_{z_2}(x) - \nabla F_{z_1}(x')^\top \nabla F_{z_2}(x'))\lambda'\|$$
$$\overset{(g)}{\leq} \|\lambda - \lambda'\| + \gamma\ell_F^2\|\lambda - \lambda'\| + \gamma(\|(\nabla F_{z_1}(x) - \nabla F_{z_1}(x'))^\top \nabla F_{z_2}(x)\lambda'\|$$
$$\quad + \|\nabla F_{z_1}(x')^\top(\nabla F_{z_2}(x) - \nabla F_{z_2}(x'))\lambda'\|)$$
$$\overset{(h)}{\leq} (1 + \ell_F^2\gamma)\|\lambda - \lambda'\| + (\ell_f\ell_{F,1} + \ell_F\ell_{f,1})\gamma\|x - x'\| \qquad \text{(B.55)}$$

where $(d)$ follows from non-expansiveness of projection; $(e)$ follows from triangle inequality, $(f)$ follows from $\|\nabla F_z(x)\| \leq \ell_F$ for $x \in \{x'_t\}_{t=1}^T$, $(g)$ follows from triangle inequality; and $(h)$ follows from $\ell_{f,1}$-Lipschitz continuity of $\nabla F_z(x)\lambda'$, $\ell_{F,1}$-Lipschitz continuity of $\nabla F_z(x)$, $\|\nabla F_z(x)\| \leq \ell_F$ for $x \in \{x'_t\}_{t=1}^T$, and $\|\nabla F_z(x)\lambda'\| \leq \ell_f$ for $x \in \{x_t\}_{t=1}^T$.

Let $\ell_{g,1} = \ell_f\ell_{F,1} + \ell_F\ell_{f,1}$. Taking square on both sides of (B.55) yields

$$\|G_{x,z_1,z_2}(\lambda) - G_{x',z_1,z_2}(\lambda')\|^2$$
$$\leq \left((1 + \ell_F^2\gamma)\|\lambda - \lambda'\| + \ell_{g,1}\gamma\|x - x'\|\right)^2$$
$$= (1 + \ell_F^2\gamma)^2\|\lambda - \lambda'\|^2 + 2(1 + \ell_F^2\gamma)\ell_{g,1}\gamma\|\lambda - \lambda'\|\|x - x'\| + \ell_{g,1}^2\gamma^2\|x - x'\|^2$$
$$\leq (1 + \ell_F^2\gamma)^2\|\lambda - \lambda'\|^2 + (1 + \ell_F^2\gamma)\ell_{g,1}\gamma(\|\lambda - \lambda'\|^2 + \|x - x'\|^2) + \ell_{g,1}^2\gamma^2\|x - x'\|^2$$
$$= \left((1 + \ell_F^2\gamma)^2 + (1 + \ell_F^2\gamma)\ell_{g,1}\gamma\right)\|\lambda - \lambda'\|^2 + \left((1 + \ell_F^2\gamma)\ell_{g,1}\gamma + \ell_{g,1}^2\gamma^2\right)\|x - x'\|^2. \qquad \text{(B.56)}$$

The proof is complete. $\qquad\square$

### B.4.2 Growth recursion

**Lemma 9** (Growth recursion with approximate expansiveness). *Fix an arbitrary sequence of updates $G_1, \ldots, G_T$ and another sequence $G'_1, \ldots, G'_T$. Let $x_0 = x'_0$ be a starting point in $\Omega$ and define $\delta_t = \|x'_t - x_t\|$ where $x_t, x'_t$ are defined recursively through*

$$x_{t+1} = G_t(x_t), \quad x'_{t+1} = G'_t(x'_t) \quad (t > 0).$$

Let $\eta_t > 0, \nu_t \geq 0$, and $\varsigma_t \geq 0$. Then, for any $p > 0$, and $t \in [T]$, we have the recurrence relation (with $\delta_0 = 0$)

$$\delta_{t+1}^2 \leq \begin{cases} \eta_t \delta_t^2 + \nu_t, & G_t = G_t' \text{ is } (\eta_t, \nu_t)\text{-approximately expansive in square;} \\ (1+p)\min\{\eta_t \delta_t^2 + \nu_t, \delta_t^2\} + (1+\frac{1}{p})4\varsigma_t^2 & G_t \text{ and } G_t' \text{ are } \varsigma_t\text{-bounded,} \\ & G_t \text{ is } (\eta_t, \nu_t)\text{-approximately expansive in square.} \end{cases}$$

*Proof.* When $G_t$ and $G_t'$ are $\varsigma_t$-bounded, we can bound $\delta_{t+1}$ by

$$\begin{aligned} \delta_{t+1} = \|x_{t+1} - x_{t+1}'\| =& \|G_t(x_t) - G_t'(x_t')\| \\ =& \|G_t(x_t) - x_t - G_t'(x_t') + x_t' + x_t - x_t'\| \\ \leq& \|G_t(x_t) - x_t\| + \|G_t'(x_t') - x_t'\| + \|x_t - x_t'\| \\ \leq& 2\varsigma_t + \delta_t. \end{aligned} \tag{B.57}$$

Alternatively, when $G_t$ and $G_t'$ are $\varsigma_t$-bounded, $G_t$ is $(\eta_t, \nu_t)$-approximately expansive, we have

$$\begin{aligned} \delta_{t+1} = \|x_{t+1} - x_{t+1}'\| =& \|G_t(x_t) - G_t'(x_t')\| \\ =& \|G_t(x_t) - G_t(x_t') + G_t(x_t') - G_t'(x_t')\| \\ \leq& \|G_t(x_t) - G_t(x_t')\| + \|G_t(x_t') - G_t'(x_t')\| \\ \leq& \eta_t \delta_t + \nu_t + \|G_t(x_t') - x_t' - G_t'(x_t') + x_t'\| \\ \leq& \eta_t \delta_t + \nu_t + \|G_t(x_t') - x_t'\| + \|G_t'(x_t') - x_t'\| \\ \leq& \eta_t \delta_t + \nu_t + 2\varsigma_t. \end{aligned} \tag{B.58}$$

When $G_t = G_t'$, is $(\eta_t, \nu_t)$-approximately expansive in square, given $\delta_t^2$, $\delta_{t+1}^2$ can be bounded by

$$\delta_{t+1}^2 = \|x_{t+1} - x_{t+1}'\|^2 = \|G_t(x_t) - G_t(x_t')\|^2 \leq \eta_t \|x_t - x_t'\|^2 + \nu_t = \eta_t \delta_t^2 + \nu_t. \tag{B.59}$$

When $G_t$ and $G_t'$ are $\varsigma_t$-bounded, applying (B.57), we can bound $\delta_{t+1}^2$ by

$$\delta_{t+1}^2 \leq (\delta_t + 2\varsigma_t)^2 \leq (1+p)\delta_t^2 + (1+1/p)4\varsigma_t^2 \tag{B.60}$$

where $p > 0$ and the last inequality follows from $(a+b)^2 \leq (1+p)a^2 + (1+1/p)b^2$.

Alternatively, when $G_t$ and $G_t'$ are $\varsigma_t$-bounded, $G_t$ is $(\eta_t, \nu_t)$-approximately expansive in square, the following holds

$$\begin{aligned} \delta_{t+1}^2 = \|x_{t+1} - x_{t+1}'\|^2 =& \|G_t(x_t) - G_t'(x_t')\|^2 \\ =& \|G_t(x_t) - G_t(x_t') + G_t(x_t') - G_t'(x_t')\|^2 \\ \leq& (1+p)\|G_t(x_t) - G_t(x_t')\|^2 + (1+1/p)\|G_t(x_t') - G_t'(x_t')\|^2 \\ \leq& (1+p)(\eta_t \delta_t^2 + \nu_t) + (1+1/p)\|G_t(x_t') - x_t' - G_t'(x_t') + x_t'\|^2 \\ \leq& (1+p)(\eta_t \delta_t^2 + \nu_t) + 2(1+1/p)(\|G_t(x_t') - x_t'\|^2 + \|G_t'(x_t') - x_t'\|^2) \\ \leq& (1+p)(\eta_t \delta_t^2 + \nu_t) + (1+1/p)4\varsigma_t^2. \end{aligned} \tag{B.61}$$

The proof is complete. $\qquad\square$

### B.4.3   Upper bound of MOL uniform stability

In Theorem 5 we bound the argument stability, which is then used to derive the MOL uniform stability and PS generalization error in Theorem 2.

**Theorem 5** (Argument stability bound in strongly convex case). *Suppose Assumptions 1, 2, hold. Let $A$ be the MoDo algorithm in Algorithm 1. Choose the step sizes $\alpha_t \leq \alpha \leq \min\{1/(2\ell_{f,1}), \mu/(2\ell_{f,1}^2)\}$, and $\gamma_t \leq \gamma \leq \min\{\frac{\mu^2}{120\ell_f^2 \ell_{g,1}}, \frac{1}{8(3\ell_f^2 + 2\ell_{g,1})}\}/T$. Then it holds that*

$$\mathbb{E}_A[\|A(S) - A(S')\|^2] \leq \frac{48}{\mu n}\ell_f^2\Big(\alpha + \frac{12 + 4M\ell_f^2}{\mu n} + \frac{10M\ell_f^4 \gamma}{\mu}\Big). \tag{B.62}$$

**Proof of Theorem 5.** Under Assumptions 1, 2, Lemma 1 implies that for $\{x_t\}$ generated by the MoDo algorithm, and for all $\lambda \in \Delta^M$, and for all $m \in [M]$,

$$\|\nabla F_z(x_t)\lambda\| \le \ell_{f,1}(c_x + c_{x^*}) = \ell_f. \quad \text{and} \quad \|\nabla F_z(x_t)\| \le \|\nabla F_z(x_t)\|_{\mathrm{F}} \le \sqrt{M}\ell_f = \ell_F. \quad \text{(B.63)}$$

For notation simplicity, denote $\delta_t = \|x_t - x'_t\|$, $\zeta_t = \|\lambda_t - \lambda'_t\|$, $x_T = A_T(S)$ and $x'_T = A_T(S')$. Denote the index of the different sample in $S$ and $S'$ as $i^*$, and the set of indices selected at the $t$-th iteration as $I_t$, i.e., $I_t = \{i_{t,s}\}_{s=1}^3$. When $i^* \notin I_t$, for any $c_1 > 0$, based on Lemma 8, we have

$$
\begin{aligned}
\delta_{t+1}^2 &\le (1 - 2\alpha_t\mu + 2\alpha_t^2\ell_{f,1}^2)\delta_t^2 + 2\alpha_t\ell_F\delta_t\zeta_{t+1} + 2\alpha_t^2\ell_F^2\zeta_{t+1}^2 \\
&\le (1 - 2\alpha_t\mu + 2\alpha_t^2\ell_{f,1}^2)\delta_t^2 + \alpha_t\ell_F(c_1\delta_t^2 + c_1^{-1}\zeta_{t+1}^2) + 2\alpha_t^2\ell_F^2\zeta_{t+1}^2 \\
&\le (1 - \alpha_t\mu)\delta_t^2 + \alpha_t\ell_F(c_1\delta_t^2 + c_1^{-1}\zeta_{t+1}^2) + 2\alpha_t^2\ell_F^2\zeta_{t+1}^2 \quad \text{(B.64)}
\end{aligned}
$$

where the second last inequality is due to Young's inequality; the last inequality is due to choosing $\alpha_t \le \mu/(2\ell_{f,1}^2)$.

When $i^* \in I_t$, from Lemma 7, the $(\alpha_t\ell_f)$-boundedness of the update at $t$-th iteration, and Lemma 9, the growth recursion, for a given constant $p > 0$, we have

$$\delta_{t+1}^2 \le (1 + p)\delta_t^2 + (1 + 1/p)4\alpha_t^2\ell_f^2. \quad \text{(B.65)}$$

Taking expectation of $\delta_{t+1}^2$ over $I_t$, we have

$$
\begin{aligned}
\mathbb{E}_{I_t}[\delta_{t+1}^2] &\le \mathbb{P}(i^* \notin I_t)\Big((1 - \alpha_t\mu)\delta_t^2 + \alpha_t\ell_F c_1\delta_t^2 + (\alpha_t\ell_F c_1^{-1} + 2\alpha_t^2\ell_F^2)\mathbb{E}_{I_t}[\zeta_{t+1}^2 \mid i^* \notin I_t]\Big) \\
&\quad + \mathbb{P}(i^* \in I_t)\Big((1 + p)\delta_t^2 + (1 + 1/p)4\alpha_t^2\ell_f^2\Big) \\
&\le \Big(1 - \alpha_t(\mu - \ell_F c_1)\mathbb{P}(i^* \notin I_t) + p\mathbb{P}(i^* \in I_t)\Big)\delta_t^2 \\
&\quad + \alpha_t \underbrace{(\ell_F c_1^{-1} + 2\alpha\ell_F^2)}_{c_2}\mathbb{E}_{I_t}[\zeta_{t+1}^2 \mid i^* \notin I_t]\mathbb{P}(i^* \notin I_t) + \Big(1 + \frac{1}{p}\Big)\mathbb{P}(i^* \in I_t)4\alpha_t^2\ell_f^2.
\end{aligned}
$$
$$\text{(B.66)}$$

At each iteration of MoDo, we randomly select three independent samples (instead of one) from the training set $S$. Then the probability of selecting the different sample from $S$ and $S'$ at the $t$-th iteration, $\mathbb{P}(i^* \in I_t)$ in the above equation, can be computed as follows

$$\mathbb{P}(i^* \in I_t) = 1 - \Big(\frac{n-1}{n}\Big)^3 \le \frac{3}{n}. \quad \text{(B.67)}$$

Consequently, the probability of selecting the same sample from $S$ and $S'$ at the $t$-th iteration is $\mathbb{P}(i^* \notin I_t) = 1 - \mathbb{P}(i^* \in I_t)$.

Let $\ell_{g,1} = \ell_f\ell_{F,1} + \ell_F\ell_{f,1}$. Recalling when $i^* \notin I_t$, $\zeta_{t+1} \le (1 + \ell_F^2\gamma_t)\zeta_t + 2\gamma_t\ell_{g,1}\delta_t$ from Lemma 8, it follows that

$$
\begin{aligned}
\zeta_{t+1}^2 &\le \Big((1 + \ell_F^2\gamma_t)^2 + (1 + \ell_F^2\gamma_t)\ell_{g,1}\gamma_t\Big)\zeta_t^2 + \Big((1 + \ell_F^2\gamma_t)\ell_{g,1}\gamma_t + \ell_{g,1}^2\gamma_t^2\Big)\delta_t^2 \\
&\le \Big(1 + \underbrace{(3\ell_F^2 + 2\ell_{g,1})}_{c_3}\gamma_t\Big)\zeta_t^2 + 3\ell_{g,1}\gamma_t\delta_t^2 \quad \text{(B.68)}
\end{aligned}
$$

where the last inequality follows from $\ell_{g,1}\gamma_t \le 1$, and $\ell_F^2\gamma_t \le 1$.

And since $\zeta_t$ and $\delta_t$ are independent of $I_t$, it follows that

$$\mathbb{E}_{I_t}[\zeta_{t+1}^2 \mid i^* \notin I_t] \le (1 + c_3\gamma_t)\zeta_t^2 + 3\ell_{g,1}\gamma_t\delta_t^2. \quad \text{(B.69)}$$

Combining (B.66) and (B.69), we have

$$
\begin{aligned}
\mathbb{E}_{I_t}[\delta_{t+1}^2] &\le \Big(1 - \alpha_t(\mu - \ell_F c_1)\mathbb{P}(i^* \notin I_t) + p\mathbb{P}(i^* \in I_t)\Big)\delta_t^2 + \Big(1 + \frac{1}{p}\Big)\mathbb{P}(i^* \in I_t)4\alpha_t^2\ell_f^2 \\
&\quad + \alpha_t c_2\Big((1 + c_3\gamma_t)\zeta_t^2 + 3\ell_{g,1}\gamma_t\delta_t^2\Big)\mathbb{P}(i^* \notin I_t)
\end{aligned}
$$
$$\text{(B.70)}$$

$$=\Big(\eta_t + p\mathbb{P}(i^* \in I_t)\Big)\delta_t^2 + \alpha_t c_2\big(1 + c_3\gamma_t\big)\zeta_t^2\mathbb{P}(i^* \notin I_t) + \Big(1 + \frac{1}{p}\Big)\mathbb{P}(i^* \in I_t)4\alpha_t^2\ell_f^2$$

where we define $\eta_t = 1 - \alpha_t(\mu - \ell_F c_1 - 3c_2\ell_{g,1}\gamma_t)\mathbb{P}(i^* \notin I_t)$.

While when $i^* \in I_t$, for a given constant $p_2 > 0$, we have

$$\begin{aligned}
\zeta_{t+1} =&\|\Pi_{\Delta^M}(\lambda_t - \gamma_t h_{t,1}(x_t)^\top h_{t,2}(x_t)\lambda_t) - \Pi_{\Delta^M}(\lambda_t' - \gamma_t h_{t,1}'(x_t')^\top h_{t,2}'(x_t')\lambda_t')\| \\
\leq&\|\lambda_t - \lambda_t' - \gamma_t(h_{t,1}(x_t)^\top h_{t,2}(x_t)\lambda_t - h_{t,1}'(x_t')^\top h_{t,2}'(x_t')\lambda_t')\| \\
\leq&\|\lambda_t - \lambda_t'\| + 2\gamma_t\ell_F\ell_f \leq \zeta_t + 2\gamma_t\sqrt{M}\ell_f^2 \\
\zeta_{t+1}^2 \leq&(1 + p_2)\zeta_t^2 + (1 + 1/p_2)4\gamma_t^2 M\ell_f^4.
\end{aligned} \tag{B.71}$$

Taking expectation of $\zeta_{t+1}^2$ over $I_t$ gives

$$\begin{aligned}
\mathbb{E}_{I_t}[\zeta_{t+1}^2] =&\mathbb{E}_{I_t}[\zeta_{t+1}^2 \mid i^* \in I_t]\mathbb{P}(i^* \in I_t) + \mathbb{E}_{I_t}[\zeta_{t+1}^2 \mid i^* \notin I_t]\mathbb{P}(i^* \notin I_t) \\
\leq&\Big((1 + p_2)\zeta_t^2 + (1 + 1/p_2)4\gamma_t^2 M\ell_f^4\Big)\mathbb{P}(i^* \in I_t) + \Big((1 + c_3\gamma_t)\zeta_t^2 + 3\ell_{g,1}\gamma_t\delta_t^2\Big)\mathbb{P}(i^* \notin I_t) \\
\leq&\Big(1 + c_3\gamma_t + \frac{3}{n}p_2\Big)\zeta_t^2 + (1 + \frac{1}{p_2})4\gamma_t^2 M\ell_f^4\frac{3}{n} + 3\ell_{g,1}\gamma_t\delta_t^2.
\end{aligned} \tag{B.72}$$

Based on linearity of expectation and applying (B.72) recursively yields

$$\begin{aligned}
\mathbb{E}[\zeta_{t+1}^2] \leq& \sum_{t'=0}^t \Big((1 + \frac{1}{p_2})4\gamma^2 M\ell_f^4\frac{3}{n} + 3\ell_{g,1}\gamma\mathbb{E}[\delta_{t'}^2]\Big)\Bigg(\prod_{k=t'+1}^t \Big(1 + c_3\gamma + \frac{3}{n}p_2\Big)\Bigg) \\
=& \sum_{t'=0}^t \Big((1 + \frac{1}{p_2})4\gamma^2 M\ell_f^4\frac{3}{n} + 3\ell_{g,1}\gamma\mathbb{E}[\delta_{t'}^2]\Big)\Big(1 + c_3\gamma + \frac{3}{n}p_2\Big)^{t-t'} \\
\overset{(a)}{\leq}& \sum_{t'=0}^t \Big((1 + \frac{8T}{n})4\gamma^2 M\ell_f^4\frac{3}{n} + 3\ell_{g,1}\gamma\mathbb{E}[\delta_{t'}^2]\Big)\Big(1 + \frac{1}{2T}\Big)^{t-t'} \\
\overset{(b)}{\leq}& \sum_{t'=0}^t \Big((1 + \frac{8T}{n})4\gamma^2 M\ell_f^4\frac{3}{n} + 3\ell_{g,1}\gamma\mathbb{E}[\delta_{t'}^2]\Big)e^{\frac{1}{2}} \\
\overset{(c)}{\leq}& 2\gamma \sum_{t'=0}^t \Big((1 + \frac{8T}{n})4\gamma M\ell_f^4\frac{3}{n} + 3\ell_{g,1}\mathbb{E}[\delta_{t'}^2]\Big)
\end{aligned} \tag{B.73}$$

where $(a)$ follows from choosing $\gamma_t \leq \gamma \leq 1/(8c_3 T)$, $p_2 = n/(8T)$, $(b)$ follows from $t - t' \leq T$, and $(1 + \frac{a}{T})^T \leq e^a$, and the inequality $(c)$ follows from $e^{\frac{1}{2}} < 2$.

Note that $\delta_0 = 0, \zeta_1 = 0$. Applying (B.66) at $t = 0$ gives

$$\mathbb{E}[\delta_1^2] \leq \frac{3}{n}(1 + \frac{1}{p})4\alpha^2\ell_f^2$$

which together with (B.72) gives

$$\mathbb{E}[\zeta_2^2] \leq 3\ell_{g,1}\gamma_1\delta_1^2 + (1 + \frac{1}{p_2})4\gamma_1^2 M\ell_f^4\frac{3}{n}.$$

Therefore, for $0 \leq t \leq 1$, it satisfies that

$$\begin{aligned}
\mathbb{E}[\delta_t^2] \leq& \left(\frac{3}{n}(1 + \frac{1}{p})4\alpha^2\ell_f^2 + 24M\ell_f^4 c_2\left(\frac{8\gamma T}{n} + \gamma\right)\frac{\alpha}{n}\right)\underbrace{\left(\sum_{t'=0}^{t-1}(1 - \frac{1}{2}\alpha\mu + \frac{3p}{n})^{t-t'-1}\right)}_{\beta_t} \\
=& \left(\frac{3}{n}(1 + \frac{1}{p})4\alpha^2\ell_f^2 + 24M\ell_f^4 c_2\left(\frac{8\gamma T}{n} + \gamma\right)\frac{\alpha}{n}\right)\beta_t.
\end{aligned} \tag{B.74}$$

Next, we will prove by induction that (B.74) also holds for $t > 1$.

Assuming that (B.74) holds for all $0 \le t \le k \le T - 1$, we apply (B.70) to the case where $t = k$ to obtain

$$\mathbb{E}[\delta_{k+1}^2] \le \left(\eta_k + \frac{3p}{n}\right)\mathbb{E}[\delta_k^2] + \alpha_k c_2\left(1 + c_3\gamma_k\right)\mathbb{E}[\zeta_k^2]\mathbb{P}(i^* \notin I_t) + \frac{3}{n}\left(1 + \frac{1}{p}\right)4\alpha_k^2\ell_f^2$$

$$\overset{(a)}{\le} \left(\eta_k + \frac{3p}{n}\right)\mathbb{E}[\delta_k^2]$$

$$+ 2\alpha_k c_2\gamma\left(\sum_{t'=1}^{k}\left((1 + \frac{8T}{n})\frac{12\gamma M\ell_f^4}{n} + 3\ell_{g,1}\mathbb{E}[\delta_{t'}^2]\right)\right)\mathbb{P}(i^* \notin I_t) + \frac{3}{n}\left(1 + \frac{1}{p}\right)4\alpha_k^2\ell_f^2$$

$$\overset{(b)}{\le} \underbrace{\left(\left(\eta_k + \frac{3p}{n}\right)\beta_k + 1 + 6\alpha_k c_2\ell_{g,1}\gamma\left(\sum_{t'=1}^{k}\beta_{t'}\right)\mathbb{P}(i^* \notin I_t)\right)}_{J_1}$$

$$\times \left(\frac{3}{n}\left(1 + \frac{1}{p}\right)4\alpha^2\ell_f^2 + 24M\ell_f^4 c_2\left(\frac{8\gamma T}{n} + \gamma\right)\frac{\alpha}{n}\right) \tag{B.75}$$

where $(a)$ follows from (B.73), and $(b)$ follows from (B.74) for $0 \le t \le k$ and that $\gamma k \le \gamma T \le 1$. The coefficient $J_1$ in (B.75) can be further bounded by

$$J_1 = \left(\eta_k + \frac{3p}{n}\right)\beta_k + 1 + 6\alpha_k c_2\ell_{g,1}\gamma(\sum_{t'=1}^{k}c_{t'})\mathbb{P}(i^* \notin I_t)$$

$$\overset{(c)}{\le} \left(\eta_k + \frac{3p}{n}\right)\beta_k + 1 + 6\alpha_k c_2\ell_{g,1}k\gamma\beta_k\mathbb{P}(i^* \notin I_t)$$

$$\overset{(d)}{\le} \left(1 - \alpha_k(\mu - \ell_F c_1 - 3c_2\ell_{g,1}\gamma(1 + 2k))\mathbb{P}(i^* \notin I_t) + \frac{3p}{n}\right)\beta_k + 1$$

$$\overset{(e)}{\le} \left(1 - \frac{1}{2}\alpha\mu + \frac{3p}{n}\right)\beta_k + 1 \tag{B.76}$$

where $(c)$ follows from $\beta_t \le \beta_{t+1}$, $\gamma_t \le \gamma$ for all $t = 0, \ldots, T$; $(d)$ follows from the definition of $\eta_k$; $(e)$ is because $\gamma \le \mu^2/(120\ell_F^2\ell_{g,1}T)$, $\alpha \le 1/(2\ell_{f,1}) \le 1/(2\mu)$ and choosing $c_1 = \mu/(4\ell_F)$ leads to

$$\ell_F c_1 + 3c_2\ell_{g,1}\gamma(1 + 2k)\gamma \le \ell_F c_1 + 6(\ell_F c_1^{-1} + 2\alpha\ell_F^2)\ell_{g,1}(k + 1)\gamma$$

$$\le \frac{1}{4}\mu + 6(4\mu^{-1} + 2\alpha)\ell_F^2\ell_{g,1}\frac{k+1}{T}\frac{\mu^2}{120\ell_F^2\ell_{g,1}} \le \frac{1}{2}\mu.$$

Combining (B.75) and (B.76) implies

$$\mathbb{E}[\delta_{k+1}^2] \le \left(\left(1 - \frac{1}{2}\alpha\mu + \frac{3p}{n}\right)\beta_k + 1\right)\left(\frac{3}{n}(1 + \frac{1}{p})4\alpha^2\ell_f^2 + 24M\ell_f^4 c_2\left(\frac{8\gamma T}{n} + \gamma\right)\frac{\alpha}{n}\right)$$

$$= c_{k+1}\left(\frac{3}{n}(1 + \frac{1}{p})4\alpha^2\ell_f^2 + 24M\ell_f^4 c_2\left(\frac{8\gamma T}{n} + \gamma\right)\frac{\alpha}{n}\right) \tag{B.77}$$

where the equality follows by the definition of $\beta_t$ given in (B.74). The above statements from (B.75)-(B.77) show that if (B.74) holds for all $t$ such that $0 \le t \le k \le T - 1$, it also holds for $t = k + 1$.

Therefore, we can conclude that for $T \ge 0$, it follows

$$\mathbb{E}[\delta_T^2] \le \beta_T\left(\frac{3}{n}(1 + \frac{1}{p})4\alpha^2\ell_f^2 + 24M\ell_f^4 c_2\left(\frac{8\gamma T}{n} + \gamma\right)\frac{\alpha}{n}\right)$$

$$= \left(\frac{3}{n}(1 + \frac{1}{p})4\alpha^2\ell_f^2 + 24M\ell_f^4 c_2\left(\frac{8\gamma T}{n} + \gamma\right)\frac{\alpha}{n}\right)\left(\sum_{k=0}^{T-1}\left(1 - \frac{1}{2}\alpha\mu + \frac{3p}{n}\right)^{T-k-1}\right)$$

$$= \left( \frac{3}{n}(1 + \frac{12}{\alpha\mu n})4\alpha^2\ell_f^2 + 24M\ell_f^4 c_2 \left( \frac{8\gamma T}{n} + \gamma \right) \frac{\alpha}{n} \right) \left( \frac{1}{4}\alpha\mu \right)^{-1} \left( 1 - \left( 1 - \frac{1}{4}\alpha\mu \right)^T \right)$$

(B.78)

where the last equality follows from taking $p = \alpha\mu n/12$, and compute the sum of geometric series. By plugging in $c_1 = \mu/(4\ell_F)$, $c_2 = \ell_F c_1^{-1} + 2\alpha\ell_F^2$, $c_3 = 3\ell_F^2 + 2\ell_{g,1}$, we have that

$$\mathbb{E}[\delta_T^2] \leq \left( \frac{3}{n}(1 + \frac{12}{\alpha\mu n})4\alpha^2\ell_f^2 + 24M\ell_f^4 c_2 c_3^{-1}\frac{\alpha}{n^2} + 24M\ell_f^4 c_2 \frac{\alpha\gamma}{n} \right) \left( \frac{1}{4}\alpha\mu \right)^{-1}$$

$$\leq \frac{48}{\mu n}\ell_f^2 \left( \alpha + \frac{12}{\mu n} + \frac{2M\ell_f^2 c_2 c_3^{-1}}{n} + 2M\ell_f^2 c_2\gamma \right)$$

$$\leq \frac{48}{\mu n}\ell_f^2 \left( \alpha + \frac{12 + 4M\ell_f^2}{\mu n} + \frac{10M\ell_f^4\gamma}{\mu} \right)$$

(B.79)

where the last inequality follows from $c_2 = \ell_F^2(4\mu^{-1} + 2\alpha) \leq 5M\ell_f^2\mu^{-1}$, and $c_2 c_3^{-1} \leq 5\ell_F^2\mu^{-1}/(3\ell_F^2) \leq 2\mu^{-1}$. □

### B.4.4 Lower bound of MOL uniform stability

In this section, we construct Example 1 with a lower bound of stability for the MoDo algorithm. Before proceeding to the example, we first define $\mathbf{1}$ as the all-one vector in $\mathbb{R}^M$, $\widetilde{\Delta}^M := \{\lambda \in \mathbb{R}^M \mid \mathbf{1}^\top\lambda = 1\}$, and $P_\mathbf{1} := I - \frac{1}{M}\mathbf{1}\mathbf{1}^\top$. Then given any vector $u \in \mathbb{R}^M$, $\Pi_{\widetilde{\Delta}^M}(u) = P_\mathbf{1}u + \frac{1}{M}\mathbf{1}$.

**Example 1.** *Recall that $S = \{z_1, z_2, \ldots, z_j, \ldots z_n\}$, $S' = \{z_1, z_2, \ldots, z'_j, \ldots, z_n\}$, where $S$ and $S'$ differ only in the $j$-th data point. Define the $m$-th objective function as*

$$f_{z,m}(x) = \frac{1}{2}x^\top A x - b_m z^\top x$$

(B.80)

*where $A$ is a symmetric positive definite matrix, $\mu = 16n^{-\frac{1}{3}} > 0$ is the smallest eigenvalue of $A$, and $v$ is the corresponding eigenvector of $A$. For the datasets $S$, and $S'$, let $z_i = c_i v$, with $\mathbb{E}_{z\in S}[z] = \mu v$, $\mathbb{E}_{z\in S'}[z] = \mu' v$, $z_j - z'_j = v$, i.e., $\mu - \mu' = \frac{1}{n}$. For simplicity, let $M = 2$, $b = [1, 1 + \sqrt{2}]^\top$ such that $P_\mathbf{1}b = b_P = [-\frac{1}{\sqrt{2}}, \frac{1}{\sqrt{2}}]^\top$, where $b_P$ is the eigenvector of $P_\mathbf{1}$ with eigenvalue 1.*

**Technical challenges.**   One challenge of deriving the lower bound lies in the projection operator when updating $\lambda$. Unlike deriving the upper bound, where the projection operator can be handled by its non-expansive property, due to the nature of inequality constrained quadratic programming, neither a simple closed-form solution can be obtained, nor a non-trivial tight lower bound can be derived in general. We overcome this challenge by showing that when $\gamma = \mathcal{O}(T^{-1})$, the Euclidean projection onto the simplex is equivalent to a linear operator in Lemma 10. Another challenge compared to deriving the lower bound for single-objective learning is that the update of MoDo involves two coupled sequences, $\{x_t\}$ and $\{\lambda_t\}$. The update of $x_t$ and $x'_t$ involves different weighting parameters $\lambda_t$ and $\lambda'_t$, where $\{x_t\}$, $\{\lambda_t\}$ and $\{x'_t\}$, $\{\lambda'_t\}$ are generated by the MoDo algorithm on neighboring training data $S$ and $S'$, respectively. We overcome this challenge by deriving a recursive relation of the vector $[x_t - x'_t; \lambda_t - \lambda'_t]$ in Lemma 11.

**Organization of proof.**   Lemma 10 proves that under proper choice of initialization of $\lambda$ and step size $\gamma$, the projection of the updated $\lambda$ onto simplex $\Delta^M$ is equal to the projection of that onto the set $\widetilde{\Delta}^M := \{\lambda \in \mathbb{R}^M \mid \mathbf{1}^\top\lambda = 1\}$. And thus the projection is equivalent to a linear transformation. Thanks to Lemma 10, we are able to derive a recursive relation the vector $[x_t - x'_t; \lambda_t - \lambda'_t]$ in Lemma 11. Finally, relying on the recursive relation, we derive a lower bound for the recursion of $[\mathbb{E}_A\|x_t - x'_t\|; \mathbb{E}_A\|\lambda_t - \lambda'_t\|]$, depending on a $2 \times 2$ transition matrix. And based on its eigen decomposition, we could compute the $T$-th power of such a transition matrix, which is used to derive the final lower bound of $\mathbb{E}_A\|A(S) - A(S')\|$ in Theorem 6.

**Lemma 10.** *Suppose Assumptions 1, 2 hold. For MoDo algorithm, choose $\lambda_0 = \frac{1}{M}\mathbf{1}$, $\gamma \leq \frac{1}{2MT\ell_F\ell_f}$, and define*

$$\lambda_t^+ := \lambda_t - \gamma\nabla F_{z_{t,1}}(x_t)^\top \nabla F_{z_{t,2}}(x_t)\lambda_t$$

(B.81)

*then the update of $\lambda_t$ for MoDo algorithm is $\lambda_{t+1} = \Pi_{\Delta^M}(\lambda_t^+)$.*

*Define the set* $\widetilde{\Delta}^M := \{\lambda \in \mathbb{R}^M \mid \mathbf{1}^\top \lambda = 1\}$, $P_{\mathbf{1}} := I - \frac{1}{M}\mathbf{1}\mathbf{1}^\top$, $\lambda_{P,t} := \Pi_{\widetilde{\Delta}^M}(\lambda_t^+) = P_{\mathbf{1}}\lambda_t^+ + \frac{1}{M}\mathbf{1}$. *Then for $t = 0, \ldots, T - 1$, it holds that*

$$\lambda_{t+1} = P_{\mathbf{1}}\left(\lambda_t - \gamma \nabla F_{z_{t,1}}(x_t)^\top \nabla F_{z_{t,2}}(x_t)\lambda_t\right) + \frac{1}{M}\mathbf{1} = P_{\mathbf{1}}\lambda_t^+ + \frac{1}{M}\mathbf{1}. \qquad (B.82)$$

*Proof.* By the update of $\lambda_t$, we have

$$\begin{aligned}
\|\lambda_{t+1} - \lambda_t\| &= \|\Pi_{\Delta^M}\left(\lambda_t - \gamma \nabla F_{z_{t,1}}(x_t)^\top \nabla F_{z_{t,2}}(x_t)\lambda_t\right) - \lambda_t\| \\
&\leq \|\lambda_t - \gamma \nabla F_{z_{t,1}}(x_t)^\top \nabla F_{z_{t,2}}(x_t)\lambda_t - \lambda_t\| \\
&\leq \gamma \|\nabla F_{z_{t,1}}(x_t)^\top \nabla F_{z_{t,2}}(x_t)\lambda_t\| \leq \gamma \ell_F \ell_f
\end{aligned} \qquad (B.83)$$

where the last inequality follows from Lemma 6, with $\ell_f = \ell_{f,1}(c_x + c_{x^*})$.

Then for all $t \in [T - 1]$, it holds that

$$\|\lambda_t - \lambda_0\| = \left\|\sum_{k=0}^{t-1} \lambda_{k+1} - \lambda_k\right\| \leq \sum_{k=0}^{t-1} \|\lambda_{k+1} - \lambda_k\| \leq \gamma t \ell_F \ell_f \leq \frac{t}{2MT} \qquad (B.84)$$

where the last inequality follows from $\gamma \leq \frac{1}{2MT\ell_F\ell_f}$.

Then for $t \in [T - 1]$, it holds that

$$\begin{aligned}
\|\lambda_t^+ - \lambda_0\| &\leq \|\lambda_t^+ - \lambda_t\| + \|\lambda_t - \lambda_0\| \\
&\leq \gamma \|\nabla F_{z_{t,1}}(x_t)^\top \nabla F_{z_{t,2}}(x_t)\lambda_t\| + \gamma t \ell_F \ell_f \leq \gamma(t+1)\ell_F \ell_f \leq \gamma T \ell_F \ell_f \leq \frac{1}{2M}.
\end{aligned}$$
$$(B.85)$$

By the update of $\lambda_t$, and the definition of projection,

$$\lambda_{t+1} = \Pi_{\Delta^M}(\lambda_t^+) = \arg\min_{\lambda \in \Delta^M} \|\lambda - \lambda_t^+\|^2. \qquad (B.86)$$

Also we have

$$\lambda_{P,t} = \Pi_{\widetilde{\Delta}^M}(\lambda_t^+) = \arg\min_{\lambda \in \widetilde{\Delta}^M} \|\lambda - \lambda_t^+\|^2. \qquad (B.87)$$

Let $\lambda_{P,t} = [\lambda_{P,t,1}, \ldots, \lambda_{P,t,M}]^\top$. Then it holds that

$$|\lambda_{P,t,m} - \lambda_{0,m}| \leq \|\lambda_{P,t} - \lambda_0\| = \|\Pi_{\widetilde{\Delta}^M}(\lambda_t^+) - \lambda_0\| \overset{(a)}{\leq} \|\lambda_t^+ - \lambda_0\| \overset{(b)}{\leq} \frac{1}{2M}$$

where $(a)$ follows from non-expansiveness of projection and that $\lambda_0 \in \widetilde{\Delta}^M$; $(b)$ follows from (B.85). Therefore, each element of $\lambda_{P,t}$ satisfies

$$0 \leq \frac{1}{M} - \frac{1}{2M} \leq \lambda_{0,m} - |\lambda_{P,t,m} - \lambda_{0,m}| \leq \lambda_{P,t,m} \leq \lambda_{0,m} + |\lambda_{P,t,m} - \lambda_{0,m}| \leq \frac{3}{2M} \leq 1$$
$$(B.88)$$

which shows that $\lambda_{P,t} = \Pi_{\widetilde{\Delta}^M}(\lambda_t^+) \in \Delta^M$. Therefore it holds that,

$$\|\lambda_{P,t} - \lambda_t^+\|^2 \overset{(c)}{\geq} \min_{\lambda \in \Delta^M} \|\lambda - \lambda_t^+\|^2 \overset{(d)}{\geq} \min_{\lambda \in \widetilde{\Delta}^M} \|\lambda - \lambda_t^+\|^2 \overset{(e)}{=} \|\lambda_{P,t} - \lambda_t^+\|^2 \qquad (B.89)$$

where $(c)$ is because $\lambda_{P,t} \in \Delta^M$; $(d)$ is because $\Delta^M \subset \widetilde{\Delta}^M$ by the definition of the simplex; $(e)$ is because $\lambda_{P,t} = \Pi_{\widetilde{\Delta}^M}(\lambda_t^+)$. Then the equality holds that

$$\|\lambda_{P,t} - \lambda_t^+\|^2 = \min_{\lambda \in \Delta^M} \|\lambda - \lambda_t^+\|^2 \qquad (B.90)$$

and

$$P_{\mathbf{1}}\lambda_t^+ + \frac{1}{M}\mathbf{1} = \lambda_{P,t} = \arg\min_{\lambda \in \Delta^M} \|\lambda - \lambda_t^+\|^2 = \Pi_{\lambda \in \Delta^M}(\lambda_t^+) = \lambda_{t+1}. \qquad (B.91)$$

The proof is complete. $\qquad\qquad\qquad\qquad\qquad\qquad\qquad\qquad\qquad\qquad\qquad\qquad\qquad\square$

With the help of Lemma 10, which simplifies the Euclidean projection operator as a linear operator, we then prove in Lemma 11, the recursive relation of $x_t - x_t'$ and $\lambda_t - \lambda_t'$.

**Lemma 11.** *Suppose Assumptions 1, 2 hold. Under Example 1, choose $\lambda_0 = \frac{1}{M}\mathbf{1}$, $\gamma \le \frac{1}{2MT\ell_F\ell_f}$ for the MoDo algorithm. Denote $\{x_t\}$, $\{\lambda_t\}$ and $\{x_t'\}$, $\{\lambda_t'\}$ as the sequences generated by the MoDo algorithm with dataset $S$ and $S'$, respectively. Then it holds that*

$$x_t - x_t' = \varphi_{x,t}v, \quad and \quad \lambda_t - \lambda_t' = \varphi_{\lambda,t}b_P \tag{B.92}$$

*and $\varphi_{x,t}$, $\varphi_{\lambda,t}$ satisfy the following recursion*

$$\varphi_{x,t+1} = (1 - \alpha\mu + \alpha\gamma c_{t,3}c_{t,1}\mu)\varphi_{x,t} + \alpha c_{t,3}(1 - \gamma c_{t,1}c_{t,2})\varphi_{\lambda,t} + \mathbb{1}(i_{t,3}=j)\alpha(b^\top\lambda_{t+1}')$$
$$+ \mathbb{1}(i_{t,1}=j)\gamma\alpha c_{t,3}(\mu v^\top x_t' - c_{t,2}b^\top\lambda_t') - \mathbb{1}(i_{t,2}=j)\gamma\alpha c_{t,3}c_{t,1}'b^\top\lambda_t' \tag{B.93}$$

$$\varphi_{\lambda,t+1} = (1 - \gamma c_{t,1}c_{t,2})\varphi_{\lambda,t} + \gamma c_{t,1}\mu\varphi_{x,t}$$
$$+ \mathbb{1}(i_{t,1}=j)\gamma\big(\mu(v^\top x_t') - c_{t,2}(b^\top\lambda_t')\big) - \mathbb{1}(i_{t,2}=j)\gamma c_{t,1}'(b^\top\lambda_t') \tag{B.94}$$

*where $\mathbb{1}(\cdot)$ is the indicator function.*

*Proof.* Denote $z_{t,s}$, and $z_{t,s}'$, $s \in [3]$, as the samples selected in the $t$-th iteration from $S$ and $S'$, respectively. According to the MoDo algorithm update of $x_t$, and the definition of the problem in (B.80), we have

$$x_{t+1} = x_t - \alpha\nabla F_{z_{t,3}}(x_t)\lambda_{t+1} = x_t - \alpha[Ax_t - b_1z_{t,3}, \ldots, Ax_t - b_Mz_{t,3}]\lambda_{t+1}$$
$$= x_t - \alpha Ax_t + \alpha z_{t,3}(b^\top\lambda_{t+1}). \tag{B.95}$$

The difference $x_{t+1} - x_{t+1}'$ can be computed by

$$x_{t+1} - x_{t+1}' = (I - \alpha A)(x_t - x_t') + \alpha(z_{t,3}b^\top\lambda_{t+1} - z_{t,3}'b^\top\lambda_{t+1}')$$
$$= (I - \alpha A)(x_t - x_t') + \alpha z_{t,3}b^\top(\lambda_{t+1} - \lambda_{t+1}') + \alpha(z_{t,3} - z_{t,3}')b^\top\lambda_{t+1}'$$
$$= (I - \alpha A)(x_t - x_t') + \alpha z_{t,3}b^\top(\lambda_{t+1} - \lambda_{t+1}') + \mathbb{1}(i_{t,3}=j)\alpha v b^\top\lambda_{t+1}' \tag{B.96}$$

where $\mathbb{1}(\cdot)$ denotes the indicator function, and the last equation follows from that $z_{t,3} - z_{t,3}' = 0$ if $i_{t,3} \ne j$, and $z_{t,3} - z_{t,3}' = z_j - z_j' = v$ if $i_{t,3} = j$.

By Lemma 10, in Example 1, $\lambda_{t+1} = P_\mathbf{1}\big(\lambda_t - \gamma\nabla F_{z_{t,1}}(x_t)^\top\nabla F_{z_{t,2}}(x_t)\lambda_t\big) + \frac{1}{M}\mathbf{1}$, which can be further derived as

$$\lambda_{t+1} = P_\mathbf{1}\big(\lambda_t - \gamma\nabla F_{z_{t,1}}(x_t)^\top\nabla F_{z_{t,2}}(x_t)\lambda_t\big) + \frac{1}{M}\mathbf{1}$$
$$= P_\mathbf{1}\Big(\lambda_t - \gamma\big(\mathbf{1}x_t^\top A - bz_{t,1}^\top\big)\big(Ax_t - z_{t,2}b^\top\lambda_t\big)\Big) + \frac{1}{M}\mathbf{1}$$
$$\overset{(a)}{=} \lambda_t - \gamma\big(P_\mathbf{1}\mathbf{1}x_t^\top A - P_\mathbf{1}bz_{t,1}^\top\big)\big(Ax_t - z_{t,2}b^\top\lambda_t\big)$$
$$\overset{(b)}{=} \lambda_t + \gamma b_P z_{t,1}^\top\big(Ax_t - z_{t,2}b^\top\lambda_t\big) \tag{B.97}$$

where $(a)$ follows from rearranging the equation and that $P_\mathbf{1}\lambda_t + \frac{1}{M}\mathbf{1} = \Pi_{\widetilde{\Delta}^M}(\lambda_t) = \lambda_t$ as $\lambda_t \in \widetilde{\Delta}^M$; $(b)$ follows from that $P_\mathbf{1}b = b_P$ and $P_\mathbf{1}\mathbf{1} = 0$.

The difference $\lambda_{t+1} - \lambda_{t+1}'$ can be derived as

$$\lambda_{t+1} - \lambda_{t+1}' = (\lambda_t - \lambda_t') + \gamma b_P\big(z_{t,1}^\top Ax_t - z_{t,1}'^\top Ax_t'\big) - \gamma b_P\big(z_{t,1}^\top z_{t,2}b^\top\lambda_t - z_{t,1}'^\top z_{t,2}'b^\top\lambda_t'\big)$$
$$\overset{(c)}{=} (\lambda_t - \lambda_t') + \gamma b_P z_{t,1}^\top A(x_t - x_t') + \gamma b_P(z_{t,1} - z_{t,1}')^\top Ax_t' - \gamma b_P z_{t,1}^\top z_{t,2}b^\top(\lambda_t - \lambda_t')$$
$$\quad - \gamma b_P(z_{t,1} - z_{t,1}')^\top z_{t,2}b^\top\lambda_t' - \gamma b_P z_{t,1}'^\top(z_{t,2} - z_{t,2}')b^\top\lambda_t'$$
$$\overset{(d)}{=} (\lambda_t - \lambda_t') + \gamma b_P z_{t,1}^\top A(x_t - x_t') - \gamma b_P z_{t,1}^\top z_{t,2}b^\top(\lambda_t - \lambda_t')$$
$$\quad + \mathbb{1}(i_{t,1}=j)\gamma b_P(\mu v^\top x_t' - v^\top z_{t,2}b^\top\lambda_t') - \mathbb{1}(i_{t,2}=j)\gamma b_P z_{t,1}'^\top v b^\top\lambda_t' \tag{B.98}$$

where $(c)$ follows from rearranging the equation; $(d)$ follows from that $z_{t,s} - z_{t,s}' = 0$ if $i_{t,s} \ne j$, and $z_{t,s} - z_{t,s}' = z_j - z_j' = v$ if $i_{t,s} = j$, and that $Av = \mu v$.

Combining (B.96) and (B.98) gives

$$\begin{bmatrix} x_{t+1} - x'_{t+1} \\ \lambda_{t+1} - \lambda'_{t+1} \end{bmatrix} = \begin{bmatrix} C_{x,x,t} & C_{x,\lambda,t} \\ C_{\lambda,x,t} & C_{\lambda,\lambda,t} \end{bmatrix} \begin{bmatrix} x_t - x'_t \\ \lambda_t - \lambda'_t \end{bmatrix} + \mathbb{1}(i_{t,3} = j)\alpha \begin{bmatrix} vb^\top \lambda'_{t+1} \\ 0 \end{bmatrix} - \mathbb{1}(i_{t,2} = j)\gamma \begin{bmatrix} \alpha c_{t,3} c'_{t,1} vb^\top \lambda'_t \\ b_P c'_{t,1} b^\top \lambda'_t \end{bmatrix}$$

$$+ \mathbb{1}(i_{t,1} = j)\gamma \begin{bmatrix} \alpha c_{t,3} v(\mu v^\top x'_t - c_{t,2} b^\top \lambda'_t) \\ b_P(\mu v^\top x'_t - c_{t,2} b^\top \lambda'_t) \end{bmatrix} \tag{B.99}$$

where the matrices are defined as

$$C_{x,x,t} = I - \alpha A + \alpha\gamma z_{t,3} b^\top b_P z_{t,1}^\top A = I - \alpha A + \alpha\gamma c_{t,3} c_{t,1} \mu vv^\top \tag{B.100a}$$

$$C_{x,\lambda,t} = \alpha z_{t,3} b^\top (I - \gamma b_P z_{t,1}^\top z_{t,2} b^\top) = \alpha c_{t,3} vb^\top (1 - \gamma c_{t,1} c_{t,2}) \tag{B.100b}$$

$$C_{\lambda,x,t} = \gamma b_P z_{t,1}^\top A = \gamma c_{t,1} \mu b_P v^\top \tag{B.100c}$$

$$C_{\lambda,\lambda,t} = (I - \gamma b_P z_{t,1}^\top z_{t,2} b^\top) = (I - \gamma c_{t,1} c_{t,2} b_P b^\top). \tag{B.100d}$$

Next we show by induction that

$$x_t - x'_t = \varphi_{x,t} v, \quad \text{and} \quad \lambda_t - \lambda'_t = \varphi_{\lambda,t} b_P. \tag{B.101}$$

First, when $t = 0$, $x_0 - x'_0 = 0 = \varphi_{x,0} v$ and $\lambda_0 - \lambda'_0 = \varphi_{\lambda,0} b_P$ with $\varphi_{x,0} = 0$ and $\varphi_{\lambda,0} = 0$. Therefore (B.101) holds at $t = 0$. Supposing that (B.101) holds for $t = k$, next we show that it also holds at $t = k + 1$.

At $t = k + 1$, applying (B.99) for $x_{k+1} - x'_{k+1}$, and substituting (B.100a), (B.100b) yields

$$\begin{aligned} x_{k+1} - x'_{k+1} &= [C_{x,x,k} \quad C_{x,\lambda,k}] \begin{bmatrix} x_k - x'_k \\ \lambda_k - \lambda'_k \end{bmatrix} + \mathbb{1}(i_{k,3} = j)\alpha vb^\top \lambda'_{k+1} \\ &\quad + \mathbb{1}(i_{k,1} = j)\gamma\alpha c_{k,3} v(\mu v^\top x'_k - c_{k,2} b^\top \lambda'_k) - \mathbb{1}(i_{k,2} = j)\gamma\alpha c_{k,3} c'_{k,1} vb^\top \lambda'_k \\ &= (I - \alpha A + \alpha\gamma c_{k,3} c_{k,1} \mu vv^\top)\varphi_{x,k} v + \alpha c_{k,3} v(1 - \gamma c_{k,1} c_{k,2})\varphi_{\lambda,k} + \mathbb{1}(i_{k,3} = j)\alpha vb^\top \lambda'_{k+1} \\ &\quad + \mathbb{1}(i_{k,1} = j)\gamma\alpha c_{k,3} v(\mu v^\top x'_k - c_{k,2} b^\top \lambda'_k) - \mathbb{1}(i_{k,2} = j)\gamma\alpha c_{k,3} c'_{k,1} vb^\top \lambda'_k \\ &= (1 - \alpha\mu + \alpha\gamma c_{k,3} c_{k,1} \mu)\varphi_{x,k} v + \alpha c_{k,3}(1 - \gamma c_{k,1} c_{k,2})\varphi_{\lambda,k} v + \mathbb{1}(i_{k,3} = j)\alpha(b^\top \lambda'_{k+1})v \\ &\quad + \mathbb{1}(i_{k,1} = j)\gamma\alpha c_{k,3} v(\mu v^\top x'_k - c_{k,2} b^\top \lambda'_k) - \mathbb{1}(i_{k,2} = j)\gamma\alpha c_{k,3} c'_{k,1} vb^\top \lambda'_k \\ &= \varphi_{x,k+1} v \end{aligned} \tag{B.102}$$

where $\varphi_{x,k+1}$ is computed by

$$\begin{aligned} \varphi_{x,k+1} &= (1 - \alpha\mu + \alpha\gamma c_{k,3} c_{k,1} \mu)\varphi_{x,k} + \alpha c_{k,3}(1 - \gamma c_{k,1} c_{k,2})\varphi_{\lambda,k} + \mathbb{1}(i_{k,3} = j)\alpha(b^\top \lambda'_{k+1}) \\ &\quad + \mathbb{1}(i_{k,1} = j)\gamma\alpha c_{k,3}(\mu v^\top x'_k - c_{k,2} b^\top \lambda'_k) - \mathbb{1}(i_{k,2} = j)\gamma\alpha c_{k,3} c'_{k,1} b^\top \lambda'_k. \end{aligned} \tag{B.103}$$

Therefore, for all $t \in [T]$, it holds that $x_t - x'_t = \varphi_{x,t} v$.

At $t = k + 1$, apply (B.99) for $\lambda_{k+1} - \lambda'_{k+1}$, and substitute (B.100c), (B.100d) yields

$$\begin{aligned} \lambda_{k+1} - \lambda'_{k+1} &= [C_{\lambda,x,k} \quad C_{\lambda,\lambda,k}] \begin{bmatrix} x_k - x'_k \\ \lambda_k - \lambda'_k \end{bmatrix} \\ &\quad + \mathbb{1}(i_{k,1} = j)\gamma(\mu b_P v^\top x'_k - c_{k,2} b_P b^\top \lambda'_k) - \mathbb{1}(i_{k,2} = j)\gamma c'_{k,1} b_P b^\top \lambda'_k \\ &= \gamma c_{k,1} \mu b_P v^\top \varphi_{x,k} v + (I - \gamma c_{k,1} c_{k,2} b_P b^\top)b_P \varphi_{\lambda,k} \\ &\quad + \mathbb{1}(i_{k,1} = j)\gamma(\mu b_P v^\top x'_k - c_{k,2} b_P b^\top \lambda'_k) - \mathbb{1}(i_{k,2} = j)\gamma c'_{k,1} b_P b^\top \lambda'_k \\ &= \gamma c_{t,1} \mu\varphi_{x,k} b_P + (1 - \gamma c_{k,1} c_{k,2})\varphi_{\lambda,k} b_P \\ &\quad + \mathbb{1}(i_{k,1} = j)\gamma(\mu(v^\top x'_k) - c_{k,2}(b^\top \lambda'_k))b_P - \mathbb{1}(i_{k,2} = j)\gamma c'_{k,1}(b^\top \lambda'_k)b_P \\ &= \varphi_{\lambda,k+1} b_P \end{aligned} \tag{B.104}$$

where $\varphi_{\lambda,k+1}$ is computed by

$$\begin{aligned} \varphi_{\lambda,k+1} &= \gamma c_{k,1} \mu\varphi_{x,k} + (1 - \gamma c_{k,1} c_{k,2})\varphi_{\lambda,k} \\ &\quad + \mathbb{1}(i_{k,1} = j)\gamma(\mu(v^\top x'_k) - c_{k,2}(b^\top \lambda'_k)) - \mathbb{1}(i_{k,2} = j)\gamma c'_{k,1}(b^\top \lambda'_k). \end{aligned} \tag{B.105}$$

Therefore, for all $t \in [T]$, it holds that $\lambda_t - \lambda'_t = \varphi_{\lambda,t} b_P$. $\qquad\square$

Lemma 11 provides the recursive relation of $x_t - x_t'$ and $\lambda_t - \lambda_t'$. And Lemma 12 below provides another property used to derive the lower bound of $\mathbb{E}[\|x_T - x_T'\|]$ in Theorem 6.

**Lemma 12.** *Suppose Assumptions 1, 2 hold. Under Example 1, choose $x_0 = x_0' = 7v$, $\alpha = \frac{1}{4\mu T}$ for the MoDo algorithm. Denote $\{x_t\}$, $\{\lambda_t\}$ and $\{x_t'\}$, $\{\lambda_t'\}$ as the sequences generated by the MoDo algorithm with dataset $S$ and $S'$, respectively. Then it holds that*

$$v^\top \mathbb{E}_A[Ax_t' - z_{t,2}b^\top \lambda_t'] \geq 0 \quad \text{and} \quad b^\top \mathbb{E}_A[\lambda_{t+1}'] \geq b^\top \mathbb{E}_A[\lambda_t']. \tag{B.106}$$

*Proof.* From the update of $x_t'$, we have

$$\begin{aligned} x_{t+1}' &= x_t' - \alpha Ax_t' + \alpha z_{t,3}'(b^\top \lambda_{t+1}') \\ &= x_t' - \alpha Ax_t' + \alpha c_{t,3}'v(b^\top \lambda_{t+1}') \end{aligned} \tag{B.107}$$

Suppose $x_t' = c_{x,t}'v$, then $x_{t+1}' = c_{x,t+1}'v$ with

$$c_{x,t+1}' = (1 - \alpha\mu)c_{x,t}' + \alpha c_{t,3}'(b^\top \lambda_{t+1}')$$
$$\text{and} \quad \mathbb{E}_A[x_{t+1}'] = (1 - \alpha\mu)\mathbb{E}_A[x_t'] + \alpha\mu'v\mathbb{E}_A(b^\top \lambda_{t+1}')$$

Applying the above inequality recursively gives

$$\begin{aligned} v^\top \mathbb{E}_A[x_t'] &= v^\top (1 - \alpha\mu)^t x_0 + \alpha\mu'\Big(\sum_{t'=0}^{t-1}(1 - \alpha\mu)^{t-1-t'}\mathbb{E}_A(b^\top \lambda_{t'+1}')\Big) \\ &\geq v^\top(1-\alpha\mu)^t x_0 + \alpha\mu' \frac{1 - (1-\alpha\mu)^t}{\alpha\mu} \qquad b^\top \lambda \geq 1 \text{ for all } \lambda \in \Delta^M \\ &= (1-\alpha\mu)^t\big(v^\top x_0 - \mu'\mu^{-1}\big) + \mu'\mu^{-1}. \end{aligned}$$

Since $x_0 = 7v$, $\mu'\mu^{-1} \leq 1$, it holds that

$$v^\top \mathbb{E}_A[x_t'] = (1-\alpha\mu)^t\big(7 - \mu'\mu^{-1}\big) + \mu'\mu^{-1} \geq 6(1-\alpha\mu)^t + \mu'\mu^{-1}.$$

Then it follows that

$$\begin{aligned} v^\top \mathbb{E}_A[Ax_t' - z_{t,2}b^\top \lambda_t'] &= \mu\mathbb{E}_A[v^\top x_t' - b^\top \lambda_t'] \geq \mu\Big(6(1-\alpha\mu)^t + \mu'\mu^{-1} - (1 + \sqrt{2})\Big) \\ &\geq \mu\Big(6(1 - \alpha\mu t) - (1+\sqrt{2})\Big) \qquad\qquad \mu'\mu^{-1} \geq 0 \\ &\geq \mu\Big(6(1 - \frac{1}{4}) - (1+\sqrt{2})\Big) \geq 0 \qquad\qquad \alpha = \frac{1}{4\mu T} \end{aligned}$$

By the update of $\lambda_t'$ from (B.97),

$$\begin{aligned} b^\top \mathbb{E}_A[\lambda_{t+1}' - \lambda_t'] &= b^\top \gamma b_P \mathbb{E}_A[z_{t,1}'^\top\big(Ax_t' - z_{t,2}'b^\top\lambda_t'\big)] \\ &= \gamma\mathbb{E}_A[\mu'v^\top\big(Ax_t' - z_{t,2}'b^\top\lambda_t'\big)] \\ &\qquad\qquad\qquad\qquad \mathbb{E}_A[z_{t,1}'] = \mu'v, z_{t,1}' \text{ independent of } \lambda_t' \text{ and } x_t' \\ &= \gamma\mu'v^\top\mathbb{E}_A[Ax_t' - \mu'vb^\top\lambda_t'] \qquad \mathbb{E}_A[z_{t,2}'] = \mu'v, z_{t,2}' \text{ independent of } \lambda_t' \\ &\geq \gamma\mu'v^\top\mathbb{E}_A[Ax_t' - \mu vb^\top\lambda_t'] \qquad\quad \mu' \leq \mu, b^\top\lambda \geq 1 \text{ for all } \lambda \in \Delta^M \\ &= \gamma\mu'v^\top\mathbb{E}_A[Ax_t' - z_{t,2}b^\top\lambda_t'] \geq 0. \qquad \mathbb{E}_A[z_{t,2}] = \mu v, z_{t,2} \text{ independent of } \lambda_t' \end{aligned}$$

The proof is complete. $\qquad\qquad\qquad\qquad\qquad\qquad\qquad\qquad\qquad\qquad\qquad\qquad\qquad\qquad\square$

**Theorem 6.** *Suppose Assumptions 1 and 2 hold. Under Example 1 with $M = 2$, choose $\lambda_0 = \frac{1}{M}\mathbf{1}$, $x_0 = x_0' = 7v$, $\alpha = \frac{1}{4\mu T}$, $0 < \gamma \leq \frac{1}{2MT\ell_F\ell_f}$, and $T \leq 4n^{\frac{2}{3}}$ for the MoDo algorithm. Denote $\{x_t\}$, $\{\lambda_t\}$ and $\{x_t'\}$, $\{\lambda_t'\}$ as the sequences generated by the MoDo algorithm with dataset $S$ and $S'$, respectively. Then it holds that*

$$\mathbb{E}[\|x_T - x_T'\|] \geq \frac{\gamma T}{2n^2} + \frac{1}{16n}. \tag{B.108}$$

*Proof.* Denote $\delta_t = \|x_t - x_t'\|$, $\zeta_t = \|\lambda_t - \lambda_t'\|$. From Lemma 11, it holds that

$$\mathbb{E}[\delta_{t+1}] = \mathbb{E}[|\varphi_{x,t+1}|\|v\|] = \mathbb{E}[|\varphi_{x,t+1}|] \geq \mathbb{E}[\varphi_{x,t+1}] \tag{B.109}$$

$$\mathbb{E}[\zeta_{t+1}] = \mathbb{E}[|\varphi_{\lambda,t+1}|\|v\|] = \mathbb{E}[|\varphi_{\lambda,t+1}|] \geq \mathbb{E}[\varphi_{\lambda,t+1}] \tag{B.110}$$

where $\varphi_{x,t}$, $\varphi_{\lambda,t}$ satisfy

$$\varphi_{x,t+1} = (1 - \alpha\mu + \alpha\gamma c_{t,3}c_{t,1}\mu)\varphi_{x,t} + \alpha c_{t,3}(1 - \gamma c_{t,1}c_{t,2})\varphi_{\lambda,t} + \mathbb{1}(i_{t,3} = j)\alpha(b^\top \lambda_{t+1}')$$
$$+ \mathbb{1}(i_{t,1} = j)\gamma\alpha c_{t,3}(\mu v^\top x_t' - c_{t,2}b^\top \lambda_t') - \mathbb{1}(i_{t,2} = j)\gamma\alpha c_{t,3}c_{t,1}'b^\top \lambda_t' \tag{B.111}$$

$$\varphi_{\lambda,t+1} = (1 - \gamma c_{t,1}c_{t,2})\varphi_{\lambda,t} + \gamma c_{t,1}\mu\varphi_{x,t}$$
$$+ \mathbb{1}(i_{t,1} = j)\gamma\big(\mu(v^\top x_t') - c_{t,2}(b^\top \lambda_t')\big) - \mathbb{1}(i_{t,2} = j)\gamma c_{t,1}'(b^\top \lambda_t'). \tag{B.112}$$

The expectation of $\varphi_{x,t+1}$ can be further bounded as

$$\mathbb{E}[\varphi_{x,t+1}] = \mathbb{E}[(1 - \alpha\mu + \alpha\gamma c_{t,3}c_{t,1}\mu)\varphi_{x,t} + \alpha c_{t,3}(1 - \gamma c_{t,1}c_{t,2})\varphi_{\lambda,t} + \frac{1}{n}\alpha(b^\top \lambda_{t+1}')$$
$$+ \frac{1}{n}\gamma\alpha c_{t,3}(\mu v^\top x_t' - c_{t,2}b^\top \lambda_t') - \frac{1}{n}\gamma\alpha c_{t,3}c_{t,1}'b^\top \lambda_t']$$
$$\overset{(a)}{\geq} (1 - \alpha\mu(1 - \gamma\mu^2))\mathbb{E}[\varphi_{x,t}] + \alpha\mu(1 - \gamma\mu^2)\mathbb{E}[\varphi_{\lambda,t}] + \frac{1}{n}\alpha b^\top \mathbb{E}[\lambda_{t+1}'] - \frac{1}{n}\gamma\alpha\mu\mu'b^\top \mathbb{E}[\lambda_t']$$
$$\geq (1 - \alpha\mu(1 - \gamma\mu^2))\mathbb{E}[\varphi_{x,t}] + \alpha\mu(1 - \gamma\mu^2)\mathbb{E}[\varphi_{\lambda,t}] + \frac{1}{n}\alpha b^\top \mathbb{E}[\lambda_t'](1 - \gamma\mu^2) \tag{B.113}$$

where $(a)$ follows from $\mathbb{E}_A[c_{t,s}] = \mu$, $\mathbb{E}_A[c_{t,s}'] = \mu' \leq \mu$, $\mathbb{E}[\lambda_{t+1}'] \geq \mathbb{E}[\lambda_t']$ by Lemma 12, and the fact that $c_{t,s}$ is independent of $\varphi_{x,t}$.

Similarly, the expectation of $\varphi_{\lambda,t+1}$ can be further bounded as

$$\mathbb{E}[\varphi_{\lambda,t+1}] = \mathbb{E}[(1 - \gamma c_{t,1}c_{t,2})\varphi_{\lambda,t} + \gamma c_{t,1}\mu\varphi_{x,t} + \frac{1}{n}\gamma\big(\mu(v^\top x_t') - c_{t,2}(b^\top \lambda_t')\big) - \frac{1}{n}\gamma c_{t,1}'(b^\top \lambda_t')]$$
$$\overset{(b)}{\geq} (1 - \gamma\mu^2)\mathbb{E}[\varphi_{\lambda,t}] + \gamma\mu^2\mathbb{E}[\varphi_{x,t}] - \frac{1}{n}\gamma\mu'b^\top \mathbb{E}[\lambda_t'] \tag{B.114}$$

where $(b)$ follows from $\mathbb{E}_A[c_{t,s}] = \mu$, $\mathbb{E}_A[c_{t,s}'] = \mu' \leq \mu$, and Lemma 12.

The above arguments prove that

$$\begin{bmatrix} \mathbb{E}[\delta_{t+1}] \\ \mathbb{E}[\zeta_{t+1}] \end{bmatrix} \geq \begin{bmatrix} \mathbb{E}[\varphi_{x,t+1}] \\ \mathbb{E}[\varphi_{\lambda,t+1}] \end{bmatrix} \geq \underbrace{\begin{bmatrix} (1 - \alpha\mu(1 - \gamma\mu^2)) & \alpha\mu(1 - \gamma\mu^2) \\ \gamma\mu^2 & (1 - \gamma\mu^2) \end{bmatrix}}_{B} \begin{bmatrix} \mathbb{E}[\varphi_{x,t}] \\ \mathbb{E}[\varphi_{\lambda,t}] \end{bmatrix} + \frac{1}{n}\begin{bmatrix} \alpha(1 - \gamma\mu^2) \\ -\mu'\gamma \end{bmatrix}$$

$$\tag{B.115}$$

where the inequality for vectors denotes the inequality of each corresponding element in the vectors, and matrix $B$ has $v_{B,1} = [1, 1]^\top$ as an eigenvector associated with the eigenvalue 1 because

$$Bv_{B,1} = \begin{bmatrix} (1 - \alpha\mu(1 - \gamma\mu^2)) & \alpha\mu(1 - \gamma\mu^2) \\ \gamma\mu^2 & (1 - \gamma\mu^2) \end{bmatrix} \begin{bmatrix} 1 \\ 1 \end{bmatrix} = \begin{bmatrix} 1 \\ 1 \end{bmatrix}. \tag{B.116}$$

Similarly, since

$$B \begin{bmatrix} \alpha(1 - \gamma\mu^2) \\ -\gamma\mu \end{bmatrix} = \begin{bmatrix} (1 - \alpha\mu(1 - \gamma\mu^2)) & \alpha\mu(1 - \gamma\mu^2) \\ \gamma\mu^2 & (1 - \gamma\mu^2) \end{bmatrix} \begin{bmatrix} \alpha(1 - \gamma\mu^2) \\ -\gamma\mu \end{bmatrix}$$
$$= \begin{bmatrix} (1 - \alpha\mu(1 - \gamma\mu^2))\alpha(1 - \gamma\mu^2) - \gamma\mu\alpha\mu(1 - \gamma\mu^2) \\ \gamma\mu^2\alpha(1 - \gamma\mu^2) - \gamma\mu(1 - \gamma\mu^2) \end{bmatrix}$$
$$= \begin{bmatrix} (1 - \alpha\mu)(1 - \gamma\mu^2)\alpha(1 - \gamma\mu^2) \\ -(1 - \alpha\mu)(1 - \gamma\mu^2)\gamma\mu \end{bmatrix} = (1 - \alpha\mu)(1 - \gamma\mu^2)\begin{bmatrix} \alpha(1 - \gamma\mu^2) \\ -\gamma\mu \end{bmatrix}, \tag{B.117}$$

then $v_{B,2} = [\alpha(1 - \gamma\mu^2), -\gamma\mu]^\top$ is another eigenvector of $B$ with a positive eigenvalue $(1 - \alpha\mu)(1 - \gamma\mu^2) < 1$. Let $Q_B = [v_{B,1}, v_{B,2}]$, which can be expressed as

$$Q_B = [v_{B,1}, v_{B,2}] = \begin{bmatrix} 1 & \alpha(1 - \gamma\mu^2) \\ 1 & -\gamma\mu \end{bmatrix}. \tag{B.118}$$

Then $B$ has eigenvalue decomposition $B = Q_B \Lambda_B Q_B^{-1}$, where $\Lambda_B = \text{diag}([1, (1 - \alpha\mu)(1 - \gamma\mu^2)])$, and thus $B^t = Q_B \Lambda_B^t Q_B^{-1}$ for $t \in [T]$.

Let $[\alpha(1 - \gamma\mu^2), -\mu'\gamma]^\top = Q_B[c_{B,1}, c_{B,2}]^\top$, where $[c_{B,1}, c_{B,2}]^\top = Q_B^{-1}[\alpha(1 - \gamma\mu^2), -\mu'\gamma]^\top$ can be computed by

$$
\begin{bmatrix} c_{B,1} \\ c_{B,2} \end{bmatrix} = Q_B^{-1} \begin{bmatrix} \alpha(1 - \gamma\mu^2) \\ -\mu'\gamma \end{bmatrix} = -\frac{1}{\alpha(1 - \gamma\mu^2) + \gamma\mu} \begin{bmatrix} -\gamma\mu & -\alpha(1 - \gamma\mu^2) \\ -1 & 1 \end{bmatrix} \begin{bmatrix} \alpha \\ -\mu'\gamma \end{bmatrix}
$$

$$
= \frac{1}{\alpha(1 - \gamma\mu^2) + \gamma\mu} \begin{bmatrix} \alpha\gamma(\mu - \mu')(1 - \gamma\mu^2) \\ \alpha + \mu'\gamma \end{bmatrix} \geq \begin{bmatrix} \frac{1}{2n}\gamma \\ 1 \end{bmatrix} \tag{B.119}
$$

where the last inequality follows from $\mu - \mu' = \frac{1}{n}$ and $\alpha(1 - \gamma\mu^2) \geq \frac{1}{2}\alpha = 1/(8\mu T) \geq \gamma\mu$ for $c_{B,1} \geq \frac{1}{2n}\gamma$, and $\alpha\mu^2 = \mu/(4T) = 4n^{-\frac{1}{3}}T^{-1} \geq n^{-1} = \mu - \mu'$, so that $\alpha + \mu'\gamma = \alpha + \gamma(\mu' - \mu + \mu) \geq \alpha + \gamma(\mu - \alpha\mu^2) = \alpha(1 - \gamma\mu^2) + \gamma\mu$ for $c_{B,2} \geq 1$.

Since all elements in $B$ are positive, multiplying $B$ on both sides preserves inequality. Applying (B.115) recursively yields

$$
\begin{bmatrix} \mathbb{E}[\delta_T] \\ \mathbb{E}[\zeta_T] \end{bmatrix} \geq \sum_{t=0}^{T-1} B^{T-1-t} \frac{1}{n} \begin{bmatrix} \alpha(1 - \gamma\mu^2) \\ -\mu'\gamma \end{bmatrix} = \sum_{t=0}^{T-1} B^{T-1-t} \frac{1}{n} \begin{bmatrix} \alpha(1 - \gamma\mu^2) \\ -\mu'\gamma \end{bmatrix} = \sum_{t=0}^{T-1} B^{T-1-t} \frac{1}{n} Q_B \begin{bmatrix} c_{B,1} \\ c_{B,2} \end{bmatrix}
$$

$$
= \frac{1}{n} \sum_{t=0}^{T-1} 1^{T-1-t} c_{B,1} v_{B,1} + \frac{1}{n} \sum_{t=0}^{T-1} \left((1 - \alpha\mu)(1 - \gamma\mu^2)\right)^{T-1-t} c_{B,2} v_{B,2}
$$

$$
\geq \frac{T}{n} c_{B,1} v_{B,1} + \frac{1}{8n\alpha} c_{B,2} v_{B,2} \geq \frac{\gamma T}{2n^2} v_{B,1} + \frac{1}{8n\alpha} v_{B,2} \tag{B.120}
$$

where the last inequality follows from $c_{B,1} \geq \frac{1}{2n}\gamma$, and $c_{B,2} \geq 1$. Plugging in $v_{B,1} = [1, 1]^\top$ and $v_{B,2} = [\alpha(1 - \gamma\mu^2), -\gamma\mu]^\top$, and since $(1 - \gamma\mu^2) \geq \frac{1}{2}$, it follows that $\mathbb{E}[\delta_T] \geq \frac{\gamma T}{2n^2} + \frac{1}{16n}$. $\qquad\square$

### B.4.5 Proof of Theorem 2

**Proof of Theorem 2.** Combining the argument stability in Theorem 5, and Assumption 1, the MOL uniform stability can be bounded by

$$
\sup_z \mathbb{E}_A[\|\nabla F_z(A(S)) - \nabla F_z(A(S'))\|_F^2]
$$

$$
\leq \mathbb{E}_A[\ell_{F,1}^2 \|A(S) - A(S')\|^2] \qquad \text{by Assumption 1}
$$

$$
\leq \frac{48}{\mu n} \ell_f^2 \ell_{F,1}^2 \left( \alpha + \frac{12 + 4M\ell_f^2}{\mu n} + \frac{10M\ell_f^4 \gamma}{\mu} \right). \tag{B.121}
$$

Then based on Propositions 2-3, we have

$$
\mathbb{E}_{A,S}[R_{\text{gen}}(A(S))] \leq \mathbb{E}_{A,S}[\|\nabla F(A(S)) - \nabla F_S(A(S))\|_F] \qquad \text{by Proposition 2}
$$

$$
\leq 4\epsilon_F + \sqrt{n^{-1} \mathbb{E}_S\left[\mathbb{V}_{z\sim\mathcal{D}}(\nabla F_z(A(S)))\right]} \qquad \text{by Proposition 3}
$$

$$
= \mathcal{O}(n^{-\frac{1}{2}}). \qquad \text{by (B.121)}
$$

The proof of the upper bound is complete. We then prove the MOL uniform stability lower bound based on the argument uniform stability lower bound in Theorem 6. By the strong convexity of the function $f_{m,z}(x)$, for all $m \in [M]$

$$
\sup_z \mathbb{E}_A[\|\nabla F_z(A(S)) - \nabla F_z(A(S'))\|_F^2] \geq \mathbb{E}_A[M\mu^2 \|A(S) - A(S')\|^2] \qquad \text{by Assumption 2}
$$

$$
\geq \frac{M\mu^2}{256n^2}. \qquad \text{by Theorem 6 and Jensen's inequality}
$$

The proof of the lower bound is complete. $\qquad\square$

## C  Bounding the optimization error

### C.1  Auxiliary lemmas

**Lemma 13** (Uniqueness of CA direction). *Given $Q \in \mathbb{R}^{d \times M}$, then $d_Q := Q\lambda^*$ with $\lambda^* \in \arg\min_{\lambda \in \Delta^M} \|Q\lambda\|^2$ exists, and $d_Q$ is unique.*

*Proof.* This is a standard result due to convexity of the subproblem. Proof is given in [9, Section 2]. $\square$

**Lemma 14.** *For any $x \in \mathbb{R}^d$, define $\lambda^*(x)$ such that*

$$\lambda^*(x) \in \arg\min_{\lambda \in \Delta^M} \|\nabla F(x)\lambda\|^2. \tag{C.1}$$

*Then, for any $x \in \mathbb{R}^d$ and $\lambda \in \Delta^M$, it holds that*

$$\langle \nabla F(x)\lambda^*(x), \nabla F(x)\lambda \rangle \geq \|\nabla F(x)\lambda^*(x)\|^2, \tag{C.2}$$

$$\text{and } \|\nabla F(x)\lambda - \nabla F(x)\lambda^*(x)\|^2 \leq \|\nabla F(x)\lambda\|^2 - \|\nabla F(x)\lambda^*(x)\|^2. \tag{C.3}$$

*Proof.* By the first order optimality condition for (C.1), for any $x \in \mathbb{R}^d$ and $\lambda \in \Delta^M$, we have

$$\langle \nabla F(x)^\top \nabla F(x)\lambda^*(x), \lambda - \lambda^*(x) \rangle \geq 0. \tag{C.4}$$

By rearranging the above inequality, we obtain

$$\langle \nabla F(x)\lambda^*(x), \nabla F(x)\lambda \rangle \geq \|\nabla F(x)\lambda^*(x)\|^2, \tag{C.5}$$

which is precisely the first inequality in the claim. Furthermore, we can also have

$$\begin{aligned}
\|\nabla F(x)\lambda - \nabla F(x)\lambda^*(x)\|^2 &= \|\nabla F(x)\lambda\|^2 + \|\nabla F(x)\lambda^*(x)\|^2 - 2\langle \nabla F(x)\lambda^*(x), \nabla F(x)\lambda \rangle \\
&\leq \|\nabla F(x)\lambda\|^2 + \|\nabla F(x)\lambda^*(x)\|^2 - 2\|\nabla F(x)\lambda^*(x)\|^2 \\
&= \|\nabla F(x)\lambda\|^2 - \|\nabla F(x)\lambda^*(x)\|^2, \tag{C.6}
\end{aligned}$$

which is the desired second inequality in the claim. Hence, the proof is complete. $\square$

**Lemma 15** (Continuity of $\lambda_\rho^*(x)$). *Given any $\rho > 0$ and $x \in \mathbb{R}^d$, define $\lambda_\rho^*(x) = \arg\min_{\lambda \in \Delta^M} \frac{1}{2}\|\nabla F_S(x)\lambda\|^2 + \frac{1}{2}\rho\|\lambda\|^2$, then the following inequality holds*

$$\|\lambda_\rho^*(x) - \lambda_\rho^*(x')\| \leq \rho^{-1}\|\nabla F(x)^\top \nabla F(x) - \nabla F(x')^\top \nabla F(x')\|. \tag{C.7}$$

*Suppose either 1) Assumptions 1, 3 hold, or 2) Assumptions 1, 2 hold, with $\ell_F$ defined in Lemma 1. Then for $x \in \{x_t\}_{t=1}^T$, $x' \in \{x_t'\}_{t=1}^T$ generated by MoDo algorithm on training dataset $S$ and $S'$, respectively, it implies that*

$$\|\lambda_\rho^*(x) - \lambda_\rho^*(x')\| \leq 2\rho^{-1}\ell_{F,1}\ell_F\|x - x'\|. \tag{C.8}$$

*Proof.* We provide proof leveraging the convergence properties of the projected gradient descent algorithm on strongly convex objectives below. Consider the problem $\min_{\lambda \in \Delta^M} g(\lambda; x, \rho) = \frac{1}{2}\|\nabla F_S(x)\lambda\|^2 + \frac{1}{2}\rho\|\lambda\|^2$, which is $\rho$-strongly convex. Let $\{\lambda_{\rho,k}(x)\}$ for $k = 0, 1, \ldots, K$ denote the sequence obtained from applying projected gradient descent (PGD) on the objective $g(\lambda; x, \rho) = \frac{1}{2}\|\nabla F_S(x)\lambda\|^2 + \frac{1}{2}\rho\|\lambda\|^2$, i.e.,

$$\begin{aligned}
\lambda_{\rho,k+1}(x) &= \Pi_{\Delta^M}\Big(\lambda_{\rho,k}(x) - \eta\nabla F_S(x)^\top \nabla F_S(x)\lambda_{\rho,k}(x) - \eta\rho\lambda_{\rho,k}(x)\Big) \\
&= \Pi_{\Delta^M}\Big(\big((1 - \eta\rho)I - \eta\nabla F_S(x)^\top \nabla F_S(x)\big)\lambda_{\rho,k}(x)\Big) \tag{C.9}
\end{aligned}$$

where $\eta$ is the step size that $\eta < 1/(\|\nabla F_S(x)^\top \nabla F_S(x)\| + \rho)$. Note that both $\rho, \eta$ are independent of $K$. By the convergence result of PGD on strongly convex objective, we know that $\lambda_\rho^*(x)$ is the limit point of $\{\lambda_{\rho,k}(x)\}_{k=0}^\infty$. By the non-expansiveness of projection, we have

$$\|\lambda_{\rho,k+1}(x) - \lambda_{\rho,k+1}(x')\|$$

$$\leq \|\big((1-\eta\rho)I - \eta\nabla F_S(x)^\top \nabla F_S(x)\big)\lambda_{\rho,k}(x) - \big((1-\eta\rho)I - \eta\nabla F_S(x')^\top \nabla F_S(x')\big)\lambda_{\rho,k}(x')\|$$

$$\leq \|(1-\eta\rho)I - \eta\nabla F_S(x)^\top \nabla F_S(x)\|\|\lambda_{\rho,k}(x) - \lambda_{\rho,k}(x')\|$$

$$+ \eta\|(\nabla F_S(x)^\top \nabla F_S(x) - \nabla F_S(x')^\top \nabla F_S(x'))\lambda_{\rho,k}(x')\|$$

$$\leq (1-\eta\rho)\|\lambda_{\rho,k}(x) - \lambda_{\rho,k}(x')\| + \eta\|(\nabla F_S(x)^\top \nabla F_S(x) - \nabla F_S(x')^\top \nabla F_S(x'))\lambda_{\rho,k}(x')\|. \tag{C.10}$$

Since $\lambda_{\rho,0}(x) = \lambda_{\rho,0}(x') = \lambda_{\rho,0}$, apply the above inequality recursively from $k = 0, 1, \ldots, K-1$, we have

$$\|\lambda_{\rho,K}(x) - \lambda_{\rho,K}(x')\| \leq \eta\|(\nabla F_S(x)^\top \nabla F_S(x) - \nabla F_S(x')^\top \nabla F_S(x'))\lambda_{\rho,k}(x')\|\Big(\sum_{k=0}^{K-1}(1-\eta\rho)^k\Big)$$

$$= \eta\|(\nabla F_S(x)^\top \nabla F_S(x) - \nabla F_S(x')^\top \nabla F_S(x'))\lambda_{\rho,k}(x')\|\frac{1-(\eta\rho)^K}{\eta\rho}$$

$$\leq \rho^{-1}(1-(\eta\rho)^K)\|\nabla F_S(x)^\top \nabla F_S(x) - \nabla F_S(x')^\top \nabla F_S(x')\|. \tag{C.11}$$

Then it follows that

$$\|\lambda_\rho^*(x) - \lambda_\rho^*(x')\| \leq \lim_{K\to\infty}\big(\|\lambda_\rho^*(x) - \lambda_{\rho,K}(x)\| + \|\lambda_\rho^*(x') - \lambda_{\rho,K}(x')\| + \|\lambda_{\rho,K}(x) - \lambda_{\rho,K}(x')\|\big)$$

$$\leq \lim_{K\to\infty}\big(\|\lambda_\rho^*(x) - \lambda_{\rho,K}(x)\| + \|\lambda_\rho^*(x') - \lambda_{\rho,K}(x')\|\big)$$

$$+ \lim_{K\to\infty}\rho^{-1}(1-(\eta\rho)^K)\|\nabla F_S(x)^\top \nabla F_S(x) - \nabla F_S(x')^\top \nabla F_S(x')\|$$

$$\overset{(a)}{\leq} \rho^{-1}\|\nabla F_S(x)^\top \nabla F_S(x) - \nabla F_S(x')^\top \nabla F_S(x')\| + \lim_{K\to\infty}2\sqrt{\frac{4}{\rho\eta K}}$$

$$\leq \rho^{-1}\|\nabla F_S(x)^\top \nabla F_S(x) - \nabla F_S(x')^\top \nabla F_S(x')\| \tag{C.12}$$

where $(a)$ follows from $\lim_{K\to\infty}1-(\eta\rho)^K = 1$, and from the convergence of PGD [2, Theorem 1.1] on $\rho$-strongly convex objectives that

$$\|\lambda_\rho^*(x) - \lambda_{\rho,K}(x)\|^2 \leq \frac{2}{\rho}\Big(g(\lambda_{\rho,K}(x); x, \rho) - g(\lambda_\rho^*(x); x, \rho)\Big) \leq \frac{2}{\rho}\frac{\|\lambda_{\rho,0}(x) - \lambda_\rho^*(x)\|^2}{2\eta K} \leq \frac{4}{\rho\eta K}.$$

This proves (C.7).

In addition, under Assumptions 1, 3, the above result directly implies that

$$\|\lambda_\rho^*(x) - \lambda_\rho^*(x')\| \leq \rho^{-1}\|\nabla F_S(x)^\top \nabla F_S(x) - \nabla F_S(x')^\top \nabla F_S(x')\|$$

$$\leq \rho^{-1}\|\nabla F_S(x) + \nabla F_S(x')\|\|\nabla F_S(x) - \nabla F_S(x')\|$$

$$\leq 2\rho^{-1}\ell_{F,1}\ell_F\|x - x'\|. \tag{C.13}$$

While under Assumptions 1 and 2, and for $\ell_F$ defined in Lemma 1, and for $x \in \{x_t\}_{t=1}^T$, $x' \in \{x'_t\}_{t=1}^T$ generated by MoDo algorithm on training dataset $S$ and $S'$, respectively, $\|\nabla F_S(x)\| \leq \ell_F$, $\|\nabla F_S(x')\| \leq \ell_F$, which along with (C.12) implies that

$$\|\lambda_\rho^*(x) - \lambda_\rho^*(x')\| \leq \rho^{-1}\|\nabla F_S(x) + \nabla F_S(x')\|\|\nabla F_S(x) - \nabla F_S(x')\| \leq 2\rho^{-1}\ell_{F,1}\ell_F\|x - x'\|.$$

The proof is complete. $\qquad\square$

**Lemma 16.** *For any $\rho > 0$ and $x \in \mathbb{R}^d$, define $\lambda^*(x) = \arg\min_{\lambda\in\Delta^M}\|\nabla F_S(x)\lambda\|^2$, and $\lambda_\rho^*(x) = \arg\min_{\lambda\in\Delta^M}\|\nabla F_S(x)\lambda\|^2 + \rho\|\lambda\|^2$, then we have*

$$0 \leq \|\nabla F_S(x)\lambda_\rho^*(x)\|^2 - \|\nabla F_S(x)\lambda^*(x)\|^2 \leq \rho\left(1 - \frac{1}{M}\right). \tag{C.14}$$

*Proof.* Since $\lambda^*(x) = \arg\min_{\lambda\in\Delta^M}\|\nabla F_S(x)\lambda\|^2$, therefore

$$\|\nabla F_S(x)\lambda_\rho^*(x)\|^2 - \|\nabla F_S(x)\lambda^*(x)\|^2 = \|\nabla F_S(x)\lambda_\rho^*(x)\|^2 - \min_{\lambda\in\Delta^M}\|\nabla F_S(x)\lambda\|^2 \geq 0. \tag{C.15}$$

Since $\lambda_\rho^*(x) = \arg\min_{\lambda \in \Delta^M} \|\nabla F_S(x)\lambda\|^2 + \rho\|\lambda\|^2$, therefore

$$\|\nabla F_S(x)\lambda^*(x)\|^2 + \rho\|\lambda^*(x)\|^2 - \|\nabla F_S(x)\lambda_\rho^*(x)\|^2 - \rho\|\lambda_\rho^*(x)\|^2 \geq 0. \qquad \text{(C.16)}$$

Rearranging the above inequality gives

$$\|\nabla F_S(x)\lambda_\rho^*(x)\|^2 - \|\nabla F_S(x)\lambda^*(x)\|^2 \leq \rho\|\lambda^*(x)\|^2 - \rho\|\lambda_\rho^*(x)\|^2 \leq \rho\left(1 - \frac{1}{M}\right). \qquad \text{(C.17)}$$

$\qquad\qquad\qquad\qquad\qquad\qquad\qquad\qquad\qquad\qquad\qquad\qquad\qquad\qquad\qquad\qquad\qquad\qquad\qquad\quad$ $\square$

**Lemma 17.** *Consider $\{x_t\}$, $\{\lambda_t\}$ generated by the MoDo algorithm. For all $\lambda \in \Delta^M$, it holds that*

$$2\gamma_t \mathbb{E}_A \langle \lambda_t - \lambda, (\nabla F_S(x_t)^\top \nabla F_S(x_t))\lambda_t \rangle$$
$$\leq \mathbb{E}_A\|\lambda_t - \lambda\|^2 - \mathbb{E}_A\|\lambda_{t+1} - \lambda\|^2 + \gamma_t^2 \mathbb{E}_A\|(\nabla F_{z_{t,1}}(x_t)^\top \nabla F_{z_{t,2}}(x_t))\lambda_t\|^2, \qquad \text{(C.18)}$$
*and* $\quad \gamma_t \mathbb{E}_A(\|\nabla F_S(x_t)\lambda_t\|^2 - \|\nabla F_S(x_t)\lambda\|^2)$
$$\leq \mathbb{E}_A\|\lambda_t - \lambda\|^2 - \mathbb{E}_A\|\lambda_{t+1} - \lambda\|^2 + \gamma_t^2 \mathbb{E}_A\|(\nabla F_{z_{t,1}}(x_t)^\top \nabla F_{z_{t,2}}(x_t))\lambda_t\|^2. \qquad \text{(C.19)}$$

*Proof.* By the update of $\lambda$, for all $\lambda \in \Delta^M$, we have

$$\|\lambda_{t+1} - \lambda\|^2$$
$$= \|\Pi_{\Delta^M}(\lambda_t - \gamma_t(\nabla F_{z_{t,1}}(x_t)^\top \nabla F_{z_{t,2}}(x_t))\lambda_t) - \lambda\|^2$$
$$\leq \|\lambda_t - \gamma_t(\nabla F_{z_{t,1}}(x_t)^\top \nabla F_{z_{t,2}}(x_t))\lambda_t - \lambda\|^2$$
$$= \|\lambda_t - \lambda\|^2 - 2\gamma_t\langle\lambda_t - \lambda, (\nabla F_{z_{t,1}}(x_t)^\top \nabla F_{z_{t,2}}(x_t))\lambda_t\rangle + \gamma_t^2\|(\nabla F_{z_{t,1}}(x_t)^\top \nabla F_{z_{t,2}}(x_t))\lambda_t\|^2.$$

Taking expectation over $z_{t,1}, z_{t,2}$ on both sides and rearranging proves (C.18).

By the convexity of the problem, $\min_{\lambda \in \Delta^M} \frac{1}{2}\|\nabla F_S(x_t)\lambda\|^2$, we have

$$\gamma_t \mathbb{E}_A(\|\nabla F_S(x_t)\lambda_t\|^2 - \|\nabla F_S(x_t)\lambda\|^2)$$
$$\leq 2\gamma_t \mathbb{E}_A\langle\lambda_t - \lambda, (\nabla F_S(x_t)^\top \nabla F_S(x_t))\lambda_t\rangle$$
$$\overset{\text{(C.18)}}{\leq} \mathbb{E}_A\|\lambda_t - \lambda\|^2 - \mathbb{E}_A\|\lambda_{t+1} - \lambda\|^2 + \gamma_t^2 \mathbb{E}_A\|(\nabla F_{z_{t,1}}(x_t)^\top \nabla F_{z_{t,2}}(x_t))\lambda_t\|^2. \qquad \text{(C.20)}$$

Rearranging the above inequality proves (C.19). $\qquad\qquad\qquad\qquad\qquad\qquad\qquad\qquad$ $\square$

## C.2 Proof of Lemma 2 – Distance to CA direction

**Organization of proof.** In Lemma 18, we prove the upper bound of the distance to CA direction, $\frac{1}{T}\sum_{t=0}^{T-1} \mathbb{E}_A[\|\nabla F_S(x_t)\lambda_t - \nabla F_S(x_t)\lambda^*(x_t)\|^2]$, in terms of two average of sequences, $S_{1,T}$, and $S_{2,T}$. Then under either Assumptions 1, 3, or Assumptions 1, 2, we prove the upper bound of $S_{1,T}$, and $S_{2,T}$, and thus the average-iterate distance to CA direction in Lemma 2.

**Lemma 18.** *Suppose Assumption 1 holds. Let $\{x_t\}$, $\{\lambda_t\}$ be the sequences produced by the MoDo algorithm. With a positive constant $\bar\rho > 0$, define*

$$S_{1,T} = \frac{1}{T}\sum_{t=0}^{T-1} \mathbb{E}_A\|(\nabla F_{z_{t,1}}(x_t)^\top \nabla F_{z_{t,2}}(x_t))\lambda_t\|^2 \qquad \text{(C.21a)}$$

$$S_{2,T} = \frac{1}{T}\sum_{t=0}^{T-1} \mathbb{E}_A\|\nabla F_S(x_{t+1}) + \nabla F_S(x_t)\|\|\nabla F_{z_{t,3}}\lambda_{t+1}\|. \qquad \text{(C.21b)}$$

*Then it holds that*

$$\frac{1}{T}\sum_{t=0}^{T-1} \mathbb{E}_A[\|\nabla F_S(x_t)\lambda_t\|^2 - \|\nabla F_S(x_t)\lambda^*(x_t)\|^2] \leq \bar\rho + \frac{4}{\gamma T}(1 + \bar\rho^{-1}\alpha\ell_{F,1}TS_{2,T}) + \gamma S_{1,T}.$$

$$\text{(C.22)}$$

*Proof.* Define $\lambda_{\bar{\rho}}^*(x_t) = \arg\min_{\lambda \in \Delta^M} \frac{1}{2}\|\nabla F_S(x_t)\lambda\|^2 + \frac{\bar{\rho}}{2}\|\lambda\|^2$ with $\bar{\rho} > 0$. Note that $\bar{\rho}$ is strictly positive and is used only for analysis but not for algorithm update.

Substituting $\lambda = \lambda_{\bar{\rho}}^*(x_t)$ in Lemma 17, (C.19), we have

$$\gamma_t \mathbb{E}_A(\|\nabla F_S(x_t)\lambda_t\|^2 - \|\nabla F_S(x_t)\lambda_{\bar{\rho}}^*(x_t)\|^2)$$
$$\leq \mathbb{E}_A\|\lambda_t - \lambda_{\bar{\rho}}^*(x_t)\|^2 - \mathbb{E}_A\|\lambda_{t+1} - \lambda_{\bar{\rho}}^*(x_t)\|^2 + \gamma_t^2 \mathbb{E}_A\|(\nabla F_{z_{t,1}}(x_t)^\top \nabla F_{z_{t,2}}(x_t))\lambda_t\|^2. \quad \text{(C.23)}$$

Setting $\gamma_t = \gamma > 0$, taking expectation and telescoping the above inequality gives

$$\frac{1}{T}\sum_{t=0}^{T-1} \mathbb{E}_A[\|\nabla F_S(x_t)\lambda_t\|^2 - \|\nabla F_S(x_t)\lambda_{\bar{\rho}}^*(x_t)\|^2]$$

$$\leq \frac{1}{T}\sum_{t=0}^{T-1}\frac{1}{\gamma}\mathbb{E}_A[\|\lambda_t - \lambda_{\bar{\rho}}^*(x_t)\|^2 - \|\lambda_{t+1} - \lambda_{\bar{\rho}}^*(x_t)\|^2] + \frac{1}{T}\sum_{t=0}^{T-1}\gamma\mathbb{E}_A\|(\nabla F_{z_{t,1}}(x_t)^\top \nabla F_{z_{t,2}}(x_t))\lambda_t\|^2$$

$$= \frac{1}{\gamma T}\underbrace{\Big(\sum_{t=0}^{T-1}\mathbb{E}_A[\|\lambda_t - \lambda_{\bar{\rho}}^*(x_t)\|^2 - \|\lambda_{t+1} - \lambda_{\bar{\rho}}^*(x_t)\|^2]\Big)}_{I_1} + \frac{1}{T}\sum_{t=0}^{T-1}\gamma\mathbb{E}_A\|(\nabla F_{z_{t,1}}(x_t)^\top \nabla F_{z_{t,2}}(x_t))\lambda_t\|^2$$

$$\text{(C.24)}$$

where $I_1$ can be further derived as

$$I_1 = \sum_{t=0}^{T-1} \mathbb{E}_A\|\lambda_t - \lambda_{\bar{\rho}}^*(x_t)\|^2 - \mathbb{E}_A\|\lambda_{t+1} - \lambda_{\bar{\rho}}^*(x_t)\|^2$$

$$= \mathbb{E}_A\|\lambda_0 - \lambda_{\bar{\rho}}^*(x_0)\|^2 - \mathbb{E}_A\|\lambda_T - \lambda_{\bar{\rho}}^*(x_T)\|^2 + \sum_{t=0}^{T-2}\mathbb{E}_A[\|\lambda_{t+1} - \lambda_{\bar{\rho}}^*(x_{t+1})\|^2 - \|\lambda_{t+1} - \lambda_{\bar{\rho}}^*(x_t)\|^2]$$

$$\leq \mathbb{E}_A\|\lambda_0 - \lambda_{\bar{\rho}}^*(x_0)\|^2 - \mathbb{E}_A\|\lambda_T - \lambda_{\bar{\rho}}^*(x_T)\|^2$$

$$+ \sum_{t=0}^{T-2}\mathbb{E}_A[\|2\lambda_{t+1} - \lambda_{\bar{\rho}}^*(x_{t+1}) - \lambda_{\bar{\rho}}^*(x_t)\|\|\lambda_{\bar{\rho}}^*(x_{t+1}) - \lambda_{\bar{\rho}}^*(x_t)\|]$$

$$\leq 4 + 4\sum_{t=0}^{T-2}\mathbb{E}_A\|\lambda_{\bar{\rho}}^*(x_{t+1}) - \lambda_{\bar{\rho}}^*(x_t)\| \quad \text{(C.25)}$$

where $\|\lambda_{\bar{\rho}}^*(x_{t+1}) - \lambda_{\bar{\rho}}^*(x_t)\|$, by Lemma 15, can be bounded by

$$\|\lambda_{\bar{\rho}}^*(x_{t+1}) - \lambda_{\bar{\rho}}^*(x_t)\| \leq \bar{\rho}^{-1}\|\nabla F_S(x_{t+1}) + \nabla F_S(x_t)\|\|\nabla F_S(x_{t+1}) - \nabla F_S(x_t)\|$$
$$\leq \bar{\rho}^{-1}\ell_{F,1}\|\nabla F_S(x_{t+1}) + \nabla F_S(x_t)\|\|x_{t+1} - x_t\|$$
$$\leq \bar{\rho}^{-1}\alpha\ell_{F,1}\|\nabla F_S(x_{t+1}) + \nabla F_S(x_t)\|\|\nabla F_{z_{t,3}}\lambda_{t+1}\|. \quad \text{(C.26)}$$

Hence, it follows that

$$I_1 \leq 4 + 4\bar{\rho}^{-1}\alpha\ell_{F,1}\sum_{t=0}^{T-1}\mathbb{E}_A\|\nabla F_S(x_{t+1}) + \nabla F_S(x_t)\|\|\nabla F_{z_{t,3}}\lambda_{t+1}\|$$

$$= 4 + 4\bar{\rho}^{-1}\alpha\ell_{F,1}T S_{2,T} \quad \text{(C.27)}$$

plugging which into (C.24) gives

$$\frac{1}{T}\sum_{t=0}^{T-1}\mathbb{E}_A[\|\nabla F_S(x_t)\lambda_t\|^2 - \|\nabla F_S(x_t)\lambda_{\bar{\rho}}^*(x_t)\|^2] \leq \frac{4}{\gamma T}(1 + \bar{\rho}^{-1}\alpha\ell_{F,1}T S_{2,T}) + \gamma S_{1,T}. \quad \text{(C.28)}$$

Define $\lambda^*(x_t) \in \arg\min_{\lambda \in \Delta^M}\|\nabla F_S(x_t)\lambda\|^2$. Then

$$\frac{1}{T}\sum_{t=0}^{T-1}\mathbb{E}_A[\|\nabla F_S(x_t)\lambda_t\|^2 - \|\nabla F_S(x_t)\lambda^*(x_t)\|^2]$$

$$=\frac{1}{T}\sum_{t=0}^{T-1}\mathbb{E}_A[\|\nabla F_S(x_t)\lambda_t\|^2 - \|\nabla F_S(x_t)\lambda_{\bar\rho}^*(x_t)\|^2 + \|\nabla F_S(x_t)\lambda_{\bar\rho}^*(x_t)\|^2 - \|\nabla F_S(x_t)\lambda^*(x_t)\|^2]$$

$$\overset{(C.28)}{\leq}\frac{4}{\gamma T}(1+\bar\rho^{-1}\alpha\ell_{F,1}TS_{2,T}) + \gamma S_{1,T} + \frac{1}{T}\sum_{t=0}^{T-1}\mathbb{E}_A[\|\nabla F_S(x_t)\lambda_{\bar\rho}^*(x_t)\|^2 - \|\nabla F_S(x_t)\lambda^*(x_t)\|^2]$$

$$\leq\frac{4}{\gamma T}(1+\bar\rho^{-1}\alpha\ell_{F,1}TS_{2,T}) + \gamma S_{1,T} + \bar\rho \tag{C.29}$$

where the last inequality follows from Lemma 16. The proof is complete. $\square$

**Proof of Lemma 2.** Building on the result in Lemma 18, and by the convexity of the subproblem, $\min_{\lambda\in\Delta^M}\frac{1}{2}\|\nabla F_S(x_t)\lambda\|^2$, and Lemma 14, we have

$$\frac{1}{T}\sum_{t=0}^{T-1}\mathbb{E}_A[\|\nabla F_S(x_t)\lambda_t - \nabla F_S(x_t)\lambda^*(x_t)\|^2] \leq \frac{1}{T}\sum_{t=0}^{T-1}\mathbb{E}_A[\|\nabla F_S(x_t)\lambda_t\|^2 - \|\nabla F_S(x_t)\lambda^*(x_t)\|^2]$$

$$\leq\bar\rho + \frac{4}{\gamma T}(1+\bar\rho^{-1}\alpha\ell_{F,1}TS_{2,T}) + \gamma S_{1,T}. \tag{C.30}$$

By Assumptions 1, 3 or Assumptions 1, 2 and Lemma 1, we have

$$S_{1,T} = \frac{1}{T}\sum_{t=0}^{T-1}\mathbb{E}_A\|(\nabla F_{z_{t,1}}(x_t)^\top\nabla F_{z_{t,2}}(x_t))\lambda_t\|^2 \leq (\ell_f\ell_F)^2 \leq M\ell_f^4 \tag{C.31}$$

$$S_{2,T} = \frac{1}{T}\sum_{t=0}^{T-1}\mathbb{E}_A\|\nabla F_S(x_{t+1}) + \nabla F_S(x_t)\|\|\nabla F_{z_{t,3}}\lambda_{t+1}\| \leq 2\ell_f\ell_F. \tag{C.32}$$

Substituting $S_{1,T}, S_{2,T}$ in (C.30) with the above bound yields

$$\frac{1}{T}\sum_{t=0}^{T-1}\mathbb{E}_A[\|\nabla F_S(x_t)\lambda_t - \nabla F_S(x_t)\lambda^*(x_t)\|^2] \leq \bar\rho + \frac{4}{\gamma T}(1+2\bar\rho^{-1}\alpha T\ell_{F,1}\ell_f\ell_F) + \gamma M\ell_f^4. \tag{C.33}$$

Because $\ell_{F,1}\ell_F \leq M\ell_{f,1}\ell_f$, choosing $\bar\rho = 2(\alpha M\ell_{f,1}\ell_f^2/\gamma)^{\frac{1}{2}}$ yields

$$\frac{1}{T}\sum_{t=0}^{T-1}\mathbb{E}_A[\|\nabla F_S(x_t)\lambda_t - \nabla F_S(x_t)\lambda^*(x_t)\|^2] \overset{(a)}{\leq} \bar\rho + \frac{4}{\gamma T}(1+2\bar\rho^{-1}\alpha TM\ell_{f,1}\ell_f^2) + \gamma M\ell_f^4$$

$$=\frac{4}{\gamma T} + 6\sqrt{M\ell_{f,1}\ell_f^2\frac{\alpha}{\gamma}} + \gamma M\ell_f^4 \tag{C.34}$$

where $(a)$ follows from Lemma 14. This proves the result. $\square$

## C.3 Proof of Theorem 3 – PS optimization error

**Technical contributions.** The optimization error bound in Theorem 3 is improved with either relaxed assumption or improved convergence rate compared to prior stochastic MOL algorithms [52, 10, 30] (see Table 2). This is achieved by 1) instead of bounding the approximation error to $\lambda^*(x_t)$, we bound that to the CA direction $d(x_t) = -\nabla F_S(x_t)\lambda^*(x_t)$ as a whole, and 2) instead of using the descent lemma of $F_S(x_t)\lambda^*(x_t)$ with a dynamic weight, we use that of $F_S(x_t)\lambda$ with a fixed weight (see Lemma 19, (C.36)), thereby improving the tightness of the bound.

**Organization of proof.** In Lemma 19, we prove the upper bound of the PS optimization error, $\frac{1}{T}\sum_{t=0}^{T-1}\mathbb{E}_A\|\nabla F_S(x_t)\lambda_t^*(x_t)\|^2$, in terms of three average of sequences, $S_{1,T}, S_{3,T}$, and $S_{4,T}$. Then we prove the upper bound of $S_{1,T}, S_{3,T}$, and $S_{4,T}$, and thus the PS optimization error either in the nonconvex case under Assumptions 1, 3 or in the strongly convex case under Assumptions 1, 2. Combining the results leads to Theorem 3.

**Lemma 19.** *Suppose Assumption 1 holds. Consider the sequence $\{x_t\}, \{\lambda_t\}$ generated by MoDo in unbounded domain for $x$. Define*

$$S_{1,T} = \frac{1}{T} \sum_{t=0}^{T-1} \mathbb{E}_A \| \nabla F_{z_{t,1}}(x_t)^\top \nabla F_{z_{t,2}}(x_t) \lambda_t \|^2 \tag{C.35a}$$

$$S_{3,T} = \frac{1}{T} \sum_{t=0}^{T-1} \mathbb{E}_A \| \nabla F_{z_{t,1}}(x_t)^\top \nabla F_{z_{t,2}}(x_t) \lambda_t \| \| \nabla F_S(x_t)^\top \nabla F_S(x_t) \lambda_1 \| \tag{C.35b}$$

$$S_{4,T} = \frac{1}{T} \sum_{t=0}^{T-1} \mathbb{E}_A \| \nabla F_{z_{t,3}}(x_t) \lambda_{t+1} \|^2. \tag{C.35c}$$

*Then it holds that*

$$\frac{1}{T} \sum_{t=0}^{T-1} \mathbb{E}_A \| \nabla F_S(x_t) \lambda_t^*(x_t) \|^2 \leq \frac{1}{2\alpha T} \mathbb{E}_A [F_S(x_1) - F_S(x_{T+1})] \lambda_1 + \frac{1}{2} \gamma S_{1,T} + \gamma S_{3,T} + \frac{1}{2} \alpha \ell_{f,1} S_{4,T}.$$

*Proof.* By the $\ell_{f,1}$-Lipschitz smoothness of $F_S(x)\lambda$ for all $\lambda \in \Delta^M$, we have

$$\begin{aligned} F_S(x_{t+1})\lambda - F_S(x_t)\lambda &\leq \langle \nabla F_S(x_t)\lambda, x_{t+1} - x_t \rangle + \frac{\ell_{f,1}}{2} \|x_{t+1} - x_t\|^2 \\ &= -\alpha_t \langle \nabla F_S(x_t)\lambda, \nabla F_{z_{t,3}}(x_t)\lambda_{t+1} \rangle + \frac{\ell_{f,1}}{2} \alpha_t^2 \| \nabla F_{z_{t,3}}(x_t)\lambda_{t+1} \|^2. \end{aligned} \tag{C.36}$$

Taking expectation over $z_{t,3}$ on both sides of the above inequality gives

$$\mathbb{E}_{z_{t,3}}[F_S(x_{t+1})]\lambda - F_S(x_t)\lambda \leq -\alpha_t \langle \nabla F_S(x_t)\lambda, \nabla F_S(x_t)\lambda_{t+1} \rangle + \frac{\ell_{f,1}}{2} \alpha_t^2 \mathbb{E}_{z_{t,3}} \| \nabla F_{z_{t,3}}(x_t)\lambda_{t+1} \|^2. \tag{C.37}$$

By Lemma 17, (C.18), we have

$$\begin{aligned} &2\gamma_t \mathbb{E}_A \langle \lambda_t - \lambda, (\nabla F_S(x_t)^\top \nabla F_S(x_t))\lambda_t \rangle \\ &\leq \mathbb{E}_A \|\lambda_t - \lambda\|^2 - \mathbb{E}_A \|\lambda_{t+1} - \lambda\|^2 + \gamma_t^2 \mathbb{E}_A \|(\nabla F_{z_{t,1}}(x_t)^\top \nabla F_{z_{t,2}}(x_t))\lambda_t\|^2. \end{aligned} \tag{C.38}$$

Rearranging the above inequality and letting $\gamma_t = \gamma > 0$ gives

$$\begin{aligned} -\mathbb{E}_A \langle \lambda, \nabla F_S(x_t)^\top \nabla F_S(x_t)\lambda_t \rangle \leq &- \mathbb{E}_A \langle \lambda_t, (\nabla F_S(x_t)^\top \nabla F_S(x_t))\lambda_t \rangle + \frac{1}{2\gamma} \mathbb{E}_A(\|\lambda_t - \lambda\|^2 - \|\lambda_{t+1} - \lambda\|^2) \\ &+ \frac{1}{2} \gamma \mathbb{E}_A \|(\nabla F_{z_{t,1}}(x_t)^\top \nabla F_{z_{t,2}}(x_t))\lambda_t\|^2 \\ \leq &- \mathbb{E}_A \| \nabla F_S(x_t)\lambda_t \|^2 + \frac{1}{2\gamma} \mathbb{E}_A(\|\lambda_t - \lambda\|^2 - \|\lambda_{t+1} - \lambda\|^2) \\ &+ \frac{1}{2} \gamma \mathbb{E}_A \|(\nabla F_{z_{t,1}}(x_t)^\top \nabla F_{z_{t,2}}(x_t))\lambda_t\|^2. \end{aligned} \tag{C.39}$$

Plugging the above inequality into (C.37), and setting $\alpha_t = \alpha > 0$, we have

$$\begin{aligned} \mathbb{E}_A[F_S(x_{t+1})\lambda - F_S(x_t)\lambda] \leq &- \alpha \mathbb{E}_A \langle \nabla F_S(x_t)\lambda, \nabla F_S(x_t)\lambda_{t+1} \rangle + \frac{\ell_{f,1}}{2} \alpha^2 \mathbb{E}_A \| \nabla F_{z_{t,3}}(x_t)\lambda_{t+1} \|^2 \\ \leq &- \alpha \mathbb{E}_A \| \nabla F_S(x_t)\lambda_t \|^2 + \frac{\alpha}{2\gamma} \mathbb{E}_A[\|\lambda_t - \lambda\|^2 - \|\lambda_{t+1} - \lambda\|^2] \\ &+ \alpha \mathbb{E}_A \langle \nabla F_S(x_t)\lambda, \nabla F_S(x_t)(\lambda_t - \lambda_{t+1}) \rangle + \frac{1}{2} \alpha^2 \ell_{f,1} \mathbb{E}_A \| \nabla F_{z_{t,3}}(x_t)\lambda_{t+1} \|^2 \\ &+ \frac{1}{2} \alpha \gamma \mathbb{E}_A \|(\nabla F_{z_{t,1}}(x_t)^\top \nabla F_{z_{t,2}}(x_t))\lambda_t\|^2. \end{aligned} \tag{C.40}$$

Taking telescope sum and rearranging yields, for all $\lambda \in \Delta^M$,

$$\frac{1}{T}\sum_{t=0}^{T-1}\mathbb{E}_A\|\nabla F_S(x_t)\lambda_t\|^2$$

$$\leq \frac{1}{2\gamma T}\sum_{t=0}^{T-1}\mathbb{E}_A[\|\lambda_t - \lambda\|^2 - \|\lambda_{t+1} - \lambda\|^2] + \frac{1}{\alpha T}\sum_{t=0}^{T-1}\mathbb{E}_A[F_S(x_t) - F_S(x_{t+1})]\lambda$$

$$+ \frac{1}{2T}\sum_{t=0}^{T-1}\left(\gamma\mathbb{E}_A\|\nabla F_{z_{t,1}}(x_t)^\top\nabla F_{z_{t,2}}(x_t)\lambda_t\|^2 + \alpha\ell_{f,1}\mathbb{E}_A\|\nabla F_{z_{t,3}}(x_t)\lambda_{t+1}\|^2\right.$$

$$\left.+ 2\mathbb{E}_A\langle\nabla F_S(x_t)\lambda, \nabla F_S(x_t)(\lambda_t - \lambda_{t+1})\rangle\right)$$

$$\leq \frac{1}{2\gamma T}\mathbb{E}_A[\|\lambda_1 - \lambda\|^2 - \|\lambda_{T+1} - \lambda\|^2] + \frac{1}{\alpha T}\mathbb{E}_A[F_S(x_1) - F_S(x_{T+1})]\lambda + \frac{1}{2}\gamma S_{1,T} + \gamma S_{3,T} + \frac{1}{2}\alpha\ell_{f,1}S_{4,T}.$$

$$\text{(C.41)}$$

Setting $\lambda = \lambda_1$ in the above inequality yields

$$\frac{1}{T}\sum_{t=0}^{T-1}\mathbb{E}_A\|\nabla F_S(x_t)\lambda_t\|^2 \leq \frac{1}{\alpha T}\mathbb{E}_A[F_S(x_1) - F_S(x_{T+1})]\lambda_1 + \frac{1}{2}\gamma S_{1,T} + \gamma S_{3,T} + \frac{1}{2}\alpha\ell_{f,1}S_{4,T}$$

Finally, the results follow from the definition of $\lambda_t^*(x_t)$ such that $\frac{1}{T}\sum_{t=0}^{T-1}\mathbb{E}_A\|\nabla F_S(x_t)\lambda_t^*(x_t)\|^2 \leq \frac{1}{T}\sum_{t=0}^{T-1}\mathbb{E}_A\|\nabla F_S(x_t)\lambda_t\|^2.$ □

**Proof of Theorem 3.** Lemma 19 states that, under Assumption 1, we have

$$\frac{1}{T}\sum_{t=0}^{T-1}\mathbb{E}_A\|\nabla F_S(x_t)\lambda_t^*(x_t)\|^2 \leq \frac{1}{\alpha T}\mathbb{E}_A[F_S(x_1) - F_S(x_{T+1})]\lambda_1 + \frac{1}{2}\gamma S_{1,T} + \gamma S_{3,T} + \frac{1}{2}\alpha\ell_{f,1}S_{4,T}.$$

Then we proceed to bound $S_{1,T}, S_{3,T}, S_{4,T}$. Under either Assumptions 1, 3, or Assumptions 1, 2 with $\ell_f, \ell_F$ defined in Lemma 1, we have that for all $z \in S$ and $\lambda \in \Delta^M$, $\|\nabla F_z(x_t)\lambda\| \leq \ell_f$, and $\|\nabla F_z(x_t)\| \leq \ell_F$. Then $S_{1,T}, S_{3,T}, S_{4,T}$ can be bounded below

$$S_{1,T} = \frac{1}{T}\sum_{t=0}^{T-1}\mathbb{E}_A\|(\nabla F_{z_{t,1}}(x_t)^\top\nabla F_{z_{t,2}}(x_t))\lambda_t\|^2 \leq M\ell_f^4 \tag{C.42a}$$

$$S_{3,T} = \frac{1}{T}\sum_{t=0}^{T-1}\mathbb{E}_A\|\nabla F_{z_{t,1}}(x_t)^\top\nabla F_{z_{t,2}}(x_t)\lambda_t\|\|\nabla F_S(x_t)^\top\nabla F_S(x_t)\lambda_1\| \leq \ell_F^2\ell_f^2 = M\ell_f^4 \tag{C.42b}$$

$$S_{4,T} = \frac{1}{T}\sum_{t=0}^{T-1}\mathbb{E}_A\|\nabla F_{z_{t,3}}(x_t)\lambda_{t+1}\|^2 \leq \ell_f^2 \tag{C.42c}$$

which proves that

$$\frac{1}{T}\sum_{t=0}^{T-1}\mathbb{E}_A\|\nabla F_S(x_t)\lambda_t^*(x_t)\|^2 \leq \frac{1}{\alpha T}c_F + \frac{3}{2}\gamma M\ell_f^4 + \frac{1}{2}\alpha\ell_{f,1}\ell_f^2. \tag{C.43}$$

We arrive at the results by $\frac{1}{T}\sum_{t=0}^{T-1}\mathbb{E}_A\|\nabla F_S(x_t)\lambda_t^*(x_t)\| \leq \left(\frac{1}{T}\sum_{t=0}^{T-1}\mathbb{E}_A\|\nabla F_S(x_t)\lambda_t^*(x_t)\|^2\right)^{\frac{1}{2}}$ from the Jensen's inequality and the convexity of the square function, as well as the subadditivity of square root function. □

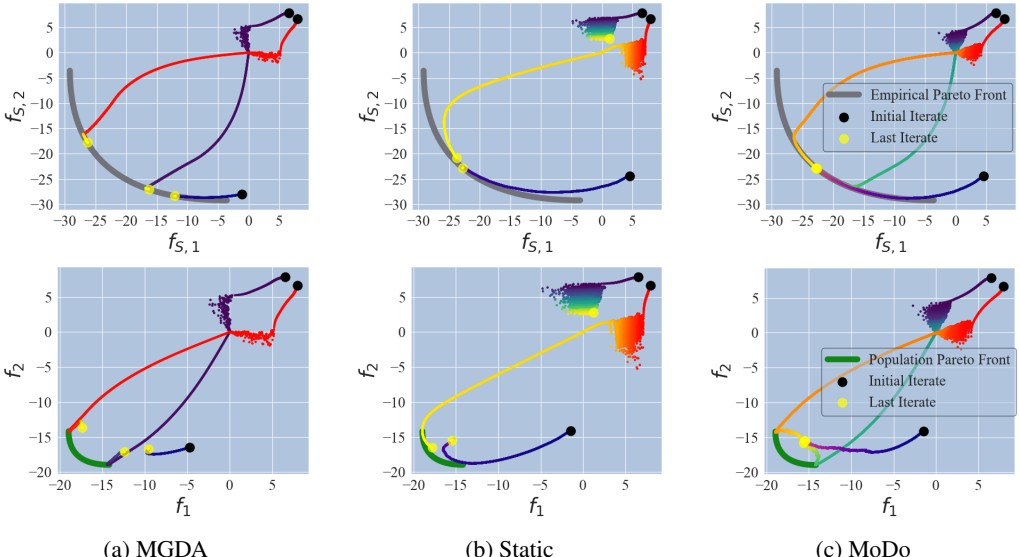

|  | (a) MGDA | (b) Static | (c) MoDo |

Figure 5: Convergence of MGDA, static weighting and MoDo to the **empirical (gray, upper)** and **population (green, lower)** Pareto fronts. Horizontal and vertical axes in figures in the first / second row are the values of the two empirical / population objectives. Three colormaps are used for the trajectories from three initializations, respectively, where the same colormaps represent the trajectories of the same initializations, and darker colors in one colormap indicate earlier iterations and lighter colors indicate later iterations.

## D    Additional experiments and implementation details

**Compute.**    Experiments are done on a machine with GPU NVIDIA RTX A5000. We use MATLAB R2021a for the synthetic experiments in strongly convex case, and Python 3.8, CUDA 11.7, Pytorch 1.8.0 for other experiments. Unless otherwise stated, all experiments are repeated with 5 random seeds. And their average performance and standard deviations are reported.

### D.1    Synthetic experiments

#### D.1.1    Experiments on strongly convex objectives

**Implementation details.**    Below we provide the details of experiments that generate Figure 3. We use the following synthetic example for the experiments in the strongly convex case. The $m$-th objective function with stochastic data sample $z$ is specified as

$$f_{z,m}(x) = \frac{1}{2}b_{1,m}x^\top A x - b_{2,m}z^\top x \tag{D.1}$$

where $b_{1,m} > 0$ for all $m \in [M]$, and $b_{2,m}$ is another scalar. We set $M = 3$, $b_1 = [b_{1,1}; b_{1,2}; b_{1,3}] = [1; 2; 1]$, and $b_2 = [b_{2,1}; b_{2,2}; b_{2,3}] = [1; 3; 2]$. The default parameters are $T = 100$, $\alpha = 0.01$, $\gamma = 0.001$. In other words, in Figure 3a, we fix $\alpha = 0.01, \gamma = 0.001$, and vary $T$; in Figure 3b, we fix $T = 100, \gamma = 0.001$, and vary $\alpha$; and in Figure 3c, we fix $T = 100, \alpha = 0.01$, and vary $\gamma$.

#### D.1.2    Experiments on nonconvex objectives

**Implementation details.**    The toy example is modified from [29] to consider stochastic data. Denote the model parameter as $x = [x_1, x_2]^\top \in \mathbb{R}^2$, stochastic data as $z = [z_1, z_2]^\top \in \mathbb{R}^2$ sampled from the standard multi-variate Gaussian distribution. And the individual empirical objectives are defined as:

$$f_{z,1}(x) = c_1(x)h_1(x) + c_2(x)g_{z,1}(x) \text{ and } f_{z,2}(x) = c_1(x)h_2(x) + c_2(x)g_{z,2}(x), \text{ where}$$
$$h_1(x) = \log(\max(|0.5(-x_1 - 7) - \tanh(-x_2)|, 0.000005)) + 6,$$
$$h_2(x) = \log(\max(|0.5(-x_1 + 3) - \tanh(-x_2) + 2|, 0.000005)) + 6,$$

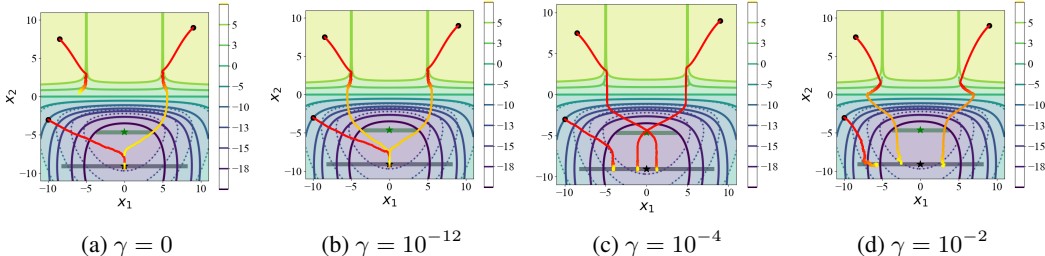

(a) $\gamma = 0$      (b) $\gamma = 10^{-12}$      (c) $\gamma = 10^{-4}$      (d) $\gamma = 10^{-2}$

Figure 6: Trajectories of MoDo under different $\gamma$ on the contour of the average of objectives. The **black** • marks initializations of the trajectories, colored from **red** (start) to **yellow** (end). The background solid/dotted contours display the landscape of the average empirical/population objectives. The **gray**/**green** bar marks empirical/population Pareto front, and the **black** ⋆/**green** ⋆ marks solution to the average objectives.

$$g_{z,1}(x) = ((-x_1 + 3.5)^2 + 0.1 * (-x_2 - 1)^2)/10 - 20 - 2 * z_1 x_1 - 5.5 * z_2 x_2,$$
$$g_{z,2}(x) = ((-x_1 - 3.5)^2 + 0.1 * (-x_2 - 1)^2)/10 - 20 + 2 * z_1 x_1 - 5.5 * z_2 x_2,$$
$$c_1(x) = \max(\tanh(0.5 * x_2), 0) \text{ and } c_2(x) = \max(\tanh(-0.5 * x_2), 0). \tag{D.2}$$

Since $z$ is zero-mean, the individual population objectives are correspondingly:

$$f_1(x) = c_1(x)h_1(x) + c_2(x)g_1(x) \text{ and } f_2(x) = c_1(x)h_2(x) + c_2(x)g_2(x), \text{ where}$$
$$g_1(x) = ((-x_1 + 3.5)^2 + 0.1 * (-x_2 - 1)^2)/10 - 20,$$
$$g_2(x) = ((-x_1 - 3.5)^2 + 0.1 * (-x_2 - 1)^2)/10 - 20. \tag{D.3}$$

The training dataset size is $n = |S| = 20$. For all methods, i.e., MGDA, static weighting, and MoDo, the number of iterations is $T = 50000$. The initialization of $\lambda$ is $\lambda_0 = [0.5, 0.5]^\top$. The hyperparameters for this experiment are summarized in Table 5.

Figure 5, in addition to Figure 1, shows the trajectories of different methods from different initializations to the empirical and population Pareto fronts (PF). With the visualized empirical and population PFs, it is clear in Figure 5a, the first row, that the three trajectories of MGDA all converge to the empirical PF, but, it stops updating the model parameter as soon as it reaches the empirical PF. However, due to the difference between the empirical and population PFs caused by finite stochastic training data, as shown in Figure 5a, the second row, not all three solutions from MGDA has small population risk, implied from the distance of the solution (yellow point) to the population PF colored in green. For static weighting method with uniform weights in Figure 5b, one trajectory is able to converge to the center of the empirical PF, which is the optimal solution of the uniform average of the two objectives. However, the other two get stuck and oscillate around suboptimal parameters for a long time, corresponding to the clusters of scattered points in the figure. Nevertheless, in the second row of Figure 5b, one empirically suboptimal solution (on the trajectory with red to yellow colormap) is able to achieve small population risk. This example demonstrates that even though static weighting method does not have small distance to CA direction, it might still be able to achieve small testing error. Finally, for MoDo in Figure 5c, the first row shows that MoDo is slower than MGDA in convergence to the empirical PF, since it only approximately solves the CA direction using

Table 5: Summary of hyper-parameter choices for nonconvex synthetic data.

|  | Static | MGDA | MoDo |
|---|---|---|---|
| optimizer of $x_t$ | Adam | Adam | Adam |
| $x_t$ step size ($\alpha_t$) | $5 \times 10^{-3}$ | $5 \times 10^{-3}$ | $5 \times 10^{-3}$ |
| $\lambda_t$ step size ($\gamma_t$) | - | - | $10^{-4}$ |
| batch size | 16 | full | 16 |

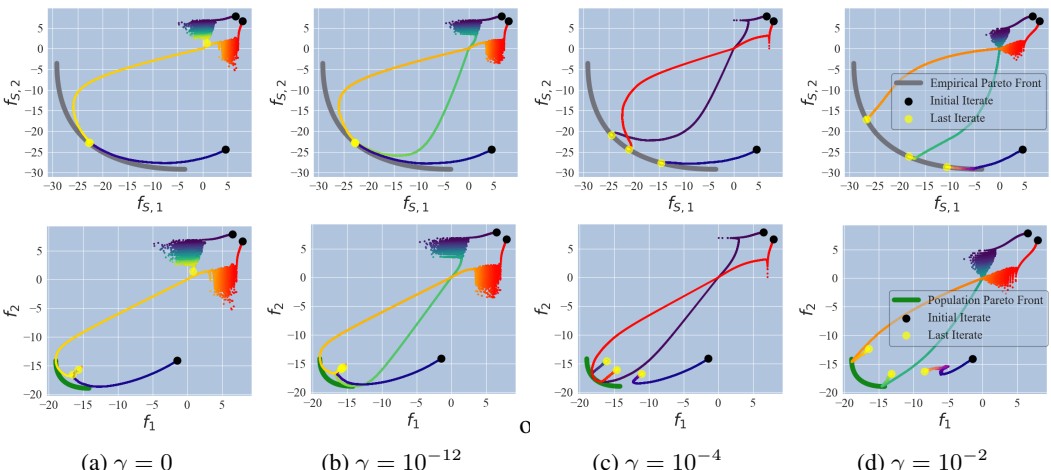

(a) $\gamma = 0$        (b) $\gamma = 10^{-12}$        (c) $\gamma = 10^{-4}$        (d) $\gamma = 10^{-2}$

Figure 7: Convergence of MoDo to the **empirical (gray, upper)** and **population (green, lower)** Pareto fronts under different $\gamma$. Horizontal and vertical axes in figures in the first / second row are the values of the two empirical / population objectives. Three colormaps are used for the trajectories from three initializations, respectively, where the same colormaps represent the trajectories of the same initializations, and darker colors in one colormap indicate earlier iterations and lighter colors indicate later iterations.

stochastic gradient. The second row shows that all three solutions of MoDo can achieve relatively small population risk, demonstrating a good generalization ability.

To demonstrate how the choice of $\gamma$ has an impact on the performance of MoDo, we further conduct experiments with different $\gamma$, and show that MoDo can recover static weighting ($\gamma = 0$) and approximate MGDA ($\gamma = 10^{-2}$). Results are visualized in Figure 6, which plots the trajectories over the contours of the average objectives, and Figure 7, which plots the empirical and population PFs. Figure 6a and Figure 7a together verify that MoDo with $\gamma = 0$ behaves the same as static weighting, cf. Figure 1d and Figure 5b. Like static weighting, for two of the initializations, they either cannot go through the valley or stuck in the valley for a long time. And if the trajectory converges, it will converge to the optimal solution of the average empirical objectives. In addition, Figure 6d and Figure 7d together indicate that MoDo with $\gamma = 10^{-2}$ behaves similarly as MGDA, cf. Figure 1c and Figure 5a. Like MGDA, it can go through the valley without getting stuck in it for a long time, and then converge to the empirical PF for all three initializations. But also like MGDA, it will stop updating or oscillating around the parameters as soon as it reaches the empirical PF, resulting in a worse population risk for one of the solutions with a trajectory colored from purple to pink to yellow. We also conduct experiments on $0 < \gamma < 10^{-2}$. With $\gamma = 10^{-12}$, as shown in Figures 6b, and 7b, MoDo is able to go through the valley and converge to the empirical PF for all initializations, but very slowly. And all initializations converge to the optimal solution of the average objectives. While with $\gamma = 10^{-4}$, as shown in Figures 6c, and 7c, MoDo is able to go through the valley and converge to the empirical PF for all initializations quickly. And it stops updating the parameters as soon as it reaches the empirical PF. The solutions of the three initializations are different from the optimal solution of the average objectives. But compared to the case when $\gamma = 10^{-2}$, the three solutions are closer to the optimal solution of the average objectives. This is because the weighting parameter $\lambda$ does not change too much during the optimization procedure with a small $\gamma$.

## D.2    Multi-task supervised learning experiments

In this section we present experiment details and additional results for comparison of static weighting, MGDA, and MoDo algorithms, under synthetic and real world multi-task supervised learning problems. We use MNIST, Office-31, Office-home, and NYU-v2 datasets.

### D.2.1    MNIST dataset experiments

**Implementation details.**    Below we provide the details and additional experimental results on MNIST image classification. We simulate a synthetic multi-objective optimization problem using

different loss functions applied for training an image classifier for MNIST handwritten digit dataset. We consider three loss functions: cross entropy, mean squared error (MSE), and Huber loss. The model architecture is a two-layer multi-layer perceptron (MLP). Each hidden layer has 512 neurons, and no hidden layer activation. The input size is 784, and the output size is 10, the number of digit classes. The training, validation, and testing data sizes are 50k, 10k, and 10k, respectively. Hyper-parameters such as step sizes are chosen based on each algorithm's validation accuracy performance, as given in Table 6.

We then discuss the results of the experiments in Table 7, which shows the performance of the last iterate for each method. Experiments are repeated 10 times with average performance and standard deviations reported. Observed from Table 7, for cross-entropy loss, MGDA performs the worst while static weighting performs the best. On the other hand, for Huber loss, MGDA performs the best while static weighting performs the worst. This is not surprising as cross-entropy loss has the largest scale and Huber loss has the smallest scale among the three losses. Since equal weights are assigned for all three objectives in the static weighting method, it tends to focus more on optimizing the loss with the largest scale. While for MGDA, it is the other way around. Compared to MGDA, MoDo performs much better on cross-entropy loss, and in the meantime, it achieves comparable performance on Huber loss. Comparing their PS population risks $R_{\mathrm{pop}}$ and the decomposed PS optimization errors $R_{\mathrm{opt}}$ and PS generalization errors $R_{\mathrm{gen}}$, MGDA has the smallest PS population risk and PS optimization error. One potential reason is that MGDA performs best on Huber loss, with smaller gradients. Another reason is that the generalization errors for all the algorithms are similar and not dominating compared to optimization errors in this setting, making the PS population risk close to the PS optimization error. Overall, MoDo demonstrates a balance among objectives with different scales and performs reasonably well on all three objectives since it combines the properties of static weighting and MGDA.

Table 6: Summary of hyper-parameter choices for MNIST image classification

|  | Static | MGDA | MoDo |
|---|---|---|---|
| optimizer of $x_t$ | SGD | SGD | SGD |
| $x_t$ step size ($\alpha_t$) | 0.1 | 5.0 | 1.0 |
| $\lambda_t$ step size ($\gamma_t$) | - | - | 1.0 |
| batch size | 64 | 64 | 64 |

Table 7: MNIST classification with cross-entropy, MSE, and Huber loss as objectives.

| Method | Cross-entropy Loss ($10^{-3}$)↓ | MSE loss Loss ($10^{-3}$)↓ | Huber loss Loss ($10^{-3}$)↓ | $R_{\mathrm{pop}}$ ($10^{-3}$)↓ | $R_{\mathrm{opt}}$ ($10^{-3}$)↓ | $|R_{\mathrm{gen}}|$ ($10^{-3}$)↓ |
|---|---|---|---|---|---|---|
| Static | **306.9±3.9** | 13.2±0.14 | 2.2±0.03 | 2.1±0.56 | 1.9±0.5 | 0.2±0.19 |
| MGDA | 363.6±4.1 | 13.5±0.13 | **1.9±0.01** | **1.3±0.24** | **1.1±0.2** | 0.2±0.13 |
| **MoDo** | 317.9±3.4 | **13.1±0.13** | 2.1±0.05 | 2.1±0.38 | 1.9±0.4 | **0.1±0.09** |

#### D.2.2 Office-31 and Office-home dataset experiments

**Implementation details.** We give the details for the experiments conducted using Office-31 and Office-home datasets, which consist of multi-domain image classification tasks. Both of these are multi-input single-task learning problems. Office-31 and Office-home consist of 31 and 65 image classes, respectively. The image domains for Office-31 are; "Amazon", which includes object images from Amazon, "DSLR", which includes high-resolution images of objects, and "Webcam", which includes low-resolution images from objects. The image domains for Office-31 are "Art", "Clipart", "Product", and "Real-world" which include images of objects taken from the respective image domains. We use the MTL benchmark framework LibMTL [26] to run experiments on both of the aforementioned datasets. For both datasets, we tune the step size of $x$ and weight decay parameters for Static and MGDA algorithms and tune the step sizes of $x$ and $\lambda$ for the MoDo algorithm. We use batch size 64 to update static weighting and MGDA, and use 2 independent samples of batch size 32 to update MoDo, for both Office-31 and Office-home. A summary of hyper-parameters used for

Table 8: Summary of hyper-parameter choices for Office-31 task

|  | Static | MGDA | MoDo |
|---|---|---|---|
| optimizer of $x_t$ | Adam | Adam | Adam |
| $x_t$ step size ($\alpha_t$) | $10^{-4}$ | $10^{-4}$ | $10^{-4}$ |
| $\lambda_t$ step size ($\gamma_t$) | - | - | $10^{-3}$ |
| weight decay | $10^{-3}$ | $10^{-7}$ | $10^{-5}$ |
| batch size | 64 | 64 | 64 |

Office-31 and Office-home for each algorithm are given in Table 8 and Table 9, respectively. All other experiment setup is shared for all algorithms, and the default LibMTL configuration is used.

The results of Office-31 and Office-home experiments are given in Tables 3 and 10, respectively (average over 5 seeds, the error indicates standard deviation). Here, we use the average per-task performance drop of metrics $S_{\mathcal{A},m}$ for method $\mathcal{A}$ with respect to corresponding baseline measures $S_{\mathcal{B},m}$ as a measure of the overall performance of a given method. Specifically, this measure is

$$\Delta\mathcal{A}\% = \frac{1}{M} \sum_{m=1}^{M} (-1)^{\ell_m} (S_{\mathcal{A},m} - S_{\mathcal{B},m})/S_{\mathcal{B},m} \times 100, \tag{D.4}$$

where $M$ is the number of tasks. Here, $\ell_m = 1$ if higher values for $S_{\mathcal{A},m}$ are better and 0 otherwise. We use the best results for each task obtained by dedicated independent task learners of each task as $S_{\mathcal{B},m}$. The independent task learners are tuned for the learning rate and weight decay parameter. For Office-31, $S_{\mathcal{B},m}$ values are 87.50% for "Amazon", 98.88% for "DSLR", and 97.32% for "Webcam". For Office-home, $S_{\mathcal{B},m}$ values are 66.98% for "Art", 82.02% for "Clipart", 91.53% for "Product", and 80.97% for "Real-world". It can be seen from Tables 3 and 10 that static weighting outperforms MGDA method in some tasks and also in terms of $\Delta\mathcal{A}\%$. However, by proper choices of hyper-parameters, MoDo performs on par or better compared to both static weighting and MGDA, and hence achieves the best overall performance in terms of $\Delta\mathcal{A}\%$.

Table 9: Summary of hyper-parameter choices for Office-home task

|  | Static | MGDA | MoDo |
|---|---|---|---|
| optimizer of $x_t$ | Adam | Adam | Adam |
| $x_t$ step size ($\alpha_t$) | $10^{-4}$ | $10^{-4}$ | $10^{-4}$ |
| $\lambda_t$ step size ($\gamma_t$) | - | - | $10^{-3}$ |
| weight decay | $10^{-3}$ | $10^{-6}$ | $10^{-5}$ |
| batch size | 64 | 64 | 64 |

Table 10: Classification results on Office-home dataset.

| Method | Art | Clipart | Product | Real-world | $\Delta\mathcal{A}\% \downarrow$ |
|---|---|---|---|---|---|
|  | Test Acc ↑ | Test Acc ↑ | Test Acc ↑ | Test Acc ↑ |  |
| Static | $64.14 \pm 1.40$ | $\mathbf{79.57 \pm 1.09}$ | $90.00 \pm 0.50$ | $78.94 \pm 0.87$ | $2.85 \pm 1.08$ |
| MGDA | $61.71 \pm 1.33$ | $73.95 \pm 0.43$ | $\mathbf{90.17 \pm 0.27}$ | $79.35 \pm 1.15$ | $5.29 \pm 0.47$ |
| **MoDo** | $\mathbf{65.50 \pm 0.55}$ | $79.44 \pm 0.29$ | $89.72 \pm 0.94$ | $\mathbf{79.65 \pm 0.67}$ | $\mathbf{2.24 \pm 0.48}$ |

### D.2.3 NYU-v2 dataset experiments

**Implementation details.** We give the details for the experiments conducted using NYU-v2 dataset, which consists of image segmentation, depth estimation, and surface normal estimation tasks. Unlike Office-31 and Office-home datasets, this is a single-input multi-task learning problem. The

dataset consists of images from indoor video sequences. We use the MTL benchmark framework LibMTL [26] to run experiments with this dataset. We tune step size of $x$ and weight decay parameters for static weighting and MGDA algorithms, and tune step sizes of $x$ and $\lambda$ for MoDo algorithm. We use batch size 4 to update static weighting and MGDA, and use 2 independent batches of size 2 to update MoDo. Experiments were run for 50 epochs for all methods. Since there was no validation set for NYU-v2, we averaged and reported the test performance of the last 10 epochs. A summary of hyper-parameters used for each algorithm is given in Table 11. All other experiment setup is shared for all algorithms, and the default LibMTL configuration is used.

Table 11: Summary of hyper-parameter choices for NYU-v2 task

|  | Static | MGDA | MoDo |
|---|---|---|---|
| optimizer of $x_t$ | Adam | Adam | Adam |
| $x_t$ step size ($\alpha_t$) | $10^{-4}$ | $10^{-4}$ | $10^{-4}$ |
| $\lambda_t$ step size ($\gamma_t$) | - | - | $10^{-3}$ |
| weight decay | $10^{-4}$ | $10^{-6}$ | $10^{-5}$ |
| batch size | 4 | 4 | 4 |

Table 12: Segmentation, depth, and surface normal estimation results on NYU-v2 dataset.

| Method | Segmetation (Higher Better) | | Depth (Lower Better) | | Surface Normal | | | | | $\triangle\mathcal{A}\%\downarrow$ |
|---|---|---|---|---|---|---|---|---|---|---|
| | | | | | Angle Distance (Lower Better) | | Within $t^{\circ}$ (Higher better) | | | |
| | mIoU | Pix Acc | Abs Err | Rel Err | Mean | Median | 11.25 | 22.5 | 30 | |
| Static | $52.02 \pm 0.69$ | $74.21 \pm 0.57$ | $\mathbf{0.3984 \pm 0.0032}$ | $\mathbf{0.1645 \pm 0.0010}$ | $23.79 \pm 0.10$ | $17.44 \pm 0.15$ | $34.07 \pm 0.17$ | $60.17 \pm 0.31$ | $71.48 \pm 0.29$ | $3.98 \pm 0.70$ |
| MGDA | $46.39 \pm 0.17$ | $70.27 \pm 0.24$ | $0.4269 \pm 0.0024$ | $0.1737 \pm 0.0009$ | $\mathbf{22.34 \pm 0.03}$ | $\mathbf{15.70 \pm 0.08}$ | $\mathbf{37.71 \pm 0.21}$ | $\mathbf{63.96 \pm 0.11}$ | $\mathbf{74.50 \pm 0.06}$ | $6.25 \pm 0.38$ |
| **MoDo** | $\mathbf{52.64 \pm 0.19}$ | $\mathbf{74.67 \pm 0.08}$ | $\mathbf{0.3984 \pm 0.0020}$ | $0.1649 \pm 0.0018$ | $23.45 \pm 0.06$ | $17.09 \pm 0.05$ | $34.79 \pm 0.11$ | $60.90 \pm 0.13$ | $72.12 \pm 0.11$ | $\mathbf{3.21 \pm 0.34}$ |

The results of NYU-v2 experiments are given in Tables 12 (average over 3 seeds, the error indicates standard deviation). Again, we use the average per-task performance drop of metrics $S_{\mathcal{A},m}$ for method $\mathcal{A}$ with respect to corresponding baseline measures $S_{\mathcal{B},m}$ as a measure of the overall performance of a given method. We use the best results for each task obtained by dedicated independent task learners of each task as $S_{\mathcal{B},m}$. The independent task learners are tuned for the learning rate and weight decay parameter. For segmentation task, $S_{\mathcal{B},m}$ values are $53.94\%$ for "mIoU", and $75.67\%$ for "Pix Acc". For depth estimation task, $S_{\mathcal{B},m}$ values are $0.3949$ for "Abs Err", and $0.1634$ for "Rel Err". For surface normal estimation task, $S_{\mathcal{B},m}$ values are $22.12$ for "Angle Distance - Mean", $15.49$ for "Angle Distance - Median", $38.35\%$ for "Within $11.25°$", $64.30\%$ for "Within $22.5°$", and $74.70\%$ for "Within $30°$". It can be seen from Table 12 that MoDo outperforms both MGDA and static weighting in some tasks and also in terms of $\triangle\mathcal{A}\%$. The worse performance of MGDA in terms of $\triangle\mathcal{A}\%$ can be due to the large bias towards the Surface normal estimation task, which seems to affect other tasks adversely.

### D.2.4 Additional experiments for comparison with other MOL baselines

In this section, we provide a comparison between MoDo and other popular MOL baselines. For this purpose, we use the same benchmark datasets as the previous section and use the experiment setup provided in [25] to run experiments with MoDo. Hence, we use experiment results provided by [25] for other baselines for comparison. Additionally, we implement MoCo [10], which is not included in [25]. For results in this section we report two holistic measures: $\triangle\mathcal{A}_{\text{st}}$, which measures performance degradation w.r.t. static scalarization (similar to [25], but lower the better), and $\triangle\mathcal{A}_{\text{id}}$, which measures performance degradation w.r.t. independent task learners (as defined in our paper)

The results on Office-31, Office-home, and NYU-v2 are given in Tables 13, 14, and 15, respectively, where MoDo outperforms all the baselines for most tasks, and has a better overall performance in $\triangle\mathcal{A}_{\text{st}}\%$ and $\triangle\mathcal{A}_{\text{id}}\%$. The hyper-parameters of MoDo for the above experiments are in Table 16.

Table 13: Comparison with other methods on Office-31 dataset.

| Method | Amazon | DSLR | Webcam | $\Delta\mathcal{A}_{st}\%\downarrow$ | $\Delta\mathcal{A}_{id}\%\downarrow$ |
|---|---|---|---|---|---|
| Static (EW) | 81.02 | 96.72 | 96.11 | 0.00 | 2.96 |
| MGDA-UB [39] | 81.02 | 95.90 | **97.77** | 0.40 | 3.32 |
| GradNorm [6] | 83.93 | 97.54 | 94.44 | -0.19 | 2.80 |
| PCGrad [48] | 82.22 | 96.72 | 95.55 | 0.40 | 3.35 |
| CAGrad [29] | 82.22 | 96.72 | 96.67 | 0.01 | 2.96 |
| RGW [25] | 84.27 | 96.72 | 96.67 | -0.81 | 2.18 |
| MoCo [10] | 85.30 | 97.54 | 97.22 | -1.70 | 1.32 |
| MoDo (ours) | **85.47** | **98.36** | 96.67 | **-1.86** | **1.17** |

Table 14: Comparison with other methods on Office-home dataset.

| Method | Art | Clipart | Product | Real-world | $\Delta\mathcal{A}_{st}\%\downarrow$ | $\Delta\mathcal{A}_{id}\%\downarrow$ |
|---|---|---|---|---|---|---|
| Static (EW) | 62.99 | 76.48 | 88.45 | 77.72 | 0.00 | 5.02 |
| MGDA-UB [25] | 64.32 | 75.29 | 89.72 | 79.35 | -1.02 | 4.04 |
| GradNorm [6] | 65.46 | 75.29 | 88.66 | 78.91 | -1.03 | 4.04 |
| PCGrad [48] | 63.94 | 76.05 | 88.87 | 78.27 | -0.53 | 4.51 |
| CAGrad [29] | 63.75 | 75.94 | 89.08 | 78.27 | -0.48 | 4.56 |
| RGW [25] | 65.08 | 78.65 | 88.66 | 79.89 | -2.30 | 2.85 |
| MoCo [10] | 64.14 | **79.85** | 89.62 | 79.57 | -2.48 | 2.68 |
| MoDo (ours) | **66.22** | 78.22 | **89.83** | **80.32** | **-3.08** | **2.11** |

Table 15: Comparison with other methods on NYU-v2 dataset.

| Method | Segmentation | | Depth | | Surface Normal | | | | | $\Delta\mathcal{A}_{st}\%\downarrow$ | $\Delta\mathcal{A}_{id}\%\downarrow$ |
|---|---|---|---|---|---|---|---|---|---|---|---|
| | (Higher Better) | | (Lower Better) | | Angle Distance (Lower Better) | | Within $t°$ (Higher better) | | | | |
| | mIoU | Pix Acc | Abs Err | Rel Err | Mean | Median | 11.25 | 22.5 | 30 | | |
| Static (EW) | 53.77 | 75.45 | 0.3845 | 0.1605 | 23.57 | 17.04 | 35.04 | 60.93 | 72.07 | 0.00 | 1.63 |
| MGDA-UB | 50.42 | 73.46 | 0.3834 | 0.1555 | 22.78 | 16.14 | 36.90 | 62.88 | 73.61 | -0.38 | 1.26 |
| GradNorm | 53.58 | 75.06 | 0.3931 | 0.1663 | 23.44 | 16.98 | 35.11 | 61.11 | 72.24 | 0.99 | 2.62 |
| PCGrad | 53.70 | 75.41 | 0.3903 | 0.1607 | 23.43 | 16.97 | 35.16 | 61.19 | 72.28 | 0.16 | 1.79 |
| CAGrad | 53.12 | 75.19 | 0.3871 | 0.1599 | **22.53** | **15.88** | **37.42** | **63.50** | **74.17** | -1.36 | 0.26 |
| RGW | 53.85 | **75.87** | 0.3772 | 0.1562 | 23.67 | 17.24 | 34.62 | 60.49 | 71.75 | -0.62 | 1.03 |
| MoCo | **54.05** | 75.58 | 0.3812 | **0.1530** | 23.39 | 16.69 | 35.65 | 61.68 | 72.60 | -1.47 | 0.18 |
| MoDo (ours) | 53.37 | 75.25 | **0.3739** | 0.1531 | 23.22 | 16.65 | 35.62 | 61.84 | 72.76 | **-1.59** | **0.07** |

Table 16: Summary of hyper-parameter choices for MoDo

| | Office-31 | Office-home | NYU-v2 |
|---|---|---|---|
| optimizer of $x_t$ | Adam | Adam | Adam |
| $x_t$ step size ($\alpha_t$) | $1 \times 10^{-4}$ | $5 \times 10^{-4}$ | $2.5 \times 10^{-4}$ |
| $\lambda_t$ step size ($\gamma_t$) | $10^{-6}$ | $10^{-3}$ | $10^{-3}$ |
| weight decay | $10^{-5}$ | $10^{-5}$ | $10^{-5}$ |
| batch size | 128 | 128 | 8 |

