# OpenReview forum: "Three-Way Trade-Off in Multi-Objective Learning: Optimization, Generalization and Conflict-Avoidance"
_NeurIPS.cc/2023/Conference — NeurIPS 2023 poster_

### Official Review · Reviewer_UWFs · 2023-06-28

**Soundness:** 3 good
**Presentation:** 3 good
**Contribution:** 3 good
**Rating:** 5
**Confidence:** 3

**Summary:**

This paper has two contributions:
(1) MoDo algorithm which is a variant of MGDA with a double sampling to obtain an unbiased stochastic estimate of the gradient problem.
(2) A solid theoretically analysis on the error of multi-objective optimization.


**Strengths:**

This paper has a very detailed analysis on the optimization error and generalization error of multi-objective optimization.

**Weaknesses:**

In the experiments, authors only compare MoDo with MGDA, but there are many other algorithms, like CAGrad, GradNorm, Uncertainty Weight. This baseline is not enough to demonstrate the effectiveness of the proposed method.

Overall, MoDo is better than MGDA. However, in two objectives out of three (table 2), MGDA is better than MoDo, also not convincing enough on the effectiveness of the proposed method.

**Questions:**

Could you summarize and simply explain the advantages of MoDo over MGDA here, smaller error or more efficient?

Why MoDo has an unbiased gradient estimation? line 142

**Limitations:**

Authors only discuss the situation of using Pareto optimal to get the aggregated gradient.
However, there are other methods other than Pareto to balance multi-objective optimization like balancing gradient magnitudes like AdaTask, GradNorm.

---

> ### Author Rebuttal · Authors · 2023-08-09
>
> Thanks for acknowledging the strengths of our work. Our point-to-point response to your comments and suggestions follows next.
>
> >**W1.** In the experiments, authors only compare MoDo with MGDA, but there are many other algorithms, like CAGrad, GradNorm, Uncertainty Weight. This baseline is not enough to demonstrate the effectiveness of the proposed method.
>
> See response to **General Response-Q2**.
>
> >**W2.** Overall, MoDo is better than MGDA. However, in two objectives out of three (table 2), MGDA is better than MoDo, but also not convincing enough on the effectiveness of the proposed method.
>
> 1. This may be a **misinterpretation** of the results for the following two reasons.
> This phenomenon is mainly due to trade-offs among different tasks. In the simulation, the loss values of DSLR and Webcam are smaller than that of Amazon. Therefore, MGDA tends to favor these tasks compared to Amazon, leading to significantly worse performance of MGDA on Amazon. This calls for a more holistic measure -- the average performance degradation of the method compared to single-task learners (last column of Table 2). It can be seen that MoDo clearly outperforms MGDA with regard to this measure. Thus, we believe that the bias of MGDA towards some tasks compared to MoDo in Table 2 does not indicate a lack of effectiveness of MoDo, but rather a limitation of MGDA.
> 2. More results in **Appendix D.2 and General Response-Q2** demonstrate the effectiveness of MoDo over MGDA.
>
> >**Q1.** Could you summarize and simply explain the advantages of MoDo over MGDA here, smaller error or more efficient?
>
> **MoDo has advantages than MGDA in both smaller optimization error and better efficiency** because:
> 1) **MoDo is better than the full-batch MGDA in efficiency**, as it does not require computing the full-batch gradients at each iteration but only stochastic estimates of the gradients.
> 2) **MoDo smaller optimization error theoretically than the vanilla stochastic version of MGDA**, as the latter is not guaranteed to converge to Pareto stationarity as proved in [47, Theorem 1] while MoDo does.
>
> >**Q2.** Why MoDo has an unbiased gradient estimation? line 142
>
> This is because if we take expectation w.r.t. stochastic samples of the update over $\lambda_ t$ in (6a), which is $\mathbb{E}_ {z_ {t,1}, z_ {t,2}}[\nabla F_ {z_ {t,1}}(x_ t)^ \top \nabla F_ {z_ {t,2}}(x_ t)\lambda_ t] = \nabla F_ S(x_ t)^ \top \nabla F_ S(x_ t)\lambda_ t$, equal to the full batch gradient of problem (4b). See also **General Response-Q3**.
>
>
> >Limitations. Authors only discuss the situation of using Pareto optimal to get the aggregated gradient. However, there are other methods other than Pareto to balance multi-objective optimization like balancing gradient magnitudes like AdaTask, GradNorm.
>
> We will include more discussion of AdaTask, GradNorm [4] in the related works section. See also the new empirical comparison in **General Response-Q2**.
>
> We hope that our responses have addressed your questions. Thank you again!

---

> > ### Author Response · Authors · 2023-08-16
> > **A kind request of your feedback**
> >
> > Dear Reviewer UWFs,
> >
> > Thank you very much for your review. While the discussion period has started several days ago, we have not received your feedback on our response. We believe we have addressed all your concerns including:
> > - Comparison with other baselines
> > - Advantages of MoDo over MGDA
> >
> > We kindly request your feedback on whether our response resolves your concerns. Your additional comments would be invaluable to us!
> >
> > Sincerely, Authors

---

> > ### Comment · Reviewer_UWFs · 2023-08-17
> > **following questions**
> >
> > Thanks for the reply of the authors!
> >
> > I feel my original rating is too low but I need to confirm some points:
> >
> > (1) The motivation of stochastic MOO algorithms? Why we want stochastic MOO? because the true gradient or full gradient is hard to get (as claimed in MoCo)?
> >
> > I have this question because MGDA and CAGrad has been introduced and worked before these stochastic MOO algorithms, I wonder how did MGDA or CAGrad to get objective gradients? maybe they just use the gradient of a mini-batch? hard to believe MGDA or CAGrad was using a full gradient (the gradient of the whole dataset)?
> >
> > yeah, this is the github (https://github.com/isl-org/MultiObjectiveOptimization/blob/master/multi_task/train_multi_task.py) of MGDA from the original authors, seems MGDA is using mini-batch gradient
> >
> > (2) In your "general response", you mentioned "vanilla mini-batch MGDA does not converge", may I ask where does this come from? any proof or empirically results?
> > note: I am not the author of MGDA, just wonder how does this multi-gradient method really work in practice

---

> > > ### Author Response · Authors · 2023-08-17
> > > **Response to following questions**
> > >
> > > Dear Reviewer UWFs,
> > >
> > > Thanks a lot for your prompt reply! Below are our answers.
> > >
> > > >**Q1.** The motivation of stochastic MOO algorithms? Why we want stochastic MOO? because the true gradient or full gradient is hard to get (as claimed in MoCo)?
> > >
> > > The motivation is that full-batch gradient requires large memory or computation and is impractical to get for large-scale problems.
> > >
> > > As the reviewer correctly pointed out, most of the implementations of practical multi-task learning algorithms use stochastic (mini-batch) gradient instead of the true / full gradient.
> > > This can be observed in the github of MGDA you mentioned, and also the github of CAGrad. However, the algorithm design and its analysis work only under the deterministic setting, which generates a significant gap. Note that this gap is not purely theoretical but also of practical relevance as evidenced in your Q2 below.
> > >
> > >
> > >
> > > >**Q2.** In your "general response", you mentioned "vanilla mini-batch MGDA does not converge", may I ask where does this come from? any proof or empirically results? note: I am not the author of MGDA, just wonder how does this multi-gradient method really work in practice
> > >
> > > Yes, this has been proved in recent works [47, Theorem 1] and [25, Section 4], which states that "There is a stochastic convex optimization problem for which MGDA, PCGrad, CAGrad do not converge to the Pareto optimal solution." Numerical examples to demonstrate this non-convergence phenomenon can be also found in Figure 1 of [47]，Figure 2 of [25], and Figure 3 of [8].
> > >
> > > However, prior works of MOO optimization convergence analysis (e.g. the analysis provided in CAGrad) analyzed the deterministic version of the algorithms, but implemented the stochastic  mini-batch gradient. This motivates a recent line of works that design different stochastic variants of MGDA algorithms [8,25,47], and provide their theoretical optimization convergence analysis.
> > >
> > >
> > >
> > > [25] Suyun Liu and Luis Nunes Vicente. "The Stochastic Multi-gradient Algorithm for Multi-objective Optimization and its Application to Supervised Machine Learning." Annals of Operations Research, 2021.
> > >
> > > [47] Shiji Zhou et al. "On the Convergence of Stochastic Multi-Objective Gradient Manipulation and Beyond." NeurIPS 2022
> > >
> > > [8] Heshan Fernando et al. "Mitigating gradient bias in multi-objective learning: A provably convergent stochastic approach." ICLR 2023
> > >
> > >
> > > ---
> > >
> > > Thank you very much for engaging in the discussion! We really appreciate your feedback and hope our answer resolves your questions. We would be happy to answer your following questions if there are any.

---

> > > > ### Comment · Reviewer_UWFs · 2023-08-17
> > > > **rating raised from 4 to 5**
> > > >
> > > > Thanks for the reply!
> > > > With the better understanding of this paper, I raised the rating

---

> > > > > ### Author Response · Authors · 2023-08-17
> > > > > **Thank you for your detailed comments**
> > > > >
> > > > > Thank you very much for your detailed comments! We will make sure to incorporate your comments in the revision.

---

### Official Review · Reviewer_L3j5 · 2023-06-29

**Soundness:** 3 good
**Presentation:** 4 excellent
**Contribution:** 3 good
**Rating:** 6
**Confidence:** 4

**Summary:**

This work considers the multi-objective learning problem. The classic idea of dynamic weighting in MOL is to take gradients from each objective and to weight them using a fixed procedure to avoid conflicts between different objectives. Empirically, however, there often seems to be performance degradation when using these methods. They discover that this is due to a tradeoff between optimization, generalization, and conflict avoidance. They propose a new algorithm, MoDo, that interpolates between static weighting and dynamic weighting and find parameters that can control this tradeoff effectively.

**Strengths:**

Overall, this is a strong work that provides a lot of insight into dynamic weighting in multi-objective optimization. The highlights are as follows:
- The paper is very intuitively written and easy to follow. Specifically, the three-way tradeoff is clear both intuitively and quantitatively. In addition, Figures 1 and 2 are very well done and extremely insightful.
- The proposed MoDo algorithm is very simple and intuitive while brilliantly highlighting the three-way tradeoff inherent in dynamic weighting algorithms. I found the remark about being in the early stopping regime for generalization error to diminish (T = o(n)) interesting, and it is also empirically highlighted later in the work.
- The tradeoffs for both strongly convex and nonconvex cases are analyzed
- The findings in this work translate to a practical explanation of the behavior of dynamic weighting algorithms

**Weaknesses:**

- While MoDo is a great theoretically inspired algorithm that controls the aforementioned tradeoffs, I am not sure how well it performs empirically. While there is an empirical result, the tasks seem rather simple. In addition, as Table 2 shows, MoDo does not consistently outperform MGDA. In practice, I feel handling the conflict avoidance tradeoff shouldn't matter much as long as the accuracy for the task is good. Therefore, I am not sure how applicable the algorithm would be in practical situations.
- As the authors mention in their limitations section, the analysis is only about one specific algorithm in one specific setting, which makes it unclear how general the principles in this work would apply when the assumptions are relaxed. I feel this given this work is meant to analyze a practical phenomenon, this is a substantial weakness, as it is unlikely that Assumptions 1-3 are all true in real-world settings (such as the image classification setting the authors test on)
- There is no intuition of the proof in the main text, and while this is common in optimization papers, it would be useful to learn about the key insights towards the proof.


**Questions:**

- Can you compare the theoretical tradeoffs in Table 1 with the empirical findings in Figure 4? How well do they match with each other?
- How can knowledge of this three-way tradeoff be used to design better algorithms in multi-task learning in the future?
- How can we tune the parameters in practice? How sensitive is the algorithm to the parameters?

**Limitations:**

Great!

---

> ### Author Rebuttal · Authors · 2023-08-09
>
> Thanks for recognizing our work as a strong one! We will respond to the weaknesses and questions point by point as follows.
>
> > **W1.** Empirical benefit of MoDo.
>
> We have more results in the submitted **Appendix D.2** to demonstrate its better performance on other datasets. Also see **General Response-Q2** for more results.
>
> > **W2.** Strong assumptions compared to practical phenomenon.
>
> It is a **misunderstanding** that our theory requires ''Assumptions 1-3 are all true''. In fact, we only generally require smoothness of the functions (Assumption 1), and provide separate discussions for nonconvex case (under Assumptions 1 and 3), and the discussion for strongly convex case (under Assumptions 1 and 2) for both optimization and generalization.
> **These assumptions are standard in prior optimization analysis [8,25,47].**
> The general smooth non-convex case covers a lot of practical problems such as using neural networks with ELU activation functions. And the strongly convex smooth case covers, e.g., the linear fine-tuning problem with l2-norm regularizer.
>
> > **W3.** No intuition of the proof in the main text.
>
> Due to space limitation, we defer the intuition of the proof to the Appendix in each section.
> The key insights of the proof are summarized as follows.
>
> 1. For **generalization**, the key is to use the error of the algorithm output resulting from perturbation of training data (stability) to bound the expected difference of performance on the testing and training data (generalization). Then using the property of stochastic sampling and the update function, the expected error of the current iteration caused by the dataset perturbation can be bounded recursively by a linear function of that of the previous iteration.
> See also **Summary of Theoretical Contributions-T2**.
> 2. For **optimization**, the key is to use the descent Lemma derived from smoothness of the functions, which shows the function value approximately decreases after each update, with an approximation error caused by the dynamically changing weight. Then we are able to bound this approximation error using the property of the update of the dynamic weight based on the convexity of the subproblem.
> See also **Summary of Theoretical Contributions-T3**.
>
> > **Q1.** Comparison of Table 1 and Figure 4.
>
> Yes, we have discussions and comparisons in Section 5.2; see line 313-329 in the main text. Since Figure 4 corresponds to the general nonconvex case, it matches the result described in Table 1, line 1 for the nonconvex (NC) case.
>
> > **Q2.** How can knowledge of this three-way tradeoff be used to design better algorithms in multi-task learning in the future?
>
> As discussed in Section 3.3, Appendix D.1.2 in our paper, our theory (Theorems 1,2) suggests that, when chosing dynamic weighting at each iteration, it is better that **the drift of the dynamic weights is not large across iterations**, to ensure better test performance.
>
> If the dynamic weights are iteratively updated, this can ensured by
> - having appropriately small learning rate for the dynamic weights;
> - do not update dynamic weight at each iteration, but rather update the dynamic weight only when there are conflicts in gradients.
>
> > **Q3.** How can we tune the parameters in practice? How sensitive is the algorithm to the parameters?
>
> We have included hyperparameter choice and sensitivity analysis in **Appendix D**. In practice, $T$ should not be too large to ensure good generalization performance. $\alpha, \gamma$ depend on the choice of $T$. For example, $\alpha = \mathcal{O} (T^{-\frac{1}{2}})$ works well, and a relatively small $\gamma$ works well in practice. Furthermore, overall performance of the algorithm is mostly sensitive to iterate step size $\alpha$, while individual task-trade offs are mostly sensitive to dynamic weighting step size $\gamma$.
>
> ---
> We hope that our responses to your comments are satisfactory. Thank you again!

---

> > ### Comment · Reviewer_L3j5 · 2023-08-15
> >
> > Thanks for the comments and clarification. Previously I was on the 5-6 borderline, but after the clarifications I still maintain that 6 is a better choice, as I still believe this work is technically solid and would have moderate impact. My main concern is still that it is not clear how these insights can be applied for future advances in multi-objective optimization.

---

> > > ### Author Response · Authors · 2023-08-15
> > > **Response to the concern of how to apply the insights in this paper**
> > >
> > > ## How these insights can be applied for future advances in multi-objective optimization
> > > Thank you very much for your quick response.
> > > It is an excellent suggestion, and in fact our ongoing work to apply the insights in this work for future advances in multi-objective optimization (MOO).
> > > Here's a breakdown of how the insights can be applied in each of the areas.
> > >
> > > **1. Theoretical Applications:**
> > > - **Analyzing other MOO algorithms.** This cannot be achieved by simply combine prior works solely on optimization and generalization analysis because they are often focusing on different settings. Applying this theoretical framework, we could also analyze the three types of errors for other dynamic weighting MOO algorithms such as MoCo, CAGrad, and PCGrad.
> > > Specifically, for generalization, Propositions 2, 3 still hold. Since the stochastic implementations of PCGrad and CAGrad are sampling determined (Definition 3), the bound in the NC case (Theorem 1) holds. Combining the optimization and generalization error bounds, we can find better hyperparameters in PCGrad and CAGrad to minimize the test risks.
> > >
> > > - **Studying benefits of MOO algorithms over static weighting.**
> > > Prior theoretical works of MOO mostly focus on optimization convergence to Pareto stationarity. However, this can also be guaranteed by static weighting. Therefore, the theoretical benefits of MOO algorithms over static weighting remain open. Our analysis addresses this critical question by demonstrating the advantages of MOO algorithms in CA distance reduction. This provides a justification for their use in cases where CA distance reduction is crucial.
> > >
> > > **2. Practical Applications:**
> > >
> > > - **Hyperparameter choice:**
> > > Our theory is suitable for analyzing the effect of hyperparameters such as step size and number of iterations on the three errors, and the total testing risk. This allows us to find better hyperparameters to minimize the test risks. For example, when performing grid search of hyperparameters for MoDo, we focus on a range with relatively small $\gamma$.
> > > - **Algorithm choice:**
> > > Comparing error bounds of different algorithms allows for a more informed algorithm selection process based on the nature of the problem. The choice of algorithm depends on which error dominates the performance.
> > > For example, if in a problem the CA distance is a major factor that prevents the algorithm to achieve good performance, then we choose to use the algorithm with the smallest CA distance. And if the generalization error dominates, we could choose e.g., MoDo with a small step size $\gamma$ or static weighting to obtain better testing performance.
> > > - **Algorithm design:**
> > > Our theory could inspire the development of new MOO algorithms that strike a better balance between among the three types of errors.
> > > For example, it suggests large drift of $\lambda$ could degrade the test error. Therefore, future algorithms could be designed to update $\lambda$ not at each iteration, but only when there is significant conflict in gradients, e.g., the angle between two gradients is larger than 90 degrees. This could potentially improve both the test error and the efficiency of the MOO algorithm.
> > >
> > > ---
> > > Following your excellent suggestion, we will also incorporate this in our discussion of future work in the revision.

---

### Official Review · Reviewer_LuAm · 2023-06-30

**Soundness:** 2 fair
**Presentation:** 3 good
**Contribution:** 3 good
**Rating:** 4
**Confidence:** 3

**Summary:**

This paper studies three-way trade-off in multi-objective learning: 1) optimization error caused by sampling and stochastic training; 2) generalization error that measures the difference between source and target sets; 3) conflict-avoidance direction error that is the bias between the calculated direction and the right one. The authors propose the MoDo algorithm to optimize this trade-off in one algorithm. Plenty of theoretical analysis supports the claims.

**Strengths:**

1. This paper considers the multi-objective algorithmic design in a bigger picture by compositing three-way trade-off together, and provides solid analysis to solve it.

2. This paper is well written. In particular, Table 1 and Figure 2 help to understand the contribution better.

**Weaknesses:**

1. The proposed algorithm needs to compute three batches of gradients to run an iteration, which is much less computationally efficient.

2. Benchmarks are too limited that only contain static and MGDA (2018 proposed), experiments have not compared with other typical methods like PCGrad and CAGrad, which have better empirical performance.

3. Several claims are out-of-date. In 34-35 "Unfortunately, the reason behind this empirical performance
degradation is not fully understood and remains an open question" is not accurate, since [1] has proved that the vanilla MGDA algorithm and also PCGrad and CAGrad will not converge to Pareto optimal. In 268 "This can be overcome by increasing the batch size during optimization [25]" is wrong, because the assumption on Lipschitzness for $\lambda$ with respect to gradients has been proved to be wrong (Proposition 2 in [1]). Also, it can be proved that even a stochastic error of the gradient can cause a significant bias in the direction, so only increasing the batch size does not work.

[1] S Zhou, W Zhang, J Jiang, W Zhong, J Gu, W Zhu - Advances in Neural Information Processing Systems, 2022

**Questions:**

1. Why use double sampling? Can MoCo completely solve the problem? Or is it possible to reduce the sample complexity?

2. Does the proposed theory can help improve PCGrad and CAGrad?

**Limitations:**

Sample efficiency of the proposed algorithm.

---

> ### Author Rebuttal · Authors · 2023-08-08
>
> Thanks for acknowledging the strengths of our work. Our point-to-point response to your comments follows next.
>
> > **W1 & Q1-3 & Limitation.** MoDo has limitation on sample efficiency. Is it possible to reduce improve?
>
> - Theoretically, MoDo is not necessarily worse on total sample complexity; see our **General Response-Q1**.
>
> - Practically, MoDo does not require more samples, because the batch size can be controlled to keep the sample size per iteration $n'$ the same.
> E.g., for the experiments in **Appendix D and General Response-Q2**, MoDo generally performs better than the baselines under the same $T\times n'$.
>
> In short, **MoDo could have better total sample complexity theoretically and practically**. While there may be room for improvement, it is beyond the scope of this paper.
>
> >**W2.** Limited benchmarks.
>
> See more results in **General Response-Q2**.
>
> >**W3.** Outdated claims. ([1] is replaced with [47] as indexed in our submission)
>
> >**W3-1.** "the reason ... is not fully understood ..." is not accurate. [47] proved...
>
> We agree **[47] partially addresses this from the optimization perspective**.
> However, "empirical performance degradation" in this context refers to the observed phenomenon of **test performance** being often worse than static weighting [15,40]. The reasons for this degradation are not fully understood theoretically, as prior works, such as [47], have focused solely on analyzing the theoretical training (optimization) error, NOT the generalization (test) error.
> In fact, it has been demonstrated in [40, Figure 2] that the training (optimization) errors are all relatively small, while the generalization performances differ. Therefore, **only analyzing the optimization error is not enough to fully understand the test performance**. Nevertheless, we will follow your great suggestion to desribe related works more precisely and acknowledge the pioneering contribution of [47] in optimization.
>
> >**W3-2.** "...can be overcome by increasing the batch size[25]..." is wrong, because the assumption...in [25] is wrong...
>
> This is a **misunderstanding**. We intend to convey that "the bias in the CA direction can be mitigated by increasing the batch size". We agree the assumption on Lipschitz $\lambda^*(x)$ in [25] is wrong. But **this is a limitation of their proofs, NOT this claim**. In fact, for $Q\in \mathbb{R}^{d\times M}$, although $\lambda^*(Q)\in \arg\min_{\lambda\in \Delta^M}||Q\lambda||^2$ is not Lipschitz continuous w.r.t. $Q$, it can be proved that the update direction is $\frac{1}{2}$-Holder continuous w.r.t. $Q$ as stated below.
> $$||Q\lambda^*(Q)-Q'\lambda^*(Q')||^ 2
> \leq 4\max (||Q||,||Q'||)||Q - Q'||.$$
> Plugging in $Q = \nabla F_S(x)$, $Q' = \nabla F_Z(x)$ with $Z$ being a stochastic batch of $S$, we have
> $$
> ||\nabla F_S(x)\lambda^*_S(x)- \mathbb{E}_Z[\nabla F_Z(x)\lambda^*_Z(x)]||^2
> \leq\mathbb{E}_Z||\nabla F_S(x)\lambda^*_S(x) - \nabla F_Z(x)\lambda^*_Z(x)||^2
> \leq 4 \ell_F \mathbb{E}_Z||\nabla F_S(x) - \nabla F_Z(x)||=O(1/\sqrt{|Z|}),
> $$
> which decreases as the batch size $|Z|$ increases. Therefore, the bias in the CA direction can be reduced by increasing the batch size.
>
> We will follow your great suggestion to revise it as "One challenge ... is the bias in the CA direction, which can be mitigated by increasing the batch size ...", and **acknowledge [47] is the first work to prove the assumption is wrong in prior work [25], and propose a bias reduction scheme to address this issue**.
>
> >**Q1-1.** Why double sampling?
>
> It is one way to mitigate the gradient bias without the momentum-based methods. See **General Response-Q3**.
>
> > **Q1-2.** Can MoCo solve the problem?
>
> If the "problem" is to reduce bias in optimization, MoCo can solve it. But the "problem" we study in this paper, the theoretical test risk, is **different**. MoCo **CANNOT completely solve** the test problem since it only provides guarantee for optimization error but not for generalization error. Even for optimization, MoDo has improved sample complexity; see **General Response-Q1**.
>
> >**Q2.** Does the proposed theory help improve PCGrad and CAGrad?
>
> Great suggestion! We summarize 2 ways to improve PCGrad and CAGrad based on our theory.
>
> **1. Controlling the change of $\lambda_t$ can improve generalization and achieve the best trade-off.**
> Our theory suggests, while dynamic weighting enables conflict avoidance, large drift of weights during update may degrade generalization. Thus, controlling this drift in PCGrad and CAGrad could improve their performance.
>
> **2. The theoretical framework for MOL is general, and can be used for other dynamic weighting methods including PCGrad and CAGrad.**
> E.g., for generalization, Propositions 2, 3 still hold. Since the stochastic implementations of PCGrad and CAGrad are sampling determined (Definition 3), the bound in the NC case (Theorem 1) holds. Combining the optimization and generalization error bounds, we can find better hyperparameters in PCGrad and CAGrad to minimize the test risks.
>
> ---
> We hope this can address your questions and raise the score. Thank you again!

---

> > ### Author Response · Authors · 2023-08-16
> > **A kind request of your feedback**
> >
> > Dear Reviewer LuAm,
> >
> > Thank you very much for your review. While the discussion period has started several days ago, we have not received your feedback on our response. We believe we have addressed all your concerns including:
> >
> > - Comparison with other baselines
> > - Sample efficiency of MoDo
> > - More precise and detailed discussion and comparison of the pioneering contribution of [47] in stochastic multi-objective optimization
> >
> > We kindly request your feedback on whether our response resolves your concerns. Your additional comments would be invaluable to us!
> >
> > Sincerely, Authors

---

> > ### Comment · Area_Chair_CsZD · 2023-08-21
> > **Thanks for the rebuttal.**
> >
> > Although the review did not engage, I'll carefully read and consider it during the decision period.
> >
> > AC

---

> > > ### Author Response · Authors · 2023-08-21
> > > **Thanks for the assistance**
> > >
> > > Dear Area Chair,
> > >
> > > Thank you very much for your efforts in the review process. We would be happy to provide any additional information if asked.
> > >
> > > Sincerely, Authors

---

### Official Review · Reviewer_CY8e · 2023-07-02

**Soundness:** 3 good
**Presentation:** 3 good
**Contribution:** 3 good
**Rating:** 7
**Confidence:** 3

**Summary:**

This paper studies the multi-objective optimization, and in particular, focus on the generalization and stability analysis. By decomposing the Pareto stationarity error into the generalization and optimization error, the authors then analyze and upper-bound these two errors respectively. The distance to the conflict-avoidant direction, which optimizes al objectives jointly, is also analyzed. A stochastic variant of MGDA named MoDo has been developed and analyzed. Based on the theoretical derivations, the authors find a three-way tradeoff among the optimization error, generalization error and the CA error. Some implications on the parameter selection and illustrations are provided.

**Strengths:**

1. The studied topic on multi-objective optimization has received increasing attention thanks to the important applications like multi-task learning. Studying the generalization, stability and the tradeoff is important and under-explored.

2. Although generalization has been studied for multi-objective optimization, stability has not been explored. This work seems to be the first one to fill this gap.

3. The tradeoff among optimization and generalization is not surprising, but their tradeoffs with the CA direction error seem to be new given the multi-objective structure.

**Weaknesses:**

1. How is (4a) equivalent to (4b)? Can the authors provide some more details?

2. Before (4a), it says that CA direction maximize the minimum descent of all objectives. But (4a) seems to say that the direction minimizes something.

3. The upper bounds in the nonconvex case may not be very tight given the large exponential dependence, and hence may not be able to exactly capture the generalization and tradeoff behaviors. Is it possible to provide a lower bound in this case? Or what challenges in getting a tight bound?

4. The analysis uses Frobenius norm, which may be large in practice. Instead, spectrum norm may be more proper. Can the authors comment on why use Frobenius norm rather than spectrum norm?

Overall, I appreciate the studied problem and the analysis in this work, but I am also open to other reviewers’ comments.

**Questions:**

See Weakness part.

**Limitations:**

See Weakness part.

---

> ### Author Rebuttal · Authors · 2023-08-06
>
> Thanks for appreciating our problem setup and analysis! We will respond to the weaknesses and questions point by point as follows.
>
> > **W1. & W2.** Derivations of (4a) and (4b).
>
> This is a **standard derivation for the MGDA algorithm**. Similar derivations are provided in [25, Section 3.1], [47, Section 2.2].
>
> Here we also include a derivation as follows. For (4a), the original problem in MOO is to maximize the minimum descent (among all objectives) along the update direction $d$, where the minimum descent given $d$ can be computed by
> $$\frac{1}{\alpha}\min_{m\in [M]} \{f_{S,m}(x) - f_{S,m}(x+\alpha d)\} \approx \min_{m\in [M]} -\langle \nabla f_{S,m}(x), d\rangle
> .$$
>
> Then with regularization on $d$ to control its norm, the problem is formulated as
> $$\max_{d \in \mathbb{R}^d} \min_{m\in [M]} -\langle \nabla f_{S,m}(x), d\rangle - \frac{1}{2}||d||^2 $$
> which is equivalent to (4a) as
> $$\min_{d \in \mathbb{R}^d} \max_{m\in [M]} \langle \nabla f_{S,m}(x), d\rangle + \frac{1}{2}||d||^2 .$$
>
> Then (4a) can be reformulated as
> \begin{align}
> \min_ {d \in \mathbb{R}^d} \max_ {m\in [M]} \langle \nabla f_ {S,m}(x), d\rangle + \frac{1}{2}||d||^2
> & =\min_ {d \in \mathbb{R}^d} \max_ {\lambda \in \Delta^M} \langle \nabla F_ {S}(x)\lambda, d\rangle + \frac{1}{2}||d||^2 \\\\
> &= \max_ {\lambda \in \Delta^M} \min_ {d \in \mathbb{R}^d} \langle \nabla F_ {S}(x)\lambda, d\rangle + \frac{1}{2}||d||^2
> \end{align}
>
> where we use the min-max theorem to change the order of min and max operators in the last equation.
>
> In this way, given $\lambda$, the optimal $d^*(x,\lambda)$ is obtained as $d^*(x,\lambda) = -\nabla F_S(x)\lambda$, and the optimal $\lambda^* \in \arg\min_{\lambda\in \Delta^M} ||\nabla F_S(x)\lambda||^2$. Combining these leads to the formulation (4b).
>
> >**W3.** The upper bounds in the nonconvex case may not be very tight given the large exponential dependence, and hence may not be able to exactly capture the generalization and tradeoff behaviors. Is it possible to provide a lower bound in this case? Or what challenges in getting a tight bound?
>
> This might be a **misunderstanding** of our theorem. Note that **we do NOT have exponential dependence** on the hyperparameters $T$ or $n$, but rather, our general bound is in the order of $\mathcal{O}(T/n)$ in the NC case (see Theorem 1). Therefore, our bound is already tighter compared to the one for single objective learning in [12], which could lead to exponential dependence on $T$ without $1/t$ step size decay even for static weighting.
>
> We overcome the exponential dependence by bounding the probability of selecting the perturbed data in all $T$ iterations and the resulting error, instead of bounding the accumulated expected error in the output parameters  through recursion over each iteration as did in [12].
> The latter could result in exponential dependence on $T$ in the NC case, since the expected error at each iteration increases over the previous one in a rate larger than 1, due to the expansiveness of the update function.
>
> >**W4.** The analysis uses Frobenius norm, which may be large in practice. Instead, spectrum norm may be more proper. Can the authors comment on why use Frobenius norm rather than spectrum norm?
>
> Thank you for the suggestion. There are two main reasons for using Frobenius norm.
>
> **1. Frobenius norm of the gradient matrix, $||\nabla F_ S(x)||_ F$ can be directly derived from the $\ell$-2 norm of each gradient $||\nabla f_ {S,m}(x)||$.**
> Therefore, the $\ell_ F$-Lipschitz continuity of $F_ z(x)$ in Frobenius norm (Assumptions 1) can be direcly derived from the $\ell_ f$-Lipschitz continuity of $f_ {z,m}(x)$ for all $m \in [M]$ with $\ell_ {F} = \sqrt{M}\ell_ {f}$, as discussed in Lemma 1. The latter assumption of $\ell_ f$-Lipschitz continuity of $f_ {z,m}(x)$ for all $m \in [M]$  were standard in prior works such as CR-MOGM [47], MoCo [8] for optimzation analysis.
>
> **2. The spectral norm and Frobenius norm can be used interchangeably up to a factor since for $A\in\mathbb{R}^{d\times M}$, $||A||\leq ||A ||_ F \leq \sqrt{r}||A ||$ where $r\leq M<<d$ is the rank of $A$.**
> In our analysis, $A$ is the gradient matrix $\nabla F_ S(x)$, and $M$, the number of objectives, is a fixed constant and relatively small compared to other factors.
>
> Nevertheless, it would interesting to derive our theory under the spectral norm or other matrix norms given appropriate assumptions.
>
> ---
> We hope that our responses to your comments are satisfactory. Thank you again!

---

> > ### Comment · Reviewer_CY8e · 2023-08-15
> > **Thanks for the response**
> >
> > I thank the reviewer for the detailed response. Based on the global response, could I ask two more questions?
> >
> > 1. Can you elaborate how MoDo achieves the $\epsilon^{-2}$ sample complexity from Theorem 3? Why is there a huge improvement over the SOTA MoCo without bounded function assumption?
> >
> > 2. The Frobenius norm may be still a little bit large. Can the authors show the dependence on the number of tasks in the bounds since it may matter in the MTL?

---

> > > ### Author Response · Authors · 2023-08-15
> > > **Response to additional questions**
> > >
> > > Thank you very much for engaging in the discussion! The answers to your questions are as follows.
> > >
> > > >**Q1-1.** Can you elaborate how MoDo achieves ${\cal O}(\epsilon^{-2})$ sample complexity from Theorem 3?
> > >
> > > For a fair comparison with prior works such as MoCo, we refer to the result provided in Appendix C.3  in page 39, Eq.(183) without taking the square root, which states the optimization error is bounded as
> > > $$\frac{1}{T}\sum_{t=1}^T
> > > \mathbb{E}_ A || \nabla F_ S(x_ t) \lambda_ t^ *(x_ t) ||^2
> > > = O(\frac{1}{\alpha T} + \alpha + \gamma)$$
> > > where choosing $\alpha = \Theta(T^ {-\frac{1}{2}}), \gamma= \Theta(T^ {-\frac{1}{2}})$, we can obtain the optimal rate
> > > $$\frac{1}{T}\sum_{t=1}^T
> > > \mathbb{E}_ A || \nabla F_ S(x_ t) \lambda_ t^ *(x_ t) ||^2
> > > = O(T^ {-\frac{1}{2}})$$
> > > which implies $O(\epsilon^ {-2})$ sample complexity.
> > >
> > >
> > >
> > > >**Q1-2.** Why is there a huge improvement over the SOTA MoCo without bounded function assumption?
> > >
> > > This is mainly due to the difference of the MoDo algorithm with double / independent sampling, and the difference in our proof techniques compared to those in MoCo. Some key ideas of the proof techniques are summarized in **Summary of Theoretical Contributions-T3**.
> > >
> > > To be more specific, in our proof we adopt the following steps:
> > >
> > > **S1.** We first use the descent lemma for $F_ S(x)\lambda$ using a fixed $\lambda$, see Eq.(176) in our Appendix. This allows cancellation of the function values when taking the telescope sum, which cannot be achieved using a dynamic $\lambda_ t$. As a result, the bounded function value assumption needs to be introduced in prior works if using a dynamic $\lambda_ t$ for the descent lemma, see e.g., Eq.(75) in the MoCo paper [8].
> > >
> > > **S2.** Then, the inner product term $\mathbb{E}_ A\langle\nabla F_ S(x_ t) \lambda, \nabla F_ S(x_ t) \lambda_ {t+1}\rangle$ in the inequality Eq.(177) is related with the gradient norm $\mathbb{E}_ A||\nabla F_ S(x_ t) \lambda_ t||^2$ by the property from the update of $\lambda_ t$ in Lemma 16, Eq.(155), which is a nice property with bias reduction due to double sampling in the subproblem.
> > >
> > > **S3.** Finally, taking telescope sum, the optimization error measured in gradient norm $\mathbb{E}_ A||\nabla F_ S(x_ t) \lambda_ t||^2$ can be bounded without introducing additional assumptions.
> > >
> > > Comparing the result without the bounded function value assumption in MoCo, Theorem 2, the inner product term in their Eq.(70), $\langle\nabla f_ m(x_ k), \nabla F(x_ k) \lambda_ k^ *-Y_ k \lambda_ k\rangle$, is bounded by Cauchy-Schwartz inequality, and then by triangle inequality, which boils down to bounding the moving average gradient error $||Y_ k - \nabla F(x_ k)||$, the dynamic weight approximation error $||\lambda_ k -\lambda_ {\rho,k}^ *||$, etc. This may not be as tight as our **S2** directly using the property in Lemma 16 from double sampling in the subproblem.
> > >
> > >
> > > >**Q2.** The Frobenius norm may be still a little bit large. Can the authors show the dependence on the number of tasks in the bounds since it may matter in the MTL?
> > >
> > > Sure, thanks for the insightful suggestion! We have restated below the bounds with explicit dependence on the number of tasks $M$.
> > >
> > > **Theorem 1:** $\epsilon_F^2 = O\Big(\frac{MT}{n}\Big)$, $\mathbb{E}_ {A,S}[R_ {\rm gen}(A(S))] = O(M^ {\frac{1}{2}} T^ {\frac{1}{2}} n^ {-\frac{1}{2}} )$, since $G^2 = O(M)$.
> > >
> > > **Theorem 2:** $\epsilon_F^2 = O\Big(\frac{M}{n}(\alpha + \frac{M}{n} + M\gamma)\Big)$, and $\mathbb{E}_ {A,S}[R_{\rm gen}(A(S))] = O(M^{\frac{1}{2}} n^ {-\frac{1}{2}} )$, when $M\leq n$ and $M\gamma \leq 1$.
> > >
> > > **Theorem 3 (without square root):**
> > > $\frac{1}{T}\sum_{t=1}^T
> > > \mathbb{E}_A || \nabla F_S(x_t) \lambda_t^*(x_t) ||^2
> > > = O(\frac{1}{\alpha T} + \alpha + M^{\frac{3}{2}}\gamma)$
> > > where $\gamma$ can be controlled to ensure $M^{\frac{3}{2}}\gamma$ to be small.
> > >
> > > We will also include discussion of the dependence on $M$ in our revision, and further improve the bound using other matrix norms in our future work.

---

> > > > ### Comment · Reviewer_CY8e · 2023-08-16
> > > > **Thanks for the clarification**
> > > >
> > > > I thank the authors for the clarification. I increase my score to 7.
> > > >
> > > > Best,
> > > > Reviewers

---

> > > > > ### Author Response · Authors · 2023-08-16
> > > > > **Thanks for your constructive comments**
> > > > >
> > > > > Thank you again for engaging in the discussion and providing constructive comments! We will incorporate the suggestion in our revision.

---

### Author Response · Authors · 2023-08-08
**Summary of Theoretical Contributions**

### Summary of Theoretical Contributions

We provide the **first ever known stability-based generalization of MOL**. This is a non-trivial extension of the results in single objective learning [12], which cannot be directly applied to MOL due to several challenges as follows.

**T1.** The **measure of Pareto Stationary (PS) risk** defined in  gradients (line 108, Eq.(2)) overcomes a key *challenge* brought by the classical function value-based risk measures -- the **unnecessarily small step size choice**.
Specifically, it requires $1/t$ step size decay in the nonconvex case, otherwise the generalization error will depend exponentially on $T$. But such choice leads to very slow convergence of the optimization error. This is addressed by the definitions of gradient based measures and sampling determined MOL algorithms, which yields a stability bound in ${\cal O}(T/n)$ without any step size decay.
See more discussions in line 175-177 below Theorem 1.

**T2.** The **generalization bound using the algorithmic stability approach** (Theorems 1,2) is the first on generalization and stability analysis for MOL, and is non-trivial compared to SGD because it involves two coupled sequences. The *challenge* is that the **classical contraction property that is often used to derive stability in the SC case does not hold**. This is addressed by controlling the change of $\lambda_ t$ by the step size $\gamma$ to derive a tighter bound.
See more discussions in Appendix B.4, line 660-672.

**T3.** The **optimization error bound** (Theorem 3) is also improved with **either relaxed assumption or improved convergence rate**. More details are summarized in **General Response-Q1**.
This is achieved by 1) instead of bounding the approximation error to $\lambda^ *(x_ t)$, we bound that to the update direction $\nabla F_ S(x_ t)\lambda^ *(x_ t)$ as a whole, and 2) instead of using the descent lemma of $F_ S(x_ t)\lambda_ t$ with a dynamic weight, we use that of $F_ S(x_ t)\lambda$ with a fixed weight, thereby improving the tightness of the bound.
See more discussions in Appendix C.3.

We would like to further explain the technical challenges and how we address them in the proof if the reviewers are interested. We look forward to further engagement with the reviewers and area chair!

Sincerely,

Authors

---

### Author Rebuttal · Authors · 2023-08-08

## General Response

We appreciate the reviewers' constructive comments. All reviewers agree the paper has made solid theoretical contributions, and it has "a bigger picture" concerning three types of errors -- optimization, generalization, and CA distance unique in MOL in a holistic framework. It also provides the first algorithm stability-based generalization bound for MOL.

Despite this, we want to re-emphasize that **our standpoint is not a new algorithm that improves over SOTA MOL algorithms**. Instead, we propose a framework for theoretical comparisons of stochastic MGDA and static weighting, complementing to the recent empirical studies [15,40] in NeurIPS 2022; see our **"Summary of Theoretical Contributions"** in the separate thread below.

The reason we use **MoDo as a stochastic variant of MGDA** is that it
- is the simplest possible unbiased stochastic variant of MGDA (as vanilla mini-batch MGDA does not converge) which has theoretical guarantees (with better rates or relaxed assumptions than [8,26,47]);
- interpolates between static and dynamic weighting, thus flexibly controlling the trade-off, which cannot be achieved by existing methods [24,25,36,44].



 Below we address three major questions from the initial reviews.

>**Q1. The benefit of MoDo in sample efficiency or error. (Reviewers LuAm, UWFs)**

Regarding the benefit, **MoDo does have benefit in terms of theoretical guarantee of sample efficiency  and  test risk**, which we explain below.

1. **In terms of sample efficiency, the overall sample complexity of MoDo could be better than other MOO variants.** Although MoDo requires computing 3 independent stochastic gradients at each iteration, it does not necessarily harm the total sample efficiency as questioned by Reviewer LuAm. This is because the **convergence rate of MoDo is faster under the same assumptions**; see the Table below. When the bounded function values assumption is removed, MoDo achieves $\mathcal{O} (\epsilon^{-2})$, significantly improving MoCo with $\mathcal{O}(\epsilon^{-10})$.
The overall sample complexity is:

```
(number of iterations T) X (number of samples per iteration n') = Tn'.
```

Since MoDo improves the number of iterations $T$ to achieve small optimization error, it could improve the overall sample complexity $T\times n'$ when $T$ dominates.

|Algorithm|Assume bounded function values|Sample complexity
|---|---|---|
|CR-MOGM [47, Theorem 3]|Yes|${\cal O} (\epsilon^{-2})$
|MoCo [8, Theorem 2]|No|${\cal O} (\epsilon^{-10})$
|MoCo [8, Theorem 4]|Yes|${\cal O} (\epsilon^{-2})$
|MoDo (Ours, Theorem 3)|No|${\cal O} (\epsilon^{-2})$

2. **In terms of test risk, MoDo has theoretical guarantee of the generalization error, thus the total test risk**, but the generalization error guarantees of MGDA, MoCo and CR-MOGM are unknown or at least not established.

>**Q2. Limited empirical evaluation or recent benchmarks. (Reviewers LuAm, UWFs, L3j5)**

1. **Misunderstanding of our key contributions.** It is worth mentioning again that **we do not claim that MoDo works better than the SOTA algorithms on multi-task learning**. Therefore, it is not our main target to demonstrate MoDo performs better than SOTA algorithms for MTL.

2. **Reasons for current baselines.** We choose static weighting as an important baseline because it has been demonstrated in recent works [15, 40] that on many tasks, static weighting outperforms the recent MOL-based algorithms such as PCGrad and CAGrad.

3. **Other benchmarks as per reviewers' requests.** We compare to PCGrad, CAGrad, and MoCo, etc. following the experimental settings in [arxiv:2111.10603]. Results are summarized in the **attached PDF, Tables 1-3** and will be included in the revision. The results show that MoDo performance is comparable or better to these baselines.

>**Q3. Why use double sampling, how it mitigates gradient bias (Reviewers LuAm, UWFs)**

The **CA distance or gradient bias** can be mitigated since MoDo uses **double/independent sampling** for update. Specifically, taking expectation w.r.t. stochastic samples of the update over $\lambda_t$ in (6a) leads to
$$\mathbb{E}_ {z_ {t,1}, z_ {t,2}}[\nabla F_ {z_ {t,1}}(x_ t)^\top \nabla F_ {z_ {t,2}}(x_ t)\lambda_ t] =
\mathbb{E}_ {z_ {t,1}}[\nabla F_ {z_{t,1}}(x_ t)]^\top \mathbb{E}_ {z_ {t,2}}[\nabla F_ {z_{t,2}}(x_ t)]\lambda_ t = \nabla F_S(x_t)^\top \nabla F_S(x_t)\lambda_t,$$
which is equal to the full batch gradient of problem (4b). This allows us to derive the convergence of distance to CA direction in Lemma 2 using the stochastic optimization framework.

However, if we use the same sample for $z_ {t,1}$ and $z_ {t,2}$, this equality does not hold, i.e.,
$$\mathbb{E}_ z[\nabla F_ z(x_ t)^\top \nabla F_ z(x_ t)\lambda_ t] \neq \nabla F_S(x_ t)^\top \nabla F_ S(x_ t)\lambda_ t.$$
As a consequence, the distance to CA direction cannot be reduced due to this bias.

We would like to make the most use of the interactive discussion function provided by Openreview to clarify any concern the reviewers may have. We look forward to the rolling dicussion and further engagement with the reviewers and area chair!

Sincerely,

Authors

---

### Author Response · Authors · 2023-08-15
**Looking forward to reviewers' feedback**


Dear AC and reviewers,

We would like to start by thanking all reviewers for the positive feedback and constructive comments given in the initial reviews. While the discussion period has started several days ago, we have not received any feedback based on our responses. We would like to use the interactive feature of OpenReview to engage the reviewers with the discussion.

In particular, in our responses, we believe we have provided

a1) summary of our theoretical contribution;

a2) additional clarifications on algorithm designs; and

a3) new numerical results with more baselines;

to fully address the reviewer's concerns. We hope our responses convince the reviewers about the merits of this work. If the reviewer has any other suggestions or comments, please don't hesitate to let us know!

Best Regards,

Authors

---

### Decision · Program_Chairs · 2023-09-21

**Decision:**

Accept (poster)

**Comment:**

The manuscript investigates multi-objective optimization with a particular emphasis on stability analysis. The work ultimately unveils a three-way tradeoff between optimization error, generalization error, and the conflict-avoidance. Empirical studies complement the theoretical results. The paper was reviewed thoroughly with a committee and the consensus recommended acceptance. The reviewer recommending rejection raised some valid concerns. After reading the paper as well as the reviews/rebuttal carefully, I agree with the consensus. However, I also recommend authors to improve their manuscript with the following aspects:
- Even if the empirical study covering other methods like PCGrad, CAGrad, GradNorm,... is not necessary, at least discussion on the implication of the theoretical results for these methods would be beneficial.
- There were unclear parts about the theory which authors successfully clarified during rebuttal. These clarifications need to be reflected in the camera-ready version.